# How Important is the Train-Validation Split in Meta-Learning?

## Abstract

Meta-learning aims to perform fast adaptation on a new task through learning a "prior" from multiple existing tasks. A common practice in meta-learning is to perform a *train-validation split* where the prior adapts to the task on one split of the data, and the resulting predictor is evaluated on another split. Despite its prevalence, the importance of the train-validation split is not well understood either in theory or in practice, particularly in comparison to the more direct *non-splitting* method, which uses all the per-task data for both training and evaluation.

We provide a detailed theoretical study on whether and when the train-validation split is helpful on the linear centroid meta-learning problem, in the asymptotic setting where the number of tasks goes to infinity. We show that the splitting method converges to the optimal prior as expected, whereas the non-splitting method does not in general without structural assumptions on the data. In contrast, if the data are generated from linear models (the realizable regime), we show that both the splitting and non-splitting methods converge to the optimal prior. Further, perhaps surprisingly, our main result shows that the non-splitting method achieves a *strictly better* asymptotic excess risk under this data distribution, even when the regularization parameter and split ratio are optimally tuned for both methods. Our results highlight that data splitting may not always be preferable, especially when the data is realizable by the model. We validate our theories by experimentally showing that the non-splitting method can indeed outperform the splitting method, on both simulations and real meta-learning tasks.

## 1 Introduction

Meta-learning, also known as "learning to learn", has recently emerged as a powerful paradigm for learning to adapt to unseen tasks (Schmidhuber, 1987). The high-level methodology in meta-learning is akin to how human beings learn new skills, which is typically done by relating to certain prior experience that makes the learning process easier. More concretely, meta-learning does not train one model for each individual task, but rather learns a "prior" model from multiple existing tasks so that it is able to quickly adapt to unseen new tasks. Meta-learning has been successfully applied to many real problems, including few-shot image classification (Finn et al., 2017; Snell et al., 2017), hyper-parameter optimization (Franceschi et al., 2018), low-resource machine translation (Gu et al., 2018) and short event sequence modeling (Xie et al., 2019).

A common practice in meta-learning algorithms is to perform a *sample splitting*, where the data within each task is divided into a *training split* which the prior uses to adapt to a task-specific predictor, and a *validation split* on which we evaluate the performance of the task-specific predictor (Nichol et al., 2018; Rajeswaran et al., 2019; Fallah et al., 2020; Wang et al., 2020a). For example, in a 5-way $k$-shot image classification task, standard meta-learning algorithms such as MAML (Finn et al., 2017) use $5k$ examples within each task as training data, and use additional examples (e.g. $k$ images, one for each class) as validation data. This sample splitting is believed to be crucial as it matches the evaluation criterion at meta-test time, where we perform adaptation on training data from a new task but evaluate its performance on unseen data from the same task.

Despite the aforementioned importance, performing the train-validation split has a potential drawback from the data efficiency perspective — Because of the split, neither the training nor the evaluation stage is able to use all the available per-task data. In the few-shot image classification example,

each task has a total of $6k$ examples available, but the train-validation split forces us to use these data separately in the two stages. Meanwhile, performing the train-validation split is also not the only option in practice: there exist algorithms such as Reptile (Nichol & Schulman, 2018) and Meta-MinibatchProx (Zhou et al., 2019) that can instead use all the per-task data for training the task-specific predictor and also perform well empirically on benchmark tasks. These algorithms modify the loss function in the outer loop so that the training loss no longer matches the meta-test loss, but may have the advantage in terms of data efficiency for the overall problem of learning the best prior. So far it is theoretically unclear how these two approaches (with/without train-validation split) compare with each other, which motivates us to ask the following

**Question**: Is the train-validation split *necessary* and *optimal* in meta-learning?

In this paper, we perform a detailed theoretical study on the importance of the train-validation split. We consider the linear centroid meta-learning problem (Denevi et al., 2018b), where for each task we learn a linear predictor that is close to a common centroid in the inner loop, and find the best centroid in the outer loop (see Section 2 for the detailed problem setup). This problem captures the essence of meta-learning with non-linear models (such as neural networks) in practice, yet is sufficiently simple that allows a precise theoretical characterization. We use a biased ridge solver as the inner loop with a (tunable) regularization parameter, and compare two outer-loop algorithms of either performing the train-validation split (the *train-val method*) or using all the per-task data for both training and evaluation (the *train-train method*). Specifically, we compare the two methods when the number of tasks $T$ is large, and examine if and how fast they converge to the (properly defined) best centroid at meta-test time. We summarize our contributions as follows:

• On the linear centroid meta-learning problem, we show that the train-validation split is necessary in the general agnostic setting: As $T \to \infty$, the train-val method converges to the optimal centroid for test-time adaptation, whereas the train-train method does not without further assumptions on the tasks (Section 3). The convergence of the train-val method is expected since its (population) training loss is equivalent to the meta-test time loss, whereas the non-convergence of the train-train method is because these two losses are not equivalent in general.

• Our main theoretical contribution is to show that the train-validation split is not necessary and even non-optimal, in the perhaps more interesting regime when there are structural assumptions on the tasks: When the data are generated from noiseless linear models, both the train-val and train-train methods converge to the common best centroid, and the train-train method achieves a strictly better (asymptotic) estimation error and test loss than the train-val method (Section 4). This is in stark contrast with the agnostic case, and suggests that data efficiency may indeed be more important when the tasks have a nice structure. Our results build on tools from random matrix theory in the proportional regime, which may be of broader technical interest.

• We perform meta-learning experiments on simulations and benchmark few-shot image classification tasks, showing that the train-train method consistently outperforms the train-val method (Section 5 & Appendix D). This validates our theories and presents empirical evidence that sample-splitting may not be crucial; methods that utilize the per-task data more efficiently may be preferred.

## 1.1 RELATED WORK

**Meta-learning and representation learning theory** Baxter (2000) provided the first theoretical analysis of meta-learning via covering numbers, and Maurer et al. (2016) improved the analysis via Gaussian complexity techniques. Another recent line of theoretical work analyzed gradient-based meta-learning methods (Denevi et al., 2018a; Finn et al., 2019; Khodak et al., 2019; Ji et al., 2020) and showed guarantees for convex losses by using tools from online convex optimization. Saunshi et al. (2020) proved the success of Reptile in a one-dimensional subspace setting. Wang et al. (2020b) compared the performance of train-train and train-val methods for learning the learning rate. Denevi et al. (2018b) proposed the linear centroid model studied in this paper, and provided generalization error bounds for train-val method; the bounds proved also hold for train-train method, so are not sharp enough to compare the two algorithms. Wang et al. (2020a) studied the convergence of gradient-based meta-learning by relating to the kernelized approximation. On the representation learning end, Du et al. (2020); Tripuraneni et al. (2020a;b) showed that ERM can successfully pool data across tasks to learn the representation. Yet the focus is on the accurate estimation of the common representation, not on the fast adaptation of the learned prior. Lastly, we remark that there

are analyses for other representation learning schemes (McNamara & Balcan, 2017; Galanti et al., 2016; Alquier et al., 2016).

**Empirical understandings of meta-learning** Raghu et al. (2020) investigated the representation learning perspective of meta-learning and showed that MAML with a full finetuning inner loop mostly learns the top-layer linear classifier and does not change the representation layers much. This result partly justifies the validity our linear centroid meta-learning problem in which the features (representations) are fixed and only a linear classifier is learned. Goldblum et al. (2020) investigated the difference of the neural representations learned by classical training (supervised learning) and meta-learning, and showed that the meta-learned representation is both better for downstream adaptation and makes classes more separated than the classically trained one. Although the classical training method in (Goldblum et al., 2020) does not perform a train-validation split, it is not exactly the same as the train-train method considered in this work as it effectively performs a supervised learning on all tasks combined and does not do a per-task adaptation.

**Multi-task learning** Multi-task learning also exploits structures and similarities across multiple tasks. The earliest idea dates back to Caruana (1997); Thrun & Pratt (1998); Baxter (2000), initially in connections to neural network models. They further motivated other approaches using kernel methods (Evgeniou et al., 2005; Argyriou et al., 2007) and multivariate linear regression models with structured sparsity (Liu et al., 2009; 2015). More recent advances on deep multi-task learning focus on learning shared intermediate representations across tasks Ruder (2017). These multi-task learning approaches usually minimize the joint empirical risk over all tasks, and the models for different tasks are enforced to share a large amount of parameters. In contrast, meta-learning only requires the models to share the same "prior", which is more flexible than multi-task learning.

## 2  PRELIMINARIES

In this paper, we consider the standard meta-learning setting, in which we observe data from $T \geq 1$ supervised learning tasks, and the goal is to find a prior (or "initialization") using the combined data, such that the $(T + 1)$-th new task may be solved sample-efficiently using the prior.

**Linear centroid meta-learning** We instantiate our study on the *linear centroid meta-learning problem* (also known as learning to learn around a common mean, Denevi et al. (2018b)), where we wish to learn a task-specific linear predictor $\mathbf{w}_t \in \mathbb{R}^d$ in the inner loop for each task $t$, and learn a "centroid" $\mathbf{w}_0$ in the outer loop that enables fast adaptation to $\mathbf{w}_t$ within each task:

Find the best centroid $\mathbf{w}_0 \in \mathbb{R}^d$ for adapting to a linear predictor $\mathbf{w}_t$ on each task $t$.

Formally, we assume that we observe training data from $T \geq 1$ tasks, where for each task index $t$ we sample a task $p_t$ (a distribution over $\mathbb{R}^d \times \mathbb{R}$) from some distribution of tasks $\Pi$, and observe $n$ examples $(\mathbf{X}_t, \mathbf{y}_t) \in \mathbb{R}^{n \times d} \times \mathbb{R}^n$ that are drawn i.i.d. from $p_t$:

$$p_t \sim \Pi, \quad (\mathbf{X}_t, \mathbf{y}_t) = \{(\mathbf{x}_{t,i}, y_{t,i})\}_{i=1}^n \quad \text{where} \quad (\mathbf{x}_{t,i}, y_{t,i}) \overset{\text{iid}}{\sim} p_t. \tag{1}$$

We do not make further assumptions on $(n, d)$; in particular, we allow the underdetermined setting $n \leq d$, in which there exists (one or many) interpolators $\widetilde{\mathbf{w}}_t$ that perfectly fit the data: $\mathbf{X}_t \widetilde{\mathbf{w}}_t = \mathbf{y}_t$.

**Inner loop: Ridge solver with biased regularization towards the centroid** Our goal in the inner loop is to find a linear predictor $\mathbf{w}_t$ that fits the data in task $t$ while being close to the given "centroid" $\mathbf{w}_0 \in \mathbb{R}^d$. We instantiate this through ridge regression (i.e. linear regression with $L_2$ regularization) where the regularization biases $\mathbf{w}_t$ towards the centroid. Formally, for any $\mathbf{w}_0 \in \mathbb{R}^d$ and any dataset $(\mathbf{X}, \mathbf{y})$, we consider the algorithm

$$\mathcal{A}_\lambda(\mathbf{w}_0; \mathbf{X}, \mathbf{y}) := \arg\min_{\mathbf{w}} \frac{1}{n} \|\mathbf{X}\mathbf{w} - \mathbf{y}\|^2 + \lambda \|\mathbf{w} - \mathbf{w}_0\|^2 = \mathbf{w}_0 + (\mathbf{X}^\top \mathbf{X} + n\lambda \mathbf{I}_d)^{-1} \mathbf{X}^\top (\mathbf{y} - \mathbf{X}\mathbf{w}_0),$$

where $\lambda > 0$ is the regularization strength (typically a tunable hyper-parameter). As we regularize by $\|\mathbf{w} - \mathbf{w}_0\|^2$, this inner solver encourages the solution to be close to $\mathbf{w}_0$, as we desire. Such a regularizer is widely used in practical meta-learning algorithms such as MetaOptNet (Lee et al.,

2019) and Meta-MinibatchProx (Zhou et al., 2019). In addition, as $\lambda \to 0$, this solver recovers gradient descent fine-tuning: we have

$$\mathcal{A}_0(\mathbf{w}_0; \mathbf{X}, \mathbf{y}) := \lim_{\lambda \to 0} \mathcal{A}_\lambda(\mathbf{w}_0; \mathbf{X}, \mathbf{y}) = \mathbf{w}_0 + \mathbf{X}^\dagger(\mathbf{y} - \mathbf{X}\mathbf{w}_0) = \arg\min_{\mathbf{X}\mathbf{w}=\mathbf{y}} \|\mathbf{w} - \mathbf{w}_0\|^2$$

(where $\mathbf{X}^\dagger \in \mathbb{R}^{d \times n}$ denotes the pseudo-inverse of $\mathbf{X}$). This is the *minimum-distance* interpolator of $(\mathbf{X}, \mathbf{y})$ and also the solution found by gradient descent [1] on $\|\mathbf{X}\mathbf{w} - \mathbf{y}\|^2$ initialized at $\mathbf{w}_0$. Therefore our ridge solver with $\lambda > 0$ can be seen as a generalized version of the gradient descent solver used in MAML (Finn et al., 2017).

**Outer loop: Finding the best centroid**    In the outer loop, our goal is to find the best centroid $\mathbf{w}_0$. The standard approach in meta-learning is to perform a *train-validation split*, that is, (1) execute the inner solver on a first split of the task-specific data, and (2) evaluate the loss on a second split, yielding a function of $\mathbf{w}_0$ that we can optimize. This two-stage procedure can be written as

$$\text{Compute } \mathbf{w}_t(\mathbf{w}_0) = \mathcal{A}_\lambda(\mathbf{w}_0; \mathbf{X}_t^{\mathsf{train}}, \mathbf{y}_t^{\mathsf{train}}), \quad \text{and} \quad \text{evaluate } \left\|\mathbf{y}_t^{\mathsf{val}} - \mathbf{X}_t^{\mathsf{val}}\mathbf{w}_t(\mathbf{w}_0)\right\|^2.$$

where $(\mathbf{X}_t^{\mathsf{train}}, \mathbf{y}_t^{\mathsf{train}}) = \{(\mathbf{x}_{t,i}, y_{t,i})\}_{i=1}^{n_1}$ and $(\mathbf{X}_t^{\mathsf{val}}, \mathbf{y}_t^{\mathsf{val}}) = \{(\mathbf{x}_{t,i}, y_{t,i})\}_{i=n_1+1}^{n}$ are two disjoint splits of the per-task data $(\mathbf{X}_t, \mathbf{y}_t)$ of size $(n_1, n_2)$, with $n_1 + n_2 = n$. Written concisely, this is to consider the "split loss"

$$\ell_t^{\mathsf{tr\text{-}val}}(\mathbf{w}_0) := \frac{1}{2n_2}\left\|\mathbf{y}_t^{\mathsf{val}} - \mathbf{X}_t^{\mathsf{val}}\mathcal{A}_\lambda(\mathbf{w}_0; \mathbf{X}_t^{\mathsf{train}}, \mathbf{y}_t^{\mathsf{train}})\right\|^2. \tag{2}$$

In this paper, we will also consider an alternative version, where we do not perform the train-validation split, but instead use *all the per-task data for both training and evaluation*. Mathematically, this is to look at the "non-split loss"

$$\ell_t^{\mathsf{tr\text{-}tr}}(\mathbf{w}_0) := \frac{1}{2n}\left\|\mathbf{y}_t - \mathbf{X}_t\mathcal{A}_\lambda(\mathbf{w}_0; \mathbf{X}_t, \mathbf{y}_t)\right\|^2. \tag{3}$$

Our overall algorithm is to solve the empirical risk minimization (ERM) problem on the $T$ observed tasks, using either one of the two losses above:

$$\widehat{L}_T^{\mathsf{tr\text{-}val}}(\mathbf{w}_0) := \frac{1}{T}\sum_{t=1}^{T}\ell_t^{\mathsf{tr\text{-}val}}(\mathbf{w}_0) \quad \text{and} \quad \widehat{L}_T^{\mathsf{tr\text{-}tr}}(\mathbf{w}_0) := \frac{1}{T}\sum_{t=1}^{T}\ell_t^{\mathsf{tr\text{-}tr}}(\mathbf{w}_0),$$
$$\widehat{\mathbf{w}}_{0,T}^{\{\mathsf{tr\text{-}val},\mathsf{tr\text{-}tr}\}} := \arg\min_{\mathbf{w}_0} \widehat{L}_T^{\{\mathsf{tr\text{-}val},\mathsf{tr\text{-}tr}\}}(\mathbf{w}_0). \tag{4}$$

Let $L^{\{\mathsf{tr\text{-}val},\mathsf{tr\text{-}tr}\}}(\mathbf{w}_0) := \mathbb{E}_{p_t \sim \Pi, (\mathbf{X}_t, \mathbf{y}_t) \sim p_t}\left[\ell_t^{\{\mathsf{tr\text{-}val},\mathsf{tr\text{-}tr}\}}(\mathbf{w}_0)\right]$ be the population risks.

**(Meta-)Test time**    The meta-test time performance of any meta-learning algorithm is a joint function of the (learned) centroid $\mathbf{w}_0$ and the inner algorithm Alg. Upon receiving a new task $p_{T+1} \sim \Pi$ and training data $(\mathbf{X}_{T+1}, \mathbf{y}_{T+1}) \in \mathbb{R}^{n \times d} \times \mathbb{R}^n$, we run the inner loop Alg with prior $\mathbf{w}_0$ on the training data, and evaluate it on an (unseen) test example $(\mathbf{x}', y') \sim p_{T+1}$:

$$L^{\mathsf{test}}(\mathbf{w}_0; \mathsf{Alg}) := \mathbb{E}_{p_{T+1} \sim \Pi}\mathbb{E}_{(\mathbf{X}_{T+1}, \mathbf{y}_{T+1}), (\mathbf{x}', y') \overset{\mathsf{iid}}{\sim} p_{T+1}}\left[\frac{1}{2}\left(\mathbf{x}'^\top \mathsf{Alg}(\mathbf{w}_0; \mathbf{X}_{T+1}, \mathbf{y}_{T+1}) - y'\right)^2\right].$$

Additionally, for both train-val and train-train methods, we need to ensure that the inner loop used for meta-test is exactly the same as that used in meta-training. Therefore, the meta-test performance for the train-val and train-train methods above should be evaluated as

$$L_{\lambda,n_1}^{\mathsf{test}}(\widehat{\mathbf{w}}_{0,T}^{\mathsf{tr\text{-}val}}) := L^{\mathsf{test}}(\widehat{\mathbf{w}}_{0,T}^{\mathsf{tr\text{-}val}}; \mathcal{A}_{\lambda,n_1}), \quad L_{\lambda,n}^{\mathsf{test}}(\widehat{\mathbf{w}}_{0,T}^{\mathsf{tr\text{-}tr}}) := L^{\mathsf{test}}(\widehat{\mathbf{w}}_{0,T}^{\mathsf{tr\text{-}tr}}; \mathcal{A}_{\lambda,n}),$$

where $\mathcal{A}_{\lambda,m}$ denotes the ridge solver with regularization strength $\lambda > 0$ on $m \le n$ data points. Finally, we let

$$\mathbf{w}_{0,\star}(\lambda; n) = \arg\min_{\mathbf{w}_0} L_{\lambda,n}^{\mathsf{test}}(\mathbf{w}_0) \tag{5}$$

denote the best centroid if the inner loop uses $\mathcal{A}_{\lambda,n}$. The performance of the train-val algorithm $\widehat{\mathbf{w}}_{0,T}^{\mathsf{tr\text{-}val}}$ should be compared against $\mathbf{w}_{0,\star}(\lambda, n_1)$, whereas the train-train algorithm $\widehat{\mathbf{w}}_{0,T}^{\mathsf{tr\text{-}tr}}$ should be compared against $\mathbf{w}_{0,\star}(\lambda, n)$.

---

[1] with a small step-size, or gradient flow.

## 2.1 Task-abundant setting through asymptotic analysis

In this paper we are interested in the *task-abundant* setting where we fix some finite $(d, n)$ and let $T$ be very large. We analyze such a task-abundant setting through the asymptotic analysis framework, that is, examine the limiting properties of the estimator (e.g. $\widehat{\mathbf{w}}_{0,T}^{\{\text{tr-val,tr-tr}\}}$) as $T \to \infty$. Here we set up the basic notation of asymptotic analysis required in this paper. We emphasize that our large $T$ setting captures practical meta-learning scenarios; for example, 5-way image classification on miniImageNet (Ravi & Larochelle, 2017) contains $\binom{64}{5}$ diverse tasks (at train time).

**Asymptotic rate of estimation & excess risk**   Let $L$ be any population risk with minimizer $\mathbf{w}_{0,\star}$ (which we assume is unique), $\widehat{L}_T$ be the empirical risk on the observed data from $T$ tasks, and $\widehat{\mathbf{w}}_{0,T}$ be the minimizer of $\widehat{L}_T$ (i.e. the ERM). We say that $\widehat{\mathbf{w}}_{0,T}$ is **consistent** if $\widehat{\mathbf{w}}_{0,T} \to \mathbf{w}_{0,\star}$ in probability as $T \to \infty$. For consistent ERMs, we define its asymptotic parameter estimation error (in MSE loss) and asymptotic excess risk as follows[2]:

$$\text{AsymMSE}(\widehat{\mathbf{w}}_{0,T}) := \lim_{T \to \infty} T \cdot \mathbb{E}\left[\left\|\widehat{\mathbf{w}}_{0,T} - \mathbf{w}_{0,\star}\right\|^2\right]$$

$$\text{AsymExcessRisk}(\widehat{\mathbf{w}}_{0,T}) := \lim_{T \to \infty} T \cdot \mathbb{E}[L(\widehat{\mathbf{w}}_{0,T}) - L(\mathbf{w}_{0,\star})].$$

We emphasize that asymptotic statements are more refined than non-asymptotic $O(\cdot)$ style upper bounds in the $T \to \infty$ limit: they already imply the {MSE, excess risk} has order $O(1/T)$ and specifies the *leading constant*.

## 3 The importance of sample splitting

We begin by analyzing whether the algorithms $\widehat{\mathbf{w}}_{0,T}^{\{\text{tr-val,tr-tr}\}}$ defined in (4) converge to the best test-time centroid $\mathbf{w}_{0,\star}(\lambda; n_1)$ or $\mathbf{w}_{0,\star}(\lambda; n)$ (defined (5)) respectively as $T \to \infty$, in the general situation where we do not make structural assumptions on the data distribution $p_t$.

**Proposition 1** (Consistency and asymptotics of train-val method). *Suppose* $\mathbb{E}_{\mathbf{x} \sim p_t}[\mathbf{x}\mathbf{x}^\top] \succ \mathbf{0}$, $\mathbb{E}_{\mathbf{x} \sim p_t}[\|\mathbf{x}\|^4] < \infty$ *and* $\mathbb{E}_{(\mathbf{x},y) \sim p_t}[\|\mathbf{x}y\|] < \infty$ *for almost surely all* $p_t \sim \Pi$. *Then for any* $\lambda > 0$ *and any* $(n_1, n_2)$ *such that* $n_1 + n_2 = n$, *the train-val method* $\widehat{\mathbf{w}}_{0,T}^{\text{tr-val}}$ *converges to the best test-time centroid:* $\widehat{\mathbf{w}}_{0,T}^{\text{tr-val}} \to \mathbf{w}_{0,\star}(\lambda, n_1)$ *almost surely as* $T \to \infty$. *Further, we have*

$$\text{AsymMSE}(\widehat{\mathbf{w}}_{0,T}^{\text{tr-val}}) = \text{tr}\big(\nabla^{-2} L_{\lambda,n_1}^{\text{test}}(\mathbf{w}_{0,\star}(\lambda, n_1)) \cdot \text{Cov}\big(\nabla \ell_t^{\text{tr-val}}(\mathbf{w}_{0,\star}(\lambda, n_1))\big) \cdot \nabla^{-2} L_{\lambda,n_1}^{\text{test}}(\mathbf{w}_{0,\star}(\lambda, n_1))\big),$$

$$\text{AsymExcessRisk}_{L_{\lambda,n_1}^{\text{test}}}(\widehat{\mathbf{w}}_{0,T}^{\text{tr-val}}) = \text{tr}\big(\nabla^{-2} L_{\lambda,n_1}^{\text{test}}(\mathbf{w}_{0,\star}(\lambda, n_1)) \cdot \text{Cov}\big(\nabla \ell_t^{\text{tr-val}}(\mathbf{w}_{0,\star}(\lambda, n_1))\big)\big).$$

**Proposition 2** (Inconsistency of train-train method). *There exists a distribution of tasks* $\Pi$ *on* $d = 1$ *satisfying the conditions in Proposition 1 on which the train-train method does not converge to the best test-time centroid: for any* $n \geq 1$ *and any* $\lambda > 0$, *the estimation error* $\|\widehat{\mathbf{w}}_{0,T}^{\text{tr-tr}} - \mathbf{w}_{0,\star}(\lambda, n)\|$ *and the excess risk* $L_{\lambda,n}^{\text{test}}(\widehat{\mathbf{w}}_{0,T}^{\text{tr-tr}}) - L_{\lambda,n}^{\text{test}}(\mathbf{w}_{0,\star}(\lambda, n))$ *are both bounded away from 0 almost surely as* $T \to \infty$.

Propositions 1 and 2 justify the importance of sample splitting: the train-val method converges to the best test-time centroid, whereas the train-train method does not converge to the best centroid in general. The reason behind Proposition 1 is simple: the population loss $L^{\text{tr-val}}$ for the train-val method is indeed equal to the test-time loss $L_{\lambda,n_1}^{\text{test}}$, making the train-val method a proper ERM for the meta-test time we care about, and thus the consistency and asymptotic normality follow from classical results for the ERM (e.g. (Van der Vaart, 2000; Liang, 2016)).

In contrast, for the train-train method, its expected loss $L^{\text{tr-tr}}$ is in general *not equivalent* to $L_{\lambda,n}^{\text{test}}$: $L^{\text{tr-tr}}$ measures the in-sample prediction error of the per-task predictor, whereas $L_{\lambda,n}^{\text{test}}$ measures the out-of-sample prediction error. Consequently, the population minimizers of $L_{\lambda,n}^{\text{test}}$ and $L^{\text{tr-tr}}$ are not equal in general, which leads to $\widehat{\mathbf{w}}_{0,\star}$ converging to the minimizer of $L^{\text{tr-tr}}$, not of $L_{\lambda,n}^{\text{test}}$. The proof of Proposition 2 constructs a simple counter-example in $d = 1$, but we expect such a mismatch to generally hold in any dimension. Appendix A gives the proofs of Proposition 1 and 2.

---

[2]These definitions assume that the expectation exists for finite $T$; the more general definition can be found in Appendix A.1.

## 4 IS SAMPLE SPLITTING ALWAYS OPTIMAL?

Proposition 2 states a negative result for the train-train method, showing that it does not converge to the best test-time centroid without further assumptions on the data distribution. However, such a negative result is inherently *worst-case*, and does not preclude the possibility that there exists a data distribution on which the train-train method can also work well. In this section, we construct a simple data distribution in which we can analyze the performance of both the train-val and the train-train methods more explicitly, showing that sample splitting is indeed not optimal, and the train-train method can work better.

**Realizable linear model**   We consider the following instantiation of the (generic) data distribution assumption in (1): We assume that each task $p_t$ is specified by a $\mathbf{w}_t \in \mathbb{R}^d$ sampled from some distribution $\Pi$ (overloading notation), and the observed data follows the noiseless linear model with ground truth parameter $\mathbf{w}_t$:

$$\mathbf{y}_t = \mathbf{X}_t \mathbf{w}_t, \tag{6}$$

where the inputs $\mathbf{x}_{t,i} \stackrel{\text{iid}}{\sim} \mathsf{N}(\mathbf{0}, \mathbf{I}_d)$ and are independent of $\mathbf{w}_t$. We assume that $\Pi$ has a finite second moment (i.e. $\mathbb{E}_{\mathbf{w}_t \sim \Pi}[\|\mathbf{w}_t\|^2] < \infty$). Note that when $n \geq d$, we are able to perfectly recover $\mathbf{w}_t$ for all $t$ (by solving linear equations), therefore the problem in the inner loop is in a sense "easy"; when $n < d$, we cannot hope for such perfect recoveries. Our goal in the outer loop is to find the best $\mathbf{w}_0$, measured by the test loss $L_{\lambda,n}^{\text{test}}$ for the train-train method and $L_{\lambda,n_1}^{\text{test}}$ for the train-val method.

### 4.1 COMPARISON OF TRAIN-TRAIN AND TRAIN-VAL ON THE REALIZABLE MODEL

We begin by showing that on this task and data distribution, the population best centroids $\mathbf{w}_{0,\star}(\lambda, n) = \arg\min_{\mathbf{w}_0} L_{\lambda,n}^{\text{test}}(\mathbf{w}_0)$ is the same for any $(\lambda, n)$, and both the train-val and train-train methods are asymptotically consistent and converge to same best centroid.

**Theorem 3** (Consistency of both train-val and train-train methods). *On the realizable linear model* (6), *the test-time meta loss for all $\lambda > 0$ and all $n$ is minimized at the same point, that is, the mean of the ground truth parameters:*

$$\mathbf{w}_{0,\star}(\lambda, n) = \arg\min_{\mathbf{w}_0} L_{\lambda,n}^{\text{test}}(\mathbf{w}_0) = \mathbf{w}_{0,\star} := \mathbb{E}_{\mathbf{w}_t \sim \Pi}[\mathbf{w}_t], \quad \text{for all } \lambda > 0, \ n.$$

*Further, both the train-val method and the train-train method are asymptotically consistent: for any $\lambda > 0$, $n$, and $(n_1, n_2)$, we have*

$$\widehat{\mathbf{w}}_{0,T}^{\text{tr-val}}(n_1, n_2; \lambda) \to \mathbf{w}_{0,\star} \quad \text{and} \quad \widehat{\mathbf{w}}_{0,T}^{\text{tr-tr}}(n; \lambda) \to \mathbf{w}_{0,\star} \quad \text{almost surely as} \ \ T \to \infty.$$

See its proof in Appendix B.1. Theorem 3 shows that both train-val and train-train methods are consistent, and they converge to the same optimal parameter $\mathbf{w}_{0,\star}$ which is the mean of $\mathbf{w}_t$. This is a consequence of the good structure in our realizable linear model (6): at a high level, $\mathbf{w}_{0,\star}$ is indeed the best centroid since it has (on average) the closest distance to a randomly sampled $\mathbf{w}_t$.

Theoerem 3 suggests that we are now able to compare performance of the two methods based on their asymptotic parameter estimation error (for estimating $\mathbf{w}_{0,\star}$). Throughout the rest of this section, let

$$R^2 := \mathbb{E}\Big[\|\mathbf{w}_t - \mathbf{w}_{0,\star}\|^2\Big] \tag{7}$$

denote the variance of $\mathbf{w}_t$. We are now ready to state our main result.

**Theorem 4** (Comparison of asymptotic MSE of the train-val and train-train methods). *In the high-dimensional limiting regime $d, n \to \infty$, $d/n \to \gamma \in (0, \infty)$, the optimal rate of the train-train method obtained by tuning the regularization $\lambda \in (0, \infty)$ satisfies*

$$\inf_{\lambda > 0} \lim_{d,n \to \infty, d/n = \gamma} \text{AsymMSE}\big(\widehat{\mathbf{w}}_{0,T}^{\text{tr-tr}}(n; \lambda)\big) = \inf_{\lambda > 0} \rho_{\lambda,\gamma} R^2 \stackrel{(\star)}{\leq} \max\left\{1 + \frac{5}{27}\gamma, \frac{5}{27} + \gamma\right\} \cdot R^2,$$

*where $\rho_{\lambda,\gamma} = 4\gamma^2 \big[(\gamma-1)^2 + (\gamma+1)\lambda\big]/(\lambda + \gamma + 1 - \sqrt{(\lambda+\gamma+1)^2 - 4\gamma})^2/\big((\lambda+\gamma+1)^2 - 4\gamma\big)^{3/2}$, and the inequality becomes equality at $\gamma = 1$. In contrast, the optimal rate of the train-val method by tuning the regularization $\lambda \in (0, \infty)$ and split ratio $s \in (0, 1)$ is*

$$\inf_{\lambda > 0, s \in (0,1)} \lim_{d,n \to \infty, d/n = \gamma} \text{AsymMSE}\big(\widehat{\mathbf{w}}_{0,T}^{\text{tr-val}}(ns, n(1-s); \lambda)\big) = (1 + \gamma)R^2.$$

*As $\max\{1 + 5\gamma/27, 5/27 + \gamma\} < 1 + \gamma \ (\forall \gamma > 0)$, the train-train method achieves a strictly better asymptotic rate than the train-val method when $\lambda$ and $s$ are optimally tuned in both methods.*

**Implications**    Comparison between the analytical upper bound $\max\{1 + 5\gamma/27, 5/27 + \gamma\}R^2$ for train-train $(1 + \gamma)R^2$ for train-val in Theorem 4 shows that the train-train method achieves a strictly better asymptotic MSE than the train-val method, for any $\gamma > 0$[3]. (See Figure 1(a) for a visualization of the exact optimal rates and the upper bound $(\star)$.) Perhaps surprisingly, this suggests that the train-train method is not only "correct" (converging to the best centroid), but can be even better than the train-val method, when the data is structured. While the "correctness" of the train-train method is a consequence of the realizable linear model, we believe its superior asymptotic MSE is due to the fact that the train-train method is able to use the data more efficiently than the train-val method.

Also, while we reached such a conclusion on this particular problem of linear centroid meta-learning, we suspect that this phenomenon to be not restricted to this problem, and may hold in more generality when data is structured or when the signal-to-noise ratio is high. As we are going to see, our real data experiments in Section 5.2 indeed suggests that the superiority of the train-train method may also hold on real meta-learning tasks with neural networks.

### 4.2    PROOF HIGHLIGHTS OF THEOREM 4

Here we sketch the technical highlights in proving Theorem 4. We defer the full proof to Appendix B.5.

**Exact asymptotic MSE for both methods**    The proof begins by calculating the exact asymptotic MSE for both methods, which we provide in the following theorem.

**Lemma 5** (Exact asymptotic rates of the train-val and train-train methods). *Suppose that* $\rho_{\text{tr-tr}} =$
$$\frac{\mathbb{E}\left[\sum_{i=1}^{d}(\sigma_i^{(n)})^2/(\sigma_i^{(n)}+\lambda)^4\right]}{\left(\mathbb{E}\left[\sum_{i=1}^{d}\sigma_i^{(n)}/(\sigma_i^{(n)}+\lambda)^2\right]\right)^2} \quad \text{and} \quad \rho_{\text{tr-val}} = \frac{\mathbb{E}\left[\left(\sum_{i=1}^{d}\lambda^2/(\sigma_i^{(n_1)}+\lambda)^2\right)^2 + (n_2+1)\sum_{i=1}^{d}\lambda^4/(\sigma_i^{(n_1)}+\lambda)^4\right]}{\left(\mathbb{E}\left[\sum_{i=1}^{d}\lambda^2/(\sigma_i^{(n_1)}+\lambda)^2\right]\right)^2},$$
*where for any* $n$, $\sigma_1^{(n)} \geq \cdots \geq \sigma_d^{(n)}$ *denotes the eigenvalues of the matrix* $\frac{1}{n}\mathbf{X}_t^\top\mathbf{X}_t \in \mathbb{R}^{d\times d}$, *where we recall* $\mathbf{X}_t \in \mathbb{R}^{n\times d}$ *is a random matrix with i.i.d. standard Gaussian entries. For any* $(n,d)$, *we have on the realizable linear model* (6) *that*

$$\text{AsymMSE}\big(\widehat{\mathbf{w}}_{0,T}^{\text{tr-tr}}(n;\lambda)\big) = dR^2\rho_{\text{tr-tr}}, \quad \text{AsymMSE}\big(\widehat{\mathbf{w}}_{0,T}^{\text{tr-val}}(n_1,n_2;\lambda)\big) = \frac{dR^2\rho_{\text{tr-val}}}{n_2}.$$

See its proof in Appendix B.2. Lemma 5 follows straightforwardly from the classical asymptotic result for empirical risk minimization (Van der Vaart, 2000) and simplifications of certain matrix traces in terms of the spectrum of the empirical covariance matrix $\frac{1}{n}\sum_{t=1}^{n}\mathbf{X}_t\mathbf{X}_t^\top$.

**Simplifying and optimizing the asymptotic MSEs**    The asymptotic MSEs of the train-train and train-val method in Lemma 5 are not yet directly comparable, as the quantities $\rho_{\text{tr-tr}}$ and $\rho_{\text{tr-val}}$ depend on the spectrum of the empirical covariance matrix as well as additional tunable parameters such as $\lambda$ and $(n_1, n_2)$ (for the train-val method). Towards proving Theorem 4, we further simplify the rates and analyze the optimal tunable parameters, using separate strategies for the two methods:

- For the train-val method, we show that the optimal tunable parameters for any $(n, d)$ is taken at a special case $\lambda = \infty$ and $(n_1, n_2) = (0, n)$, at which the rates only depends on $\frac{1}{n_1}\mathbf{X}_t^{\text{train}\top}\mathbf{X}_t^{\text{train}}$ through its rank (and thus has a simple closed-form). We state this result in Corollary 8. The proof builds on algebraic manipulations of the quantity $\rho_{\text{tr-val}}$, and can be found in Appendix B.3.

- For the train-train method, we apply random matrix theory to simplify the spectrum of $\frac{1}{n}\mathbf{X}_t^\top\mathbf{X}_t$ in the *proportional limit* where $d, n \to \infty$ and $d/n$ stays a constant (Bai & Silverstein, 2010; Anderson et al., 2010), and obtain a closed-form expression of the asymptotic MSE for any $\lambda > 0$, which we can analytically optimize over $\lambda$. We state this result in Theorem 9. The proof builds on the Steiltjes transform and its "derivative trick" (Dobriban et al., 2018), and is deferred to Appendix B.4.

---

[3]The same conclusion also holds for the asymptotic excess risk, as the Hessian of the excess risk is a rescaled identity, see Appendix B.2.

# 5 EXPERIMENTS

## 5.1 SIMULATIONS

We experiment on the realizable linear model studied in Section 4. Recall that the observed data of the $t$-th task are generated as

$$\mathbf{y}_t = \mathbf{X}_t \mathbf{w}_t, \quad \text{with} \quad \mathbf{x}_{t,i} \overset{\text{iid}}{\sim} \mathsf{N}(0, \mathbf{I}_d).$$

We independently generate $\mathbf{w}_t \overset{\text{iid}}{\sim} \mathsf{N}(\mathbf{w}_{0,\star}, \mathbf{I}_d/\sqrt{d})$, where $\mathbf{w}_{0,\star}$ is the linear centroid and the corresponding $R^2 = 1$ here. The goal is to learn the linear centroid $\mathbf{w}_{0,\star}$ using the train-train method and train-val method, i.e., minimizing $\widehat{L}_T^{\text{tr-tr}}$ and $\widehat{L}_T^{\text{tr-val}}$, respectively. Note that both $\widehat{L}_T^{\text{tr-tr}}$ and $\widehat{L}_T^{\text{tr-val}}$ are quadratic in $\mathbf{w}_0$, therefore, we can find the close-form solutions $\widehat{\mathbf{w}}_{0,T}^{\{\text{tr-tr,tr-val}\}}$. We measure the performance of train-train and train-val methods using the $\ell_2$-error $\|\mathbf{w}_{0,\star} - \widehat{\mathbf{w}}_{0,T}^{\{\text{tr-tr,tr-val}\}}\|^2$.

We present the comparison among train-train and train-val methods in Figure 1 with scatter plots representing the simulation outputs under different settings. Across all the simulations, we well-tune the regularization coefficient $\lambda$ in the train-train method, and use a sufficiently large $\lambda = 10000$ in the train-val method according to Corollary 8. The simulated results concentrate around the reference curves corresponding to our theoretical findings. This corroborates our analyses and demonstrates the better performance of train-train method on the realizable linear model.

We additionally investigate the effect of averaging the loss over multiple splits in the train-val method (a "cross-validation" type loss) rather than the vanilla single split. We show that such cross-validation can indeed improve over the vanilla single-split train-val method. We also experiment with the stronger "leave-one-out" style cross-validation and show that it achieves better MSEs than the constant-fold cross-validation (Appendix E).

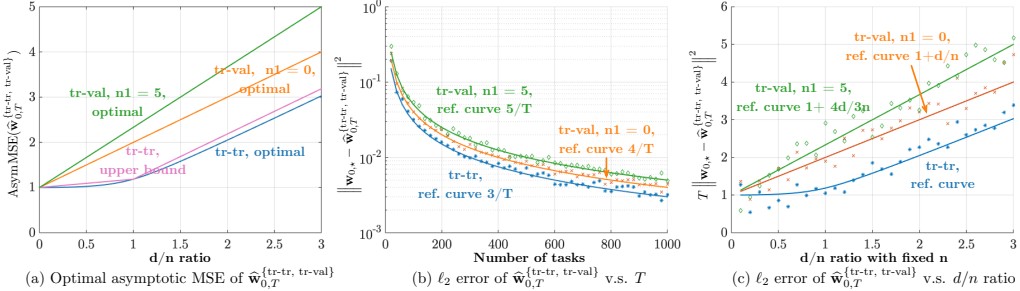

(a) Optimal asymptotic MSE of $\widehat{\mathbf{w}}_{0,T}^{\{\text{tr-tr, tr-val}\}}$    (b) $\ell_2$ error of $\widehat{\mathbf{w}}_{0,T}^{\{\text{tr-tr, tr-val}\}}$ v.s. $T$    (c) $\ell_2$ error of $\widehat{\mathbf{w}}_{0,T}^{\{\text{tr-tr, tr-val}\}}$ v.s. $d/n$ ratio

Figure 1: Panel (a) presents the optimal AsymMSE($\widehat{\mathbf{w}}_{0,T}^{\text{tr-tr}}$) (blue) in Theorem 9 via grid search, and the optimal AsymMSE($\widehat{\mathbf{w}}_{0,T}^{\text{tr-val}}$) in Corollary 8 with $n_1 = 0$ (orange) and $n_1 = 5$ (green), as well as the upper bound of AsymMSE($\widehat{\mathbf{w}}_{0,T}^{\text{tr-tr}}$) (magenta) in Corollary 4. The optimal AsymMSE($\widehat{\mathbf{w}}_{0,T}^{\{\text{tr-tr,tr-val}\}}$) are used as reference curves in plots (b) and (c). Panel (b) plots the $\ell_2$-error of $\widehat{\mathbf{w}}_{0,T}^{\{\text{tr-tr,tr-val}\}}$ as the total number of tasks increases from 20 to 1000 with an increment of 20. We fix data dimension $d = 60$ and per-task sample size $n = 20$. For the train-val method, we experiment on $n_1 = 0$ and $n_1 = 5$. Panel (c) shows the scaled (by $T$) $\ell_2$-error of $\widehat{\mathbf{w}}_{0,T}^{\{\text{tr-tr,tr-val}\}}$ as the ratio $d/n$ varies from 0 to 3 ($n = 20$ and $T = 1000$ are fixed).

## 5.2 FEW-SHOT IMAGE CLASSIFICATION

We further investigate the comparison between the train-train and train-val type methods in few-shot image classification on miniImageNet (Ravi & Larochelle, 2017) and tieredImageNet (Ren et al., 2018).

**Methods** We instantiate the train-train and train-val method in the centroid meta-learning setting with a ridge solver. The methods are almost exactly the same as in our theoretical setting in (2) and (3), with the only differences being that the parameters $\mathbf{w}_t$ (and hence $\mathbf{w}_0$) parametrize a deep

neural network instead of a linear classifier, and the loss function is the cross-entropy instead of squared loss. Mathematically, we minimize the following two loss functions:

$$L_{\lambda,n_1}^{\text{tr-val}}(\mathbf{w}_0) := \frac{1}{T}\sum_{t=1}^{T}\ell_t^{\text{tr-val}}(\mathbf{w}_0) = \frac{1}{T}\sum_{t=1}^{T}\ell\left(\arg\min_{\mathbf{w}_t}\ell(\mathbf{w}_t; \mathbf{X}_t^{\text{train}}, \mathbf{y}_t^{\text{train}}) + \lambda\|\mathbf{w}_t - \mathbf{w}_0\|^2; \mathbf{X}_t^{\text{val}}, \mathbf{y}_t^{\text{val}}\right),$$

$$L_{\lambda}^{\text{tr-tr}}(\mathbf{w}_0) := \frac{1}{T}\sum_{t=1}^{T}\ell_t^{\text{tr-tr}}(\mathbf{w}_0) = \frac{1}{T}\sum_{t=1}^{T}\ell\left(\arg\min_{\mathbf{w}_t}\ell(\mathbf{w}_t; \mathbf{X}_t, \mathbf{y}_t) + \lambda\|\mathbf{w}_t - \mathbf{w}_0\|^2; \mathbf{X}_t, \mathbf{y}_t\right),$$

where $(\mathbf{X}_t, \mathbf{y}_t)$ is the data for task $t$ of size $n$, and $(\mathbf{X}_t^{\text{train}}, \mathbf{y}_t^{\text{train}})$ and $(\mathbf{X}_t^{\text{val}}, \mathbf{y}_t^{\text{val}})$ is a split of the data of size $(n_1, n_2)$. We note that both loss functions above have been considered in prior work ($L^{\text{tr-val}}$ in iMAML (Rajeswaran et al., 2019), and $L^{\text{tr-tr}}$ in Meta-MinibatchProx (Zhou et al., 2019)), though we use slightly different implementation details from these prior work to make sure that the two methods here are exactly the same except for whether the split is used. Additional details about the implementation can be found in Appendix D.

**Experimental settings**   We adopt the episodic training procedure (Finn et al., 2017; Zhou et al., 2019; Rajeswaran et al., 2019). In meta-test, we sample a set of $N$-way $(K+1)$-shot test tasks. The first $K$ instances are for training and the remaining one is for testing. In meta-training, we use the "higher way" training strategy. We set the default choice of the train-validation split ratio to be an even split $n_1 = n_2 = n/2$ following Zhou et al. (2019); Rajeswaran et al. (2019). For example, for a 5-way 5-shot classification setting, each task contains $5 \times (5+1) = 30$ total images, and we set $n_1 = n_2 = 15$. (We additionally investigate the optimality of this split ratio in Appendix D.1.) We evaluate both methods under the transduction setting where the information is shared between the test data via batch normalization. We report the average accuracy over $2,000$ random test episodes with 95% confidence interval.

**Results**   Table 1 presents the percent classification accuracy on miniImagenet and tieredImageNet. We find that the train-train method consistently outperforms the train-val method. Specifically, on miniImageNet, train-train method outperforms train-val by 2.01% and 3.87% on the 1-shot 5-way and 5-shot 5-way tasks respectively; On tieredImageNet, train-train on average improves by about 6.40% on the four testing cases. These results show the advantages of train-train method over train-val and support our theoretical findings in Theorem 4.

Table 1: Few-shot classification accuracy (%) on the miniImageNet and tieredImageNet datasets.

| | method | 1-shot 5-way | 5-shot 5-way | 1-shot 20-way | 5-shot 20-way |
|---|---|---|---|---|---|
| miniImage | train-val | $48.76 \pm 0.87$ | $63.56 \pm 0.95$ | $17.52 \pm 0.49$ | $21.32 \pm 0.54$ |
| | train-train | $\mathbf{50.77 \pm 0.90}$ | $\mathbf{67.43 \pm 0.89}$ | $\mathbf{21.17 \pm 0.38}$ | $\mathbf{34.30 \pm 0.41}$ |
| | method | 1-shot 5-way | 5-shot 5-way | 1-shot 10-way | 5-shot 10-way |
| tieredImage | train-val | $50.61 \pm 1.12$ | $67.30 \pm 0.98$ | $29.18 \pm 0.57$ | $43.15 \pm 0.72$ |
| | train-train | $\mathbf{54.37 \pm 0.93}$ | $\mathbf{71.45 \pm 0.94}$ | $\mathbf{35.56 \pm 0.60}$ | $\mathbf{54.50 \pm 0.71}$ |

## 6   CONCLUSION

We study the importance of train-validation split on the linear-centroid meta-learning problem, and show that the necessity and optimality of train-validation split depends greatly on whether the tasks are structured: the sample splitting is necessary in general situations, and not necessary and non-optimal when the tasks are nicely structured. It would be of interest to study whether a similar conclusion holds on other meta-learning problems such as learning a representation, or whether our insights can be used towards designing meta-learning algorithms with better empirical performance.

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

# A   PROOFS FOR SECTION 3

## A.1   DETAILED INTRODUCTION ON ASYMPTOTIC ANALYSIS

Here we provide more details on the asymptotic analysis framework sketched in Section 2.1.

In typical scenarios, for consistent ERMs, the limiting distribution of $\widehat{\mathbf{w}}_{0,T}$ is asymptotically normal with a known covariance matrix, as is characterized in the following classical result (see, e.g. Van der Vaart (2000, Theorem 5.21) and also Liang (2016)).

**Proposition 6** (Asymptotic normality and excess risk of ERMs). *Assume the population minimizer $\mathbf{w}_{0,\star}$ is unique and the ERM $\widehat{\mathbf{w}}_{0,T}$ is consistent (i.e. it converges to $\mathbf{w}_{0,\star}$ in probability as $T \to \infty$). Further assume the following regularity conditions:*

*(a) There exists some random variable $A_t = A(p_t, \mathbf{X}_t, \mathbf{y}_t)$ such that $\mathbb{E}[A_t^2] < \infty$ and*

$$\|\nabla \ell_t(\mathbf{w}_1) - \nabla \ell_t(\mathbf{w}_2)\| \le A_t \|\mathbf{w}_1 - \mathbf{w}_2\|$$

*for all $\mathbf{w}_1, \mathbf{w}_2 \in \mathbb{R}^d$;*

*(b) $\mathbb{E}[\|\nabla \ell_t(\mathbf{w}_{0,\star})\|^2] < \infty$;*

*(c) $L$ is twice-differentiable with $\nabla^2 L(\mathbf{w}_{0,\star}) \succ \mathbf{0}$,*

*then the ERM $\widehat{\mathbf{w}}_{0,T}$ is asymptotically normally distributed, with*

$$\sqrt{T} \cdot (\widehat{\mathbf{w}}_{0,T} - \mathbf{w}_{0,\star}) \xrightarrow{d} \mathsf{N}\big(\mathbf{0}, \nabla^2 L(\mathbf{w}_{0,\star})^{-1} \mathrm{Cov}(\nabla \ell_t(\mathbf{w}_{0,\star})) \nabla^2 L(\mathbf{w}_{0,\star})^{-1}\big) =: P_{\mathbf{w}},$$

$$T \cdot (L(\widehat{\mathbf{w}}_{0,T}) - L(\mathbf{w}_{0,\star})) \xrightarrow{d} \boldsymbol{\Delta}^\top \nabla^2 L(\mathbf{w}_{0,\star}) \boldsymbol{\Delta} \quad \text{where } \boldsymbol{\Delta} \sim P_{\mathbf{w}}.$$

*where $\xrightarrow{d}$ denotes convergence in distribution and $\ell_t : \mathbb{R}^d \to \mathbb{R}$ is the loss function on a single task.*

When this happens, we define the asymptotic rate of estimation (in MSE loss) and asymptotic excess risk $\widehat{\mathbf{w}}_{0,T}$ as those of its limiting distribution:

$$\mathrm{AsymMSE}(\widehat{\mathbf{w}}_{0,T}) := \mathbb{E}_{\boldsymbol{\Delta} \sim P_{\mathbf{w}}}\left[\|\boldsymbol{\Delta}\|^2\right] = \mathrm{tr}\big(\nabla^2 L(\mathbf{w}_{0,\star})^{-1} \mathrm{Cov}(\nabla \ell_t(\mathbf{w}_{0,\star})) \nabla^2 L(\mathbf{w}_{0,\star})^{-1}\big)$$

$$\mathrm{AsymExcessRisk}(\widehat{\mathbf{w}}_{0,T}) := \mathbb{E}_{\boldsymbol{\Delta} \sim P_{\mathbf{w}}}\big[\boldsymbol{\Delta}^\top \nabla^2 L(\mathbf{w}_{0,\star}) \boldsymbol{\Delta}\big] = \mathrm{tr}\big(\nabla^2 L(\mathbf{w}_{0,\star})^{-1} \mathrm{Cov}(\nabla \ell_t(\mathbf{w}_{0,\star}))\big).$$

## A.2   PROOF OF PROPOSITION 1

**Equivalence of test-time risk and training loss for train-val method**   We first show that

$$L^{\mathsf{tr\text{-}val}}(\mathbf{w}_0) = \mathbb{E}[\ell_t^{\mathsf{tr\text{-}val}}(\mathbf{w}_0)] = L_{\lambda, n_1}^{\mathsf{test}}(\mathbf{w}_0)$$

for all $\mathbf{w}_0$, that is, the population meta-test loss is exactly the same as the population risk of the train-val method. This is straightforward: as the tasks are i.i.d. and $\mathcal{A}_\lambda(\mathbf{w}_0; \mathbf{X}_t^{\text{train}}, \mathbf{y}_t^{\text{train}})$ is independent of the test points $(\mathbf{X}_t^{\text{val}}, \mathbf{y}_t^{\text{val}})$, we have for any $\mathbf{w}_0$ that

$$
\mathbb{E}[\ell_t^{\text{tr-val}}(\mathbf{w}_0)] = \mathbb{E}_{p_t \sim \Pi, (\mathbf{X}_t, \mathbf{y}_t) \sim p_t}\left[\frac{1}{2n_2}\left\|\mathbf{y}_t^{\text{val}} - \mathbf{X}_t^{\text{val}}\mathcal{A}_\lambda(\mathbf{w}_0; \mathbf{X}_t^{\text{train}}, \mathbf{y}_t^{\text{train}})\right\|^2\right]
$$

$$
= \mathbb{E}_{p_t \sim \Pi, (\mathbf{X}_t, \mathbf{y}_t) \sim p_t}\left[\frac{1}{2}\left(y_{t,1}^{\text{val}} - \mathbf{x}_{t,1}^{\text{val}\top}\mathcal{A}_\lambda(\mathbf{w}_0; \mathbf{X}_t^{\text{train}}, \mathbf{y}_t^{\text{train}})\right)^2\right]
$$

$$
= \mathbb{E}_{p_{T+1} \sim \Pi, (\mathbf{X}_{T+1}, \mathbf{y}_{T+1}), (\mathbf{x}', y') \overset{\text{iid}}{\sim} p_t}\left[\frac{1}{2}\left(y' - \mathbf{x}'^{\top}\mathcal{A}_{\lambda,n_1}(\mathbf{w}_0; \mathbf{X}_{T+1}, \mathbf{y}_{T+1})\right)^2\right]
$$

$$
= L_{\lambda,n_1}^{\text{test}}(\mathbf{w}_0).
$$

Therefore the train-val method is acutally a valid ERM for the test loss $L_{\lambda,n_1}^{\text{test}}$, and it remains to show that the train-val method is (itself) consistent.

**Consistency** We expand the empirical risk of the train-val method as

$$
\widehat{L}_T^{\text{tr-val}}(\mathbf{w}_0) = \frac{1}{T}\sum_{t=1}^{T}\frac{1}{2n_2}\left\|\mathbf{y}_t^{\text{val}} - \mathbf{X}_t^{\text{val}}\mathcal{A}_\lambda(\mathbf{w}_0; \mathbf{X}_t^{\text{train}}, \mathbf{y}_t^{\text{train}})\right\|^2
$$

$$
= \frac{1}{T}\sum_{t=1}^{T}\frac{1}{2n_2}\left\|\mathbf{y}_t^{\text{val}} - \mathbf{X}_t^{\text{val}}\left[\mathbf{w}_0 + (\mathbf{X}_t^{\text{train}\top}\mathbf{X}_t^{\text{train}} + n_1\lambda\mathbf{I}_d)^{-1}\mathbf{X}_t^{\text{train}\top}(\mathbf{y}_t^{\text{train}} - \mathbf{X}_t^{\text{train}}\mathbf{w}_0)\right]\right\|^2
$$

$$
= \frac{1}{T}\sum_{t=1}^{T}\frac{1}{2n_2}\left\|\mathbf{y}_t^{\text{val}} - \mathbf{X}_t^{\text{val}}(\mathbf{X}_t^{\text{train}\top}\mathbf{X}_t^{\text{train}} + n_1\lambda\mathbf{I}_d)^{-1}\mathbf{X}_t^{\text{train}\top}\mathbf{y}_t^{\text{train}} - \mathbf{X}_t^{\text{val}}n_1\lambda(\mathbf{X}_t^{\text{train}\top}\mathbf{X}_t^{\text{train}} + n_1\lambda\mathbf{I}_d)^{-1}\mathbf{w}_0\right\|^2
$$

$$
= \frac{1}{2}\mathbf{w}_0^{\top}\mathbf{M}_T\mathbf{w}_0 - \mathbf{w}_0^{\top}\mathbf{b}_T + \text{const},
$$

where

$$
\mathbf{M}_T := \frac{1}{T}\sum_{t=1}^{T}\lambda^2(\mathbf{X}_t^{\text{train}\top}\mathbf{X}_t^{\text{train}}/n_1 + \lambda\mathbf{I}_d)^{-1}\frac{\mathbf{X}_t^{\text{val}\top}\mathbf{X}_t^{\text{val}}}{n_2}(\mathbf{X}_t^{\text{train}\top}\mathbf{X}_t^{\text{train}}/n_1 + \lambda\mathbf{I}_d)^{-1},
$$

$$
\mathbf{b}_T := \frac{1}{T}\sum_{t=1}^{T}\lambda(\mathbf{X}_t^{\text{train}\top}\mathbf{X}_t^{\text{train}}/n_1 + \lambda\mathbf{I}_d)^{-1}\cdot\frac{1}{n_2}\mathbf{X}_t^{\text{val}\top}\left(\mathbf{y}_t^{\text{val}} - \mathbf{X}_t^{\text{val}}(\mathbf{X}_t^{\text{train}\top}\mathbf{X}_t^{\text{train}} + n_1\lambda\mathbf{I}_d)^{-1}\mathbf{X}_t^{\text{train}\top}\mathbf{y}_t^{\text{train}}\right).
$$

Noticing that $(\mathbf{X}_t^{\text{train}\top}\mathbf{X}_t^{\text{train}}/n_1 + \lambda\mathbf{I}_d)^{-1} \preceq \lambda^{-1}\mathbf{I}_d$ and by the assumption that $\mathbb{E}_{(\mathbf{x},y) \sim p_t}[\mathbf{x}\mathbf{x}^{\top}] \prec \infty$, $\mathbb{E}_{(\mathbf{x},y) \sim p_t}[\mathbf{x}y] < \infty$, we have $\mathbb{E}[\|\mathbf{M}_T\|] < \infty$ and $\mathbb{E}[\|\mathbf{b}_T\|] < \infty$. Since the task $p_t$'s are i.i.d., by the law of large numbers, we have with probability one that

$$
\mathbf{M}_T \to \mathbb{E}[\mathbf{M}_T]
$$

$$
= \mathbb{E}_{p_t, (\mathbf{X}_t, \mathbf{y}_t)}\left[\lambda^2(\mathbf{X}_t^{\text{train}\top}\mathbf{X}_t^{\text{train}}/n_1 + \lambda\mathbf{I}_d)^{-1}\frac{\mathbf{X}_t^{\text{val}\top}\mathbf{X}_t^{\text{val}}}{n_2}(\mathbf{X}_t^{\text{train}\top}\mathbf{X}_t^{\text{train}}/n_1 + \lambda\mathbf{I}_d)^{-1}\right] \quad (8)
$$

$$
= \mathbb{E}_{p_t, (\mathbf{X}_t, \mathbf{y}_t)}\left[\lambda^2(\mathbf{X}_t^{\text{train}\top}\mathbf{X}_t^{\text{train}}/n_1 + \lambda\mathbf{I}_d)^{-1}\mathbf{\Sigma}_t(\mathbf{X}_t^{\text{train}\top}\mathbf{X}_t^{\text{train}}/n_1 + \lambda\mathbf{I}_d)^{-1}\right] \succ \mathbf{0},
$$

(where $\mathbf{\Sigma}_t = \mathbb{E}_{\mathbf{x} \sim p_t}[\mathbf{x}\mathbf{x}^{\top}] \succ \mathbf{0}$) and

$$
\mathbf{b}_T \to \mathbb{E}[\mathbf{b}_T]
$$

$$
= \mathbb{E}_{p_t, (\mathbf{X}_t, \mathbf{y}_t)}\left[\lambda(\mathbf{X}_t^{\text{train}\top}\mathbf{X}_t^{\text{train}}/n_1 + \lambda\mathbf{I}_d)^{-1}\cdot\frac{1}{n_2}\mathbf{X}_t^{\text{val}\top}\left(\mathbf{y}_t^{\text{val}} - \mathbf{X}_t^{\text{val}}(\mathbf{X}_t^{\text{train}\top}\mathbf{X}_t^{\text{train}} + n_1\lambda\mathbf{I}_d)^{-1}\mathbf{X}_t^{\text{train}\top}\mathbf{y}_t^{\text{train}}\right)\right]
$$

$$
< \infty
$$

$$(9)$$

as $T \to \infty$. Therefore, by Slutsky's Theorem, we have

$$
\widehat{\mathbf{w}}_{0,T} = \mathbf{M}_T^{-1}\mathbf{b}_T \to \mathbb{E}[\mathbf{M}_T]^{-1}\mathbb{E}[\mathbf{b}_T] = \arg\min_{\mathbf{w}_0} L^{\text{tr-val}}(\mathbf{w}_0) = \arg\min_{\mathbf{w}_0} L_{\lambda,n_1}^{\text{test}}(\mathbf{w}_0) = \mathbf{w}_{0,\star}(\lambda, n_1)
$$

as $T \to \infty$. This proves the consistency of the train-val method.

**Asymptotic normality**    Similar as above, we can write the per-task loss as

$$\ell_t(\mathbf{w}_0) = \frac{1}{2} \left\| \mathbf{A}_t \mathbf{w}_0 - \mathbf{c}_t \right\|^2,$$

where

$$\mathbf{A}_t = \frac{\lambda}{\sqrt{n_2}} \mathbf{X}_t^{\mathsf{val}} (\mathbf{X}_t^{\mathsf{train}\top} \mathbf{X}_t^{\mathsf{train}}/n_1 + \lambda \mathbf{I}_d)^{-1},$$

$$\mathbf{c}_t = \frac{1}{\sqrt{n_2}} \left( \mathbf{y}_t^{\mathsf{val}} - \mathbf{X}_t^{\mathsf{val}} (\mathbf{X}_t^{\mathsf{train}\top} \mathbf{X}_t^{\mathsf{train}}/n_1 + \lambda \mathbf{I}_d)^{-1} \frac{1}{n_1} \mathbf{X}_t^{\mathsf{train}\top} \mathbf{y}_t^{\mathsf{train}} \right).$$

In order to show the desired asymptotic normality result, it suffices to check the conditions in Proposition 6. First, we have

$$\nabla \ell_t(\mathbf{w}_0) = \mathbf{A}_t^{\top} (\mathbf{A}_t \mathbf{w}_0 - \mathbf{c}_t).$$

This is Lipschitz in $\mathbf{w}_0$ with Lipschitz constant

$$\left\| \mathbf{A}_t^{\top} \mathbf{A}_t \right\|_{\mathsf{op}} \le \left\| \mathbf{A}_t \right\|_{\mathsf{Fr}}^2 < \frac{1}{n_2} \left\| \mathbf{X}_t^{\mathsf{val}} \right\|_{\mathsf{Fr}}^2.$$

As $\mathbb{E}_{\mathbf{x} \sim p_t}[\|\mathbf{x}\|^4] < \infty$, the above quantity is clearly square integrable, therefore verifying (a). As $\mathbf{w}_{0,\star} = \mathbf{w}_{0,\star}(\lambda, n_1)$ is finite, we can use similar arguments as above to show (b) holds. Finally, we have already seen $L$ is twice-differentiable (since it is quadratic in $\mathbf{w}_0$) and $\nabla^2 L(\mathbf{w}_{0,\star}) \succ \mathbf{0}$, which verifies (c). Therefore the conditions of Proposition 6 hold, which yields the desired asymptotic normality result. $\qquad\square$

### A.3    Proof of Proposition 2

**High-level idea**    At a high level, this proof proceeds by showing that the train-train method is also consistent to the (population) minimizer of $L^{\mathsf{tr\text{-}tr}}$, and constructing a simple counter-example on which the minimizers of $L^{\mathsf{tr\text{-}tr}}$ is not equal to that of $L_{\lambda,n}^{\mathsf{test}}$.

**Population minimizers of $L^{\mathsf{tr\text{-}tr}}$ and $L^{\mathsf{tr\text{-}val}}$**    We begin by simplifying the non-splitting risk. We have

$$
\begin{aligned}
\ell_t^{\mathsf{tr\text{-}tr}}(\mathbf{w}_0) &= \frac{1}{2n} \left\| \mathbf{y}_t - \mathbf{X}_t \mathcal{A}_\lambda(\mathbf{w}_0; \mathbf{X}_t, \mathbf{y}_t) \right\|^2 \\
&= \frac{1}{2n} \left\| \mathbf{y}_t - \mathbf{X}_t \left[ \mathbf{w}_0 + (\mathbf{X}_t^{\top} \mathbf{X}_t + n\lambda \mathbf{I}_d)^{-1} \mathbf{X}_t^{\top} (\mathbf{y}_t - \mathbf{X}_t \mathbf{w}_0) \right] \right\|^2 \\
&= \frac{1}{2} \left\| \mathbf{A}_t \mathbf{w}_0 - \mathbf{c}_t \right\|^2,
\end{aligned}
$$

where

$$\mathbf{A}_t = \frac{1}{\sqrt{n}} n\lambda \mathbf{X}_t (\mathbf{X}_t^{\top} \mathbf{X}_t + n\lambda \mathbf{I}_d)^{-1} \quad \text{and} \quad \mathbf{c}_t = \frac{1}{\sqrt{n}} \left( \mathbf{I}_n - \mathbf{X}_t (\mathbf{X}_t^{\top} \mathbf{X}_t + n\lambda \mathbf{I}_d)^{-1} \mathbf{X}_t^{\top} \right) \mathbf{y}_t.$$

Using similar arguments as in the proof of Proposition 1 (Appendix A.2), we see that the train-train method $\widehat{\mathbf{w}}_{0,T}^{\mathsf{tr\text{-}tr}}$ converges with probability one to the minimizer of the pouplation risk $L^{\mathsf{tr\text{-}tr}}$, which is

$$
\begin{aligned}
\mathbf{w}_{0,\star}^{\mathsf{tr\text{-}tr}} &= \underset{\mathbf{w}_0}{\arg\min}\, L^{\mathsf{tr\text{-}tr}}(\mathbf{w}_0) = \left( \mathbb{E}[\mathbf{A}_t^{\top} \mathbf{A}_t] \right)^{-1} \mathbb{E}[\mathbf{A}_t^{\top} \mathbf{c}_t] \\
&= \mathbb{E}\left[ \lambda^2 (\mathbf{X}_t^{\top} \mathbf{X}_t/n + \lambda \mathbf{I}_d)^{-2} \frac{\mathbf{X}_t^{\top} \mathbf{X}_t}{n} \right]^{-1} \cdot \mathbb{E}\left[ \frac{1}{n} \lambda (\mathbf{X}_t^{\top} \mathbf{X}_t/n + \lambda \mathbf{I}_d)^{-1} \mathbf{X}_t^{\top} (\mathbf{I}_n - \mathbf{X}_t (\mathbf{X}_t^{\top} \mathbf{X}_t + n\lambda \mathbf{I}_d)^{-1} \mathbf{X}_t^{\top}) \mathbf{y}_t \right] \\
&= \mathbb{E}\left[ \lambda^2 (\mathbf{X}_t^{\top} \mathbf{X}_t/n + \lambda \mathbf{I}_d)^{-2} \frac{\mathbf{X}_t^{\top} \mathbf{X}_t}{n} \right]^{-1} \cdot \mathbb{E}\left[ \lambda^2 (\mathbf{X}_t^{\top} \mathbf{X}_t/n + \lambda \mathbf{I}_d)^{-2} \frac{1}{n} \mathbf{X}_t^{\top} \mathbf{y}_t \right].
\end{aligned}
$$

(10)

On the other hand, recall from Proposition 1 ((8) and (9)) that the population minimizer of $L_{\lambda,n}^{\text{test}}$ is

$$\mathbf{w}_{0,\star}(\lambda, n) = \arg\min_{\mathbf{w}_0} L_{\lambda,n}^{\text{test}}(\mathbf{w}_0)$$

$$= \mathbb{E}\big[\lambda^2(\mathbf{X}_t^\top\mathbf{X}_t/n + \lambda\mathbf{I}_d)^{-1}\mathbf{\Sigma}_t(\mathbf{X}_t^\top\mathbf{X}_t/n + \lambda\mathbf{I}_d)^{-1}\big]^{-1} \cdot \bigg\{\lambda\mathbb{E}\big[(\mathbf{X}_t^\top\mathbf{X}_t/n + \lambda\mathbf{I}_d)^{-1}\big]\mathbb{E}_{p_t,(\mathbf{x}',y')\sim p_t}[\mathbf{x}'y']$$

$$- \lambda\mathbb{E}\bigg[(\mathbf{X}_t^\top\mathbf{X}_t/n + \lambda\mathbf{I}_d)^{-1}\mathbf{\Sigma}_t(\mathbf{X}_t^\top\mathbf{X}_t/n + \lambda\mathbf{I}_d)^{-1}\frac{1}{n}\mathbf{X}_t^\top\mathbf{y}_t\bigg]\bigg\}. \tag{11}$$

**Construction of the counter-example** We now construct a distribution for which (10) is not equal to (11). Let $d = 1$ and let all $p_t$ be the following distribution:

$$p_t: \quad (x_{t,i}, y_{t,i}) = \begin{cases} (1,3) & \text{with probability } 1/2; \\ (3,-1) & \text{with probability } 1/2. \end{cases}$$

Clearly, we have $\mathbf{\Sigma}_t = 5$, $s_t := \mathbf{X}_t^\top\mathbf{X}_t/n \in [1, 9]$, and $\mathbb{E}_{x',y'\sim p_t}[x'y'] = 0$. Therefore we have

$$\mathbf{w}_{0,\star}^{\text{tr-tr}} = \mathbb{E}\big[(s_t + \lambda)^{-2}s_t\big]^{-1} \cdot \mathbb{E}\bigg[(s_t + \lambda)^{-2}\frac{1}{n}\sum_{i=1}^{n} x_{t,i}y_{t,i}\bigg],$$

and

$$\mathbf{w}_{0,\star}(\lambda, n) = -\mathbb{E}\big[5\lambda^2(s_t + \lambda)^{-2}\big]^{-1} \cdot \mathbb{E}\bigg[5\lambda(s_t + \lambda)^{-2}\frac{1}{n}\sum_{i=1}^{n} x_{t,i}y_{t,i}\bigg]$$

$$= -\mathbb{E}\big[\lambda(s_t + \lambda)^{-2}\big]^{-1} \cdot \mathbb{E}\bigg[(s_t + \lambda)^{-2}\frac{1}{n}\sum_{i=1}^{n} x_{t,i}y_{t,i}\bigg].$$

We now show that $\mathbf{w}_{0,\star}^{\text{tr-tr}} \neq \mathbf{w}_{0,\star}(\lambda, n)$ by showing that

$$\mathbb{E}\bigg[(s_t + \lambda)^{-2}\frac{1}{n}\sum_{i=1}^{n} x_{t,i}y_{t,i}\bigg] = \mathbb{E}\bigg[\frac{x_{t,1}y_{t,1}}{(s_t + \lambda)^2}\bigg] \neq 0$$

for any $\lambda > 0$. Indeed, conditioning on $(x_{t,1}, y_{t,1}) = (1,3)$, we know that the sum-of-squares in $s_t$ has one term that equals 1, and all others i.i.d. being 1 or 9 with probability one half. On the other hand, if we condition on $(x_{t,1}, y_{t,1}) = (3,-1)$, then we know the sum in $s_t$ has one term that equals 9 and all others i.i.d.. This means that the negative contribution in the expectation is smaller than the positive contribution, in other words

$$\mathbb{E}\bigg[\frac{x_{t,1}y_{t,1}}{(s_t + \lambda)^2}\bigg] = \frac{1}{2} \cdot 3\mathbb{E}\bigg[\frac{1}{(s_t + \lambda)^2}\bigg|(x_{t,1}, y_{t,1}) = (1,3)\bigg]$$

$$+ \frac{1}{2} \cdot -3\mathbb{E}\bigg[\frac{1}{(s_t + \lambda)^2}\bigg|(x_{t,1}, y_{t,1}) = (3,-1)\bigg] > 0.$$

This shows $\mathbf{w}_{0,\star}^{\text{tr-tr}} \neq \mathbf{w}_{0,\star}(\lambda, n)$ and consequently the $\widehat{\mathbf{w}}_{0,T}^{\text{tr-tr}}$ does not converge to $\mathbf{w}_{0,\star}(\lambda, n)$ and the difference is bounded away from zero as $T \to \infty$.

Finally, for this distribution, the risk $L_{\lambda,n}^{\text{test}}(\mathbf{w}_0)$ is strongly convex (since it has a positive second derivative), this further implies that $L_{\lambda,n}^{\text{test}}(\widehat{\mathbf{w}}_{0,T}^{\text{tr-tr}}) - L_{\lambda,n}^{\text{test}}(\mathbf{w}_{0,\star}(\lambda, n))$ is bounded away from zero almost surely as $T \to \infty$.

## B  PROOFS FOR SECTION 4

### B.1  PROOF OF THEOREM 3

We first show that $\mathbf{w}_{0,\star} = \mathbb{E}_{\mathbf{w}_t\sim\Pi}[\mathbf{w}_t]$ is a global optimizer for $L^{\text{tr-tr}}$ and $L^{\text{tr-val}}$ with any regularization coefficient $\lambda > 0$, any $n$, and any split $(n_1, n_2)$. To do this, it suffices to check that the gradient at $\mathbf{w}_{0,\star}$ is zero and the Hessian is positive definite (PD).

**Optimality of $\mathbf{w}_{0,\star}$ in both $L^{\text{tr-tr}}$ and $L^{\text{tr-val}}$.** We first look at $L^{\text{tr-tr}}$: for any $\mathbf{w}_0 \in \mathbb{R}^d$ we have

$$
\begin{aligned}
L^{\text{tr-tr}}(\mathbf{w}_0) &= \mathbb{E}[\ell_t^{\text{tr-tr}}(\mathbf{w}_0)] \\
&= \frac{1}{2n}\mathbb{E}\left[\left\|\mathbf{X}_t\mathbf{w}_t - \mathbf{X}_t\left[\left(\mathbf{X}_t^\top\mathbf{X}_t + n\lambda\mathbf{I}_d\right)^{-1}\mathbf{X}_t^\top\left(\mathbf{X}_t\mathbf{w}_t - \mathbf{X}_t\mathbf{w}_0\right) + \mathbf{w}_0\right]\right\|^2\right] \\
&= \frac{1}{2n}\mathbb{E}\left[\left\|\mathbf{X}_t\left(\mathbf{I}_d - \left(\mathbf{X}_t^\top\mathbf{X}_t + n\lambda\mathbf{I}_d\right)^{-1}\mathbf{X}_t^\top\mathbf{X}_t\right)(\mathbf{w}_t - \mathbf{w}_0)\right\|^2\right].
\end{aligned}
\tag{12}
$$

Similarly, $L^{\text{tr-val}}$ can be written as

$$
L^{\text{tr-val}}(\mathbf{w}_0)
\tag{13}
$$
$$
= \mathbb{E}[\ell_t^{\text{tr-val}}(\mathbf{w}_0)]
$$
$$
= \frac{1}{2n_2}\mathbb{E}\left[\left\|\mathbf{X}_t^{\text{val}}\mathbf{w}_t - \mathbf{X}_t^{\text{val}}\left[\left((\mathbf{X}_t^{\text{train}})^\top\mathbf{X}_t^{\text{train}} + n_1\lambda\mathbf{I}_d\right)^{-1}(\mathbf{X}_t^{\text{train}})^\top\left(\mathbf{X}_t^{\text{train}}\mathbf{w}_t - \mathbf{X}_t^{\text{train}}\mathbf{w}_0\right) + \mathbf{w}_0\right]\right\|^2\right]
$$
$$
= \frac{1}{2n_2}\mathbb{E}\left[\left\|\mathbf{X}_t^{\text{val}}\left(\mathbf{I}_d - \left((\mathbf{X}_t^{\text{train}})^\top\mathbf{X}_t^{\text{train}} + n_1\lambda\mathbf{I}_d\right)^{-1}(\mathbf{X}_t^{\text{train}})^\top\mathbf{X}_t^{\text{train}}\right)(\mathbf{w}_t - \mathbf{w}_0)\right\|^2\right].
\tag{14}
$$

We denote

$$
\mathbf{M}_t^{\text{tr-tr}} = \mathbf{X}_t\left(\mathbf{I}_d - \left(\mathbf{X}_t^\top\mathbf{X}_t + n\lambda\mathbf{I}_d\right)^{-1}\mathbf{X}_t^\top\mathbf{X}_t\right) \quad\text{and}
\tag{15}
$$

$$
\mathbf{M}_t^{\text{tr-val}} = \mathbf{X}_t^{\text{val}}\left(\mathbf{I}_d - \left((\mathbf{X}_t^{\text{train}})^\top\mathbf{X}_t^{\text{train}} + n_1\lambda\mathbf{I}_d\right)^{-1}(\mathbf{X}_t^{\text{train}})^\top\mathbf{X}_t^{\text{train}}\right)
\tag{16}
$$

to simplify the notations in (12) and (14). We take gradient of $L^{\text{tr-tr}}$ and $L^{\text{tr-val}}$ with respect to $\mathbf{w}_0$:

$$
\nabla_{\mathbf{w}_0}L^{\text{tr-tr}}(\mathbf{w}_0) = -\frac{1}{n}\mathbb{E}\left[(\mathbf{M}_t^{\text{tr-tr}})^\top\mathbf{M}_t^{\text{tr-tr}}(\mathbf{w}_t - \mathbf{w}_0)\right],
\tag{17}
$$

$$
\nabla_{\mathbf{w}_0}L^{\text{tr-val}}(\mathbf{w}_0) = -\frac{1}{n_2}\mathbb{E}\left[(\mathbf{M}_t^{\text{tr-val}})^\top\mathbf{M}_t^{\text{tr-val}}(\mathbf{w}_t - \mathbf{w}_0)\right].
\tag{18}
$$

Substituting $\mathbf{w}_{0,\star}$ into (17) and taking expectation, we deduce

$$
\nabla_{\mathbf{w}_0}L^{\text{tr-tr}}(\mathbf{w}_{0,\star}) = -\frac{1}{n}\mathbb{E}\left[(\mathbf{M}_t^{\text{tr-tr}})^\top\mathbf{M}_t^{\text{tr-tr}}(\mathbf{w}_t - \mathbf{w}_{0,\star})\right] = \mathbf{0}.
\tag{19}
$$

To see this, observe that by definition $\mathbb{E}[\mathbf{w}_t - \mathbf{w}_{0,\star}] = \mathbf{0}$. Combining with $\mathbf{w}_t$ being generated independently of $\mathbf{X}_t$, we have the first term in RHS of (19) vanish. In addition, $\mathbf{z}_t$ is independent white noise, therefore, the second term in RHS of (19) also vanishes. Following the same argument, we can show

$$
\nabla_{\mathbf{w}_0}L^{\text{tr-val}}(\mathbf{w}_{0,\star}) = \mathbf{0},
$$

since $\mathbf{X}_t'$ is also independent of $\mathbf{w}_t$. The above reasonings indicates that $\mathbf{w}_{0,\star}$ is a stationary point of both $L^{\text{tr-tr}}$ and $L^{\text{tr-val}}$. The remaining step is to check $\nabla_{\mathbf{w}_0}L^{\text{tr-tr}}(\mathbf{w}_{0,\star})$ and $\nabla_{\mathbf{w}_0}L^{\text{tr-val}}(\mathbf{w}_{0,\star})$ are PD. From (17) and (18), we derive respectively the hessian of $L^{\text{tr-tr}}$ and $L^{\text{tr-val}}$ as

$$
\nabla_{\mathbf{w}_0}^2 L^{\text{tr-tr}}(\mathbf{w}_{0,\star}) = \frac{1}{n}\mathbb{E}[(\mathbf{M}_t^{\text{tr-tr}})^\top\mathbf{M}_t^{\text{tr-tr}}] \quad\text{and}
$$

$$
\nabla_{\mathbf{w}_0}^2 L^{\text{tr-val}}(\mathbf{w}_{0,\star}) = \frac{1}{n_2}\mathbb{E}[(\mathbf{M}_t^{\text{tr-val}})^\top\mathbf{M}_t^{\text{tr-val}}].
$$

Let $\mathbf{v} \in \mathbb{R}^d$ be any nonzero vector, we check $\mathbf{v}^\top\nabla_{\mathbf{w}_0}^2 L^{\text{tr-tr}}(\mathbf{w}_{0,\star})\mathbf{v} > 0$. A key observation is that $\left(\mathbf{I}_d - \left(\mathbf{X}_t^\top\mathbf{X}_t + n\lambda\mathbf{I}_d\right)^{-1}\mathbf{X}_t^\top\mathbf{X}_t\right)$ is positive definite for any $\lambda \neq 0$. To see this, let $\sigma_1 \geq \cdots \geq \sigma_d$ be eigenvalues of $\frac{1}{n}\mathbf{X}_t^\top\mathbf{X}_t$, some algebra yields the eigenvalues of $\left(\mathbf{I}_d - \left(\mathbf{X}_t^\top\mathbf{X}_t + n\lambda\mathbf{I}_d\right)^{-1}\mathbf{X}_t^\top\mathbf{X}_t\right)$ are $\frac{\lambda}{\lambda+\sigma_i} > 0$ for $\lambda \neq 0$ and $i = 1, \ldots, d$. Hence, we deduce

$$
\mathbf{v}^\top\nabla_{\mathbf{w}_0}^2 L^{\text{tr-tr}}(\mathbf{w}_{0,\star})\mathbf{v} = \frac{1}{n}\mathbb{E}[\mathbf{v}^\top\mathbf{X}_t^\top\left(\mathbf{I}_d - \left(\mathbf{X}_t^\top\mathbf{X}_t + n\lambda\mathbf{I}_d\right)^{-1}\mathbf{X}_t^\top\mathbf{X}_t\right)^2\mathbf{X}_t\mathbf{v}] > 0,
\tag{20}
$$

since $\mathbf{X}_t$ is isotropic (an explicit computation of the hessian matrix can be found in Appendix B.2). As a consequence, we have shown that $\mathbf{w}_{0,\star}$ is a global optimum of $L^{\text{tr-tr}}$. The same argument applies to $L^{\text{tr-val}}$, and the proof is complete.

**Consistency of** $\widehat{\mathbf{w}}_{0,T}^{\{\text{tr-tr,tr-val}\}}$. To check the consistency, we need to verify the conditions (a) – (c) in Proposition 6.

For condition (a), we derive from (17) and (18) that

$$\left\|\nabla\ell_t^{\{\text{tr-tr,tr-val}\}}(\mathbf{w}_1) - \nabla\ell_t^{\{\text{tr-tr,tr-val}\}}(\mathbf{w}_2)\right\| \leq \frac{1}{n}\left\|(\mathbf{M}_t^{\{\text{tr-tr,tr-val}\}})^\top\mathbf{M}_t^{\{\text{tr-tr,tr-val}\}}\right\|_{\text{op}}\|\mathbf{w}_1 - \mathbf{w}_2\|,$$

where $n$ should be replaced by $n_2$ for the split method (we slightly abuse the notation for simplicity). It suffices to show $\mathbb{E}\left[\left\|(\mathbf{M}_t^{\{\text{tr-tr,tr-val}\}})^\top\mathbf{M}_t^{\{\text{tr-tr,tr-val}\}}\right\|_{\text{op}}\right] < \infty$, which follows from the same argument in the proof of Proposition 1. In particular, we know $\mathbf{0} \preceq \mathbf{I}_d - \left(\mathbf{X}_t^\top\mathbf{X}_t + n\lambda\mathbf{I}_d\right)^{-1}\mathbf{X}_t^\top\mathbf{X}_t \preceq \mathbf{I}_d$ and $\mathbb{E}[\|\mathbf{X}_t^\top\mathbf{X}_t\|_{\text{op}}] < \infty$ since $\mathbf{X}_t$ is Gaussian. As a consequence, for the no-split method, we have $\mathbb{E}\left[\|(\mathbf{M}_t^{\text{tr-tr}})^\top\mathbf{M}_t^{\text{tr-tr}}\|_{\text{op}}^2\right] < \infty$. For the split method, we also have $\mathbf{0} \preceq \mathbf{I}_d - \left((\mathbf{X}_t^{\text{train}})^\top\mathbf{X}_t^{\text{train}} + n\lambda\mathbf{I}_d\right)^{-1}(\mathbf{X}_t^{\text{train}})^\top\mathbf{X}_t^{\text{train}} \preceq \mathbf{I}_d$ and $\mathbb{E}[\|(\mathbf{X}_t^{\text{val}})^\top\mathbf{X}_t^{\text{val}}\|_{\text{op}}] < \infty$, which implies $\mathbb{E}\left[\|(\mathbf{M}_t^{\text{tr-val}})^\top\mathbf{M}_t^{\text{tr-val}}\|_{\text{op}}^2\right] < \infty$.

For condition (b), using a similar argument in condition (a) and combining with $R^2 = \mathbb{E}[\|\mathbf{w}_{0,\star} - \mathbf{w}_t\|^2]$, we have $\mathbb{E}[\|\nabla\ell_t^{\{\text{tr-tr,tr-val}\}}(\mathbf{w}_{0,\star})\|^2] < \infty$.

For condition (c), using (20), we directly verify that $L^{\{\text{tr-tr,tr-val}\}}$ is twice-differentiable and $\nabla^2 L^{\{\text{tr-tr,tr-val}\}} \succ \mathbf{0}$.

### B.2 Proof of Lemma 5

*Proof.* In this section we prove Lemma 5. Using the the asymptotic normality in Proposition 6, the asymptotic covariance is $\nabla^{-2}L^{\{\text{tr-tr,tr-val}\}}\text{Cov}[\nabla\ell_t^{\{\text{tr-tr,tr-val}\}}]\nabla^{-2}L^{\{\text{tr-tr,tr-val}\}}$. Therefore, in the following, we only need to find $\nabla^{-2}L^{\{\text{tr-tr,tr-val}\}}$ and $\text{Cov}[\nabla\ell_t^{\{\text{tr-tr,tr-val}\}}]$.

• **Asymptotic variance of** $\widehat{\mathbf{w}}_{0,T}^{\text{tr-tr}}$. We begin with the computation of the expected Hessian $\frac{1}{n}\mathbb{E}[(\mathbf{M}_t^{\text{tr-tr}})^\top\mathbf{M}_t^{\text{tr-tr}}]$.

$$\mathbb{E}[(\mathbf{M}_t^{\text{tr-tr}})^\top\mathbf{M}_t^{\text{tr-tr}}]$$
$$= \mathbb{E}\left[\left(\mathbf{I}_d - \left(\mathbf{X}_t^\top\mathbf{X}_t + n\lambda\mathbf{I}_d\right)^{-1}\mathbf{X}_t^\top\mathbf{X}_t\right)^\top\mathbf{X}_t^\top\mathbf{X}_t\left(\mathbf{I}_d - \left(\mathbf{X}_t^\top\mathbf{X}_t + n\lambda\mathbf{I}_d\right)^{-1}\mathbf{X}_t^\top\mathbf{X}_t\right)\right]$$
$$\stackrel{(i)}{=} \mathbb{E}\left[\mathbf{V}_t\left(\mathbf{I}_d - (\mathbf{D}_t^\top\mathbf{D}_t + n\lambda\mathbf{I}_d)^{-1}\mathbf{D}_t^\top\mathbf{D}_t\right)^\top\mathbf{D}_t^\top\mathbf{D}_t\left(\mathbf{I}_d - (\mathbf{D}_t^\top\mathbf{D}_t + n\lambda\mathbf{I}_d)^{-1}\mathbf{D}_t^\top\mathbf{D}_t\right)\mathbf{V}_t^\top\right], \tag{21}$$

where the equality $(i)$ is obtained by plugging in the SVD of $\mathbf{X}_t = \mathbf{U}_t\mathbf{D}_t\mathbf{V}_t^\top$ with $\mathbf{U}_t \in \mathbb{R}^{n\times n}$, $\mathbf{D}_t \in \mathbb{R}^{n\times d}$, and $\mathbf{V}_t \in \mathbb{R}^{d\times d}$. A key observation is that $\mathbf{U}_t$ and $\mathbf{V}_t$ are independent, since $\mathbf{X}_t$ is isotropic, i.e., homogeneous in each orthogonal direction. To see this, for any orthogonal matrices $\mathbf{Q} \in \mathbb{R}^{n\times n}$ and $\mathbf{P} \in \mathbb{R}^{d\times d}$, we know $\mathbf{X}_t$ and $\mathbf{Q}\mathbf{X}_t\mathbf{P}^\top$ share the same distribution. Moreover, we have $\mathbf{Q}\mathbf{X}_t\mathbf{P}^\top = (\mathbf{Q}\mathbf{U}_t)\mathbf{D}_t(\mathbf{P}\mathbf{V}_t)^\top$ as the SVD. This shows that the left and right singular matrices are independent and both uniformly distributed on all the orthogonal matrices of the corresponding dimensions ($\mathbb{R}^{n\times n}$ and $\mathbb{R}^{d\times d}$, respectively).

Recall that we denote $\sigma_1^{(n)} \geq \cdots \geq \sigma_d^{(n)}$ as the eigenvalues of $\frac{1}{n}\mathbf{X}_t^\top\mathbf{X}_t$. Thus, we have $\mathbf{D}_t^\top\mathbf{D}_t = \text{Diag}(n\sigma_1^{(n)}, \ldots, n\sigma_d^{(n)})$. We can further simplify (21) as

$$\mathbb{E}\left[\mathbf{V}_t\left(\mathbf{I}_d - (\mathbf{D}_t^\top\mathbf{D}_t + n\lambda\mathbf{I}_d)^{-1}\mathbf{D}_t^\top\mathbf{D}_t\right)^\top\mathbf{D}_t^\top\mathbf{D}_t\left(\mathbf{I}_d - (\mathbf{D}_t^\top\mathbf{D}_t + n\lambda\mathbf{I}_d)^{-1}\mathbf{D}_t^\top\mathbf{D}_t\right)\mathbf{V}_t^\top\right]$$
$$= \mathbb{E}\left[\mathbf{V}_t\text{Diag}\left(\frac{n\lambda^2\sigma_1^{(n)}}{(\sigma_1^{(n)} + \lambda)^2}, \ldots, \frac{n\lambda^2\sigma_d^{(n)}}{(\sigma_d^{(n)} + \lambda)^2}\right)\mathbf{V}_t^\top\right]$$
$$= \mathbb{E}\left[\sum_{i=1}^d \frac{n\lambda^2\sigma_i^{(n)}}{(\sigma_i^{(n)} + \lambda)^2}\mathbf{v}_{t,i}\mathbf{v}_{t,i}^\top\right]. \tag{22}$$

We will utilize the isotropicity of $\mathbf{X}_t$ to find (22). Recall that we have shown that $\mathbf{V}_t$ is uniform on all the orthogonal matrices. Let $\mathbf{P} \in \mathbb{R}^{d \times d}$ be any permutation matrix, then $\mathbf{V}_t \mathbf{P}$ has the same distribution as $\mathbf{V}_t$. For this permuted data matrix $\mathbf{V}_t \mathbf{P}$, (22) becomes

$$\mathbb{E}\left[\sum_{i=1}^{d} \frac{n\lambda^2 \sigma_i^{(n)}}{(\sigma_i^{(n)} + \lambda)^2} \mathbf{v}_{t,\tau_p(i)} \mathbf{v}_{t,\tau_p(i)}^\top\right] \quad \text{with } \tau_p(i) \text{ denotes the permutation of the } i\text{-th element in } \mathbf{P}.$$

Summing over all the permutations $\mathbf{P}$ (and there are totally $d!$ instances), we deduce

$$\begin{aligned}
&d! \mathbb{E}[(\mathbf{M}_t^{\text{tr-tr}})^\top \mathbf{M}_t^{\text{tr-tr}}] \\
&= \sum_{\text{all permutation } \tau_p} \mathbb{E}\left[\sum_{i=1}^{d} \frac{n\lambda^2 \sigma_i^{(n)}}{(\sigma_i^{(n)} + \lambda)^2} \mathbf{v}_{t,\tau_p(i)} \mathbf{v}_{t,\tau_p(i)}^\top\right] \\
&= (d-1)! \mathbb{E}\left[\sum_{j=1}^{d} \left[\sum_{i=1}^{d} \frac{n\lambda^2 \sigma_i^{(n)}}{(\sigma_i^{(n)} + \lambda)^2}\right] \mathbf{v}_{t,j} \mathbf{v}_{t,j}^\top\right] \\
&= (d-1)! \mathbb{E}\left[\mathbf{V}_t \text{Diag}\left(\sum_{i=1}^{d} \frac{\lambda^2 \sigma_i^{(n)}}{(\lambda + \sigma_i^{(n)})^2}, \ldots, \sum_{i=1}^{d} \frac{\lambda^2 \sigma_i^{(n)}}{(\lambda + \sigma_i^{(n)})^2}\right) \mathbf{V}_t^\top\right] \\
&= (d-1)! \mathbb{E}\left[\sum_{i=1}^{d} \frac{\lambda^2 \sigma_i^{(n)}}{(\lambda + \sigma_i^{(n)})^2} \mathbf{V}_t \mathbf{I}_d \mathbf{V}_t^\top\right] \tag{23}
\end{aligned}$$

Dividing $(d-1)!$ on both sides of (23) yields

$$\mathbb{E}[(\mathbf{M}_t^{\text{tr-tr}})^\top \mathbf{M}_t^{\text{tr-tr}}] = \frac{n}{d} \mathbb{E}\left[\sum_{i=1}^{d} \frac{\lambda^2 \sigma_i^{(n)}}{(\lambda + \sigma_i^{(n)})^2}\right] \mathbf{I}_d. \tag{24}$$

Next, we find the expected covariance matrix $\frac{1}{n^2} \mathbb{E}[\nabla \ell_t^{\text{tr-tr}}(\mathbf{w}_{0,\star})(\nabla \ell_t^{\text{tr-tr}}(\mathbf{w}_{0,\star}))^\top]$.

$$\begin{aligned}
&\mathbb{E}[\nabla \ell_t^{\text{tr-tr}}(\mathbf{w}_{0,\star})(\nabla \ell_t^{\text{tr-tr}}(\mathbf{w}_{0,\star}))^\top] \\
&= \mathbb{E}[(\mathbf{M}_t^{\text{tr-tr}})^\top \mathbf{M}_t^{\text{tr-tr}}(\mathbf{w}_{0,\star} - \mathbf{w}_t)(\mathbf{w}_{0,\star} - \mathbf{w}_t)^\top (\mathbf{M}_t^{\text{tr-tr}})^\top \mathbf{M}_t^{\text{tr-tr}}] \\
&\overset{(i)}{=} \mathbb{E}\left[\mathbf{V}_t \text{Diag}\left(\frac{n\lambda^2 \sigma_1^{(n)}}{(\sigma_1^{(n)} + \lambda)^2}, \ldots, \frac{n\lambda^2 \sigma_d^{(n)}}{(\sigma_d^{(n)} + \lambda)^2}\right) \mathbf{V}_t^\top (\mathbf{w}_{0,\star} - \mathbf{w}_t)(\mathbf{w}_{0,\star} - \mathbf{w}_t)^\top \right. \\
&\quad \left. \cdot \mathbf{V}_t \text{Diag}\left(\frac{n\lambda^2 \sigma_1^{(n)}}{(\sigma_1^{(n)} + \lambda)^2}, \ldots, \frac{n\lambda^2 \sigma_d^{(n)}}{(\sigma_d^{(n)} + \lambda)^2}\right) \mathbf{V}_t^\top\right]. \tag{25}
\end{aligned}$$

Here step $(i)$ uses the SVD of $\mathbf{X}_t$ and the computation in (22). Combining (24) and (25), we derive the asymptotic covariance matrix of using $L^{\text{tr-tr}}$ as

$$\begin{aligned}
&\text{AsymCov}(\widehat{\mathbf{w}}_{0,T}^{\text{tr-tr}}) \\
&= \mathbb{E}[\nabla^{-2} \ell_t^{\text{tr-tr}}] \text{Cov}[\nabla \ell_t^{\text{tr-tr}}(\mathbf{w}_{0,\star})] \mathbb{E}[\nabla^{-2} \ell_t^{\text{tr-tr}}] \\
&= \frac{d^2}{n^2} \left(\mathbb{E}\left[\sum_{i=1}^{d} \frac{\lambda^2 \sigma_i^{(n)}}{(\lambda + \sigma_i^{(n)})^2}\right]\right)^{-2} \\
&\quad \cdot \mathbb{E}\left[\mathbf{V}_t \text{Diag}\left(\frac{n\lambda^2 \sigma_1^{(n)}}{(\sigma_1^{(n)} + \lambda)^2}, \ldots, \frac{n\lambda^2 \sigma_d^{(n)}}{(\sigma_d^{(n)} + \lambda)^2}\right) \mathbf{V}_t^\top (\mathbf{w}_{0,\star} - \mathbf{w}_t)(\mathbf{w}_{0,\star} - \mathbf{w}_t)^\top \right. \\
&\quad \left. \cdot \mathbf{V}_t \text{Diag}\left(\frac{n\lambda^2 \sigma_1^{(n)}}{(\sigma_1^{(n)} + \lambda)^2}, \ldots, \frac{n\lambda^2 \sigma_d^{(n)}}{(\sigma_d^{(n)} + \lambda)^2}\right) \mathbf{V}_t^\top\right]. \tag{26}
\end{aligned}$$

Taking trace in (26), we deduce

$$\text{AsymMSE}(\widehat{\mathbf{w}}_{0,T}^{\text{tr-tr}})$$

$$= \operatorname{tr}(\operatorname{AsymCov}(\widehat{\mathbf{w}}_{0,T}^{\mathsf{tr\text{-}tr}}))$$

$$= \frac{d^2}{n^2} \left( \mathbb{E} \left[ \sum_{i=1}^{d} \frac{\lambda^2 \sigma_i^{(n)}}{(\lambda + \sigma_i^{(n)})^2} \right] \right)^{-2}$$

$$\cdot \operatorname{tr} \left( \mathbb{E} \left[ \mathbf{V}_t \operatorname{Diag} \left( \frac{n^2 \lambda^4 (\sigma_1^{(n)})^2}{(\sigma_1^{(n)} + \lambda)^4}, \dots, \frac{n^2 \lambda^4 (\sigma_d^{(n)})^2}{(\sigma_d^{(n)} + \lambda)^4} \right) \mathbf{V}_t^\top (\mathbf{w}_{0,\star} - \mathbf{w}_t)(\mathbf{w}_{0,\star} - \mathbf{w}_t)^\top \right] \right)$$

$$\overset{(i)}{=} \frac{d^2}{n^2} \left( \mathbb{E} \left[ \sum_{i=1}^{d} \frac{\lambda^2 \sigma_i^{(n)}}{(\lambda + \sigma_i^{(n)})^2} \right] \right)^{-2}$$

$$\cdot \frac{n^2}{d} \operatorname{tr} \left( \mathbb{E} \left[ \sum_{i=1}^{d} \frac{\lambda^4 (\sigma_i^{(n)})^2}{(\lambda + \sigma_i^{(n)})^4} \mathbf{V}_t \mathbf{I}_d \mathbf{V}_t^\top (\mathbf{w}_{0,\star} - \mathbf{w}_t)(\mathbf{w}_{0,\star} - \mathbf{w}_t)^\top \right] \right)$$

$$= d \frac{\mathbb{E} \left[ \sum_{i=1}^{d} \lambda^4 (\sigma_i^{(n)})^2 / (\lambda + \sigma_i^{(n)})^4 \right]}{\left( \mathbb{E} \left[ \sum_{i=1}^{d} \lambda^2 \sigma_i^{(n)} / (\lambda + \sigma_i^{(n)})^2 \right] \right)^2} \cdot \operatorname{tr} \left( \mathbb{E}[(\mathbf{w}_{0,\star} - \mathbf{w}_t)(\mathbf{w}_{0,\star} - \mathbf{w}_t)^\top] \right)$$

$$= dR^2 \frac{\mathbb{E} \left[ \sum_{i=1}^{d} (\sigma_i^{(n)})^2 / (\lambda + \sigma_i^{(n)})^4 \right]}{\left( \mathbb{E} \left[ \sum_{i=1}^{d} \sigma_i^{(n)} / (\lambda + \sigma_i^{(n)})^2 \right] \right)^2}, \tag{27}$$

where step $(i)$ utilizes the independence between $\mathbf{w}_t$ and $\mathbf{X}_t$ and applies the permutation trick in (23) to find $\mathbb{E} \left[ \mathbf{V}_t \operatorname{Diag} \left( \frac{n^2 \lambda^4 (\sigma_1^{(n)})^2}{(\sigma_1^{(n)} + \lambda)^4}, \dots, \frac{n^2 \lambda^4 (\sigma_d^{(n)})^2}{(\sigma_d^{(n)} + \lambda)^4} \right) \mathbf{V}_t^\top \right]$.

- **Asymptotic variance of $\widehat{\mathbf{w}}_{0,T}^{\mathsf{tr\text{-}val}}$.** Similar to the no-split case, we compute the Hessian $\frac{1}{n_2} \mathbb{E}[\nabla^2 \ell_t^{\mathsf{tr\text{-}val}}] = \frac{1}{n_2} \mathbb{E}[(\mathbf{M}_t^{\mathsf{tr\text{-}val}})^\top \mathbf{M}_t^{\mathsf{tr\text{-}val}}]$ first.

$$\mathbb{E}[(\mathbf{M}_t^{\mathsf{tr\text{-}val}})^\top \mathbf{M}_t^{\mathsf{tr\text{-}val}}]$$

$$= \mathbb{E} \Bigg[ \left( \mathbf{I}_d - \left( (\mathbf{X}_t^{\mathsf{train}})^\top \mathbf{X}_t^{\mathsf{train}} + n_1 \lambda \mathbf{I}_d \right)^{-1} (\mathbf{X}_t^{\mathsf{train}})^\top \mathbf{X}_t^{\mathsf{train}} \right)^\top (\mathbf{X}_t^{\mathsf{val}})^\top \mathbf{X}_t^{\mathsf{val}}$$

$$\cdot \left( \mathbf{I}_d - \left( (\mathbf{X}^{\mathsf{train}})_t^\top \mathbf{X}_t^{\mathsf{train}} + n_1 \lambda \mathbf{I}_d \right)^{-1} (\mathbf{X}_t^{\mathsf{train}})^\top \mathbf{X}_t^{\mathsf{train}} \right) \Bigg]$$

$$\overset{(i)}{=} n_2 \mathbb{E} \Bigg[ \left( \mathbf{I}_d - \left( (\mathbf{X}_t^{\mathsf{train}})^\top \mathbf{X}_t^{\mathsf{train}} + n_1 \lambda \mathbf{I}_d \right)^{-1} ((\mathbf{X}_t^{\mathsf{train}})^\top \mathbf{X}_t^{\mathsf{train}}) \right)^\top$$

$$\cdot \left( \mathbf{I}_d - \left( (\mathbf{X}^{\mathsf{train}})_t^\top \mathbf{X}_t^{\mathsf{train}} + n_1 \lambda \mathbf{I}_d \right)^{-1} (\mathbf{X}_t^{\mathsf{train}})^\top \mathbf{X}_t^{\mathsf{train}} \right) \Bigg]$$

$$\overset{(ii)}{=} n_2 \mathbb{E} \Big[ \mathbf{V}_t^{\mathsf{train}} \left( \mathbf{I}_d - ((\mathbf{D}_t^{\mathsf{train}})^\top \mathbf{D}_t^{\mathsf{train}} + n_1 \lambda \mathbf{I}_d)^{-1} (\mathbf{D}_t^{\mathsf{train}})^\top \mathbf{D}_t^{\mathsf{train}} \right)^2 (\mathbf{V}_t^{\mathsf{train}})^\top \Big], \tag{28}$$

where $(i)$ uses the data generating assumption $\mathbb{E}[(\mathbf{X}_t^{\mathsf{val}})^\top \mathbf{X}_t^{\mathsf{val}}] = n_2 \mathbf{I}_d$ and the independence between $\mathbf{X}_t^{\mathsf{train}}$ and $\mathbf{X}_t^{\mathsf{val}}$, and $(ii)$ follows from the SVD of $\mathbf{X}_t^{\mathsf{train}} = \mathbf{U}_t^{\mathsf{train}} \mathbf{D}_t^{\mathsf{train}} (\mathbf{V}_t^{\mathsf{train}})^\top$.

Here we denote $\sigma_1^{(n_1)} \geq \cdots \geq \sigma_d^{(n_1)}$ as the eigenvalues of $\frac{1}{n_1} (\mathbf{X}_t^{\mathsf{train}})^\top \mathbf{X}_t^{\mathsf{train}}$. Thus, we have $(\mathbf{D}_t^{\mathsf{train}})^\top \mathbf{D}_t^{\mathsf{train}} = \operatorname{Diag}(n_1 \sigma_1^{(n_1)}, \dots, n_1 \sigma_d^{(n_1)})$. We can now further simplify (28) as

$$n_2 \mathbb{E} \Big[ \mathbf{V}_t^{\mathsf{train}} \left( \mathbf{I}_d - ((\mathbf{D}_t^{\mathsf{train}})^\top \mathbf{D}_t^{\mathsf{train}} + n_1 \lambda \mathbf{I}_d)^{-1} (\mathbf{D}_t^{\mathsf{train}})^\top \mathbf{D}_t^{\mathsf{train}} \right)^2 (\mathbf{V}_t^{\mathsf{train}})^\top \Big]$$

$$\overset{(i)}{=} n_2 \mathbb{E} \left[ \mathbf{V}_t^{\mathsf{train}} \operatorname{Diag} \left( \frac{\lambda^2}{(\sigma_1^{(n_1)} + \lambda)^2}, \dots, \frac{\lambda^2}{(\sigma_d^{(n_1)} + \lambda)^2} \right) (\mathbf{V}_t^{\mathsf{train}})^\top \right]$$

$$\overset{(ii)}{=} \frac{n_2}{d} \mathbb{E} \left[ \sum_{i=1}^{d} \frac{\lambda^2}{(\lambda + \sigma_i^{(n_1)})^2} \right] \mathbf{I}_d. \tag{29}$$

Step $(i)$ follows from the same computation in (22), and step $(ii)$ uses the permutation trick in (23).

Next, we find the expected covariance matrix $\frac{1}{n_2^2}\mathbb{E}[\nabla\ell_t^{\text{tr-tr}}(\mathbf{w}_{0,\star})(\nabla\ell_t^{\text{tr-tr}}(\mathbf{w}_{0,\star}))^\top]$.

$$
\begin{aligned}
&\mathbb{E}[\nabla\ell_t^{\text{tr-tr}}(\mathbf{w}_{0,\star})(\nabla\ell_t^{\text{tr-tr}}(\mathbf{w}_{0,\star}))^\top]\\
&= \mathbb{E}[(\mathbf{M}_t^{\text{tr-tr}})^\top\mathbf{M}_t^{\text{tr-tr}}(\mathbf{w}_{0,\star}-\mathbf{w}_t)(\mathbf{w}_{0,\star}-\mathbf{w}_t)^\top(\mathbf{M}_t^{\text{tr-tr}})^\top\mathbf{M}_t^{\text{tr-tr}}]\\
&= \mathbb{E}\bigg[\mathbf{V}_t^{\text{train}}\text{Diag}\left(\frac{\lambda}{\sigma_1^{(n_1)}+\lambda},\ldots,\frac{\lambda}{\sigma_d^{(n_1)}+\lambda}\right)(\mathbf{V}_t^{\text{train}})^\top(\mathbf{X}_t^{\text{val}})^\top\\
&\qquad\cdot\mathbf{X}_t^{\text{val}}\mathbf{V}_t^{\text{train}}\text{Diag}\left(\frac{\lambda}{\sigma_1^{(n_1)}+\lambda},\ldots,\frac{\lambda}{\sigma_d^{(n_1)}+\lambda}\right)(\mathbf{V}_t^{\text{train}})^\top(\mathbf{w}_{0,\star}-\mathbf{w}_t)(\mathbf{w}_{0,\star}-\mathbf{w}_t)^\top\\
&\qquad\cdot\mathbf{V}_t^{\text{train}}\text{Diag}\left(\frac{\lambda}{\sigma_1^{(n_1)}+\lambda},\ldots,\frac{\lambda}{\sigma_d^{(n_1)}+\lambda}\right)(\mathbf{V}_t^{\text{train}})^\top(\mathbf{X}_t^{\text{val}})^\top\\
&\qquad\cdot\mathbf{X}_t^{\text{val}}\mathbf{V}_t^{\text{train}}\text{Diag}\left(\frac{\lambda}{\sigma_1^{(n_1)}+\lambda},\ldots,\frac{\lambda}{\sigma_d^{(n_1)}+\lambda}\right)(\mathbf{V}_t^{\text{train}})^\top\bigg].
\end{aligned}
\tag{30}
$$

Combining (29) and (30), we derive the asymptotic covariance matrix of using $L^{\text{tr-val}}$ as

$$
\begin{aligned}
&\text{AsymCov}(\widehat{\mathbf{w}}_{0,T}^{\text{tr-val}})\\
&= \mathbb{E}[\nabla^{-2}\ell_t^{\text{tr-val}}]\text{Cov}[\nabla\ell_t^{\text{tr-val}}(\mathbf{w}_{0,\star})]\mathbb{E}[\nabla^{-2}\ell_t^{\text{tr-val}}]\\
&= \frac{d^2}{n_2^2}\left(\mathbb{E}\left[\sum_{i=1}^d\frac{\lambda^2}{(\lambda+\sigma_i^{(n_1)})^2}\right]\right)^{-2}\\
&\qquad\cdot\mathbb{E}\bigg[\mathbf{V}_t^{\text{train}}\text{Diag}\left(\frac{\lambda}{\sigma_1^{(n_1)}+\lambda},\ldots,\frac{\lambda}{\sigma_d^{(n_1)}+\lambda}\right)(\mathbf{V}_t^{\text{train}})^\top(\mathbf{X}_t^{\text{val}})^\top\\
&\qquad\cdot\mathbf{X}_t^{\text{val}}\mathbf{V}_t^{\text{train}}\text{Diag}\left(\frac{\lambda}{\sigma_1^{(n_1)}+\lambda},\ldots,\frac{\lambda}{\sigma_d^{(n_1)}+\lambda}\right)(\mathbf{V}_t^{\text{train}})^\top(\mathbf{w}_{0,\star}-\mathbf{w}_t)(\mathbf{w}_{0,\star}-\mathbf{w}_t)^\top\\
&\qquad\cdot\mathbf{V}_t^{\text{train}}\text{Diag}\left(\frac{\lambda}{\sigma_1^{(n_1)}+\lambda},\ldots,\frac{\lambda}{\sigma_d^{(n_1)}+\lambda}\right)(\mathbf{V}_t^{\text{train}})^\top(\mathbf{X}_t^{\text{val}})^\top\\
&\qquad\cdot\mathbf{X}_t^{\text{val}}\mathbf{V}_t^{\text{train}}\text{Diag}\left(\frac{\lambda}{\sigma_1^{(n_1)}+\lambda},\ldots,\frac{\lambda}{\sigma_d^{(n_1)}+\lambda}\right)(\mathbf{V}_t^{\text{train}})^\top\bigg].
\end{aligned}
\tag{31}
$$

Taking trace in (31), we deduce

$$
\begin{aligned}
&\text{AsymMSE}(\widehat{\mathbf{w}}_{0,T}^{\text{tr-tr}})\\
&= \text{tr}(\text{AsymCov}(\widehat{\mathbf{w}}_{0,T}^{\text{tr-tr}}))\\
&= \frac{d^2}{n_2^2}\left(\mathbb{E}\left[\sum_{i=1}^d\frac{\lambda^2}{(\lambda+\sigma_i^{(n_1)})^2}\right]\right)^{-2}\\
&\qquad\cdot\text{tr}\bigg(\mathbb{E}\bigg[\mathbf{V}_t^{\text{train}}\text{Diag}\left(\frac{\lambda}{\sigma_1^{(n_1)}+\lambda},\ldots,\frac{\lambda}{\sigma_d^{(n_1)}+\lambda}\right)(\mathbf{V}_t^{\text{train}})^\top(\mathbf{X}_t^{\text{val}})^\top\\
&\qquad\cdot\mathbf{X}_t^{\text{val}}\mathbf{V}_t^{\text{train}}\text{Diag}\left(\frac{\lambda}{\sigma_1^{(n_1)}+\lambda},\ldots,\frac{\lambda}{\sigma_d^{(n_1)}+\lambda}\right)(\mathbf{V}_t^{\text{train}})^\top(\mathbf{w}_{0,\star}-\mathbf{w}_t)(\mathbf{w}_{0,\star}-\mathbf{w}_t)^\top\\
&\qquad\cdot\mathbf{V}_t^{\text{train}}\text{Diag}\left(\frac{\lambda}{\sigma_1^{(n_1)}+\lambda},\ldots,\frac{\lambda}{\sigma_d^{(n_1)}+\lambda}\right)(\mathbf{V}_t^{\text{train}})^\top(\mathbf{X}_t^{\text{val}})^\top\\
&\qquad\cdot\mathbf{X}_t^{\text{val}}\mathbf{V}_t^{\text{train}}\text{Diag}\left(\frac{\lambda}{\sigma_1^{(n_1)}+\lambda},\ldots,\frac{\lambda}{\sigma_d^{(n_1)}+\lambda}\right)(\mathbf{V}_t^{\text{train}})^\top\bigg]\bigg)
\end{aligned}
$$

$$
\begin{aligned}
\overset{(i)}{=} \frac{d^2}{n_2^2} &\left( \mathbb{E}\left[ \sum_{i=1}^{d} \frac{\lambda^2}{(\lambda + \sigma_i^{(n)})^2} \right] \right)^{-2} \\
&\cdot \operatorname{tr}\Bigg( \mathbb{E}\bigg[ \mathbf{V}_t^{\text{train}}\operatorname{Diag}\left( \frac{\lambda}{\sigma_1^{(n_1)} + \lambda}, \dots, \frac{\lambda}{\sigma_d^{(n_1)} + \lambda} \right)(\mathbf{V}_t^{\text{train}})^\top (\mathbf{X}_t^{\text{val}})^\top \\
&\quad\cdot \mathbf{X}_t^{\text{val}}\mathbf{V}_t^{\text{train}}\operatorname{Diag}\left( \frac{\lambda^2}{(\sigma_1^{(n_1)} + \lambda)^2}, \dots, \frac{\lambda^2}{(\sigma_d^{(n_1)} + \lambda)^2} \right)(\mathbf{V}_t^{\text{train}})^\top (\mathbf{X}_t^{\text{val}})^\top \\
&\quad\cdot \mathbf{X}_t^{\text{val}}\mathbf{V}_t^{\text{train}}\operatorname{Diag}\left( \frac{\lambda}{\sigma_1^{(n_1)} + \lambda}, \dots, \frac{\lambda}{\sigma_d^{(n_1)} + \lambda} \right)(\mathbf{V}_t^{\text{train}})^\top (\mathbf{w}_{0,\star} - \mathbf{w}_t)(\mathbf{w}_{0,\star} - \mathbf{w}_t)^\top \bigg] \Bigg),
\end{aligned}
\tag{32}
$$

where $(i)$ follows from the cyclic property of the matrix trace operation. Due to the isotropicity of $\mathbf{X}_t^{\text{train}}$ and $\mathbf{X}_t^{\text{val}}$, we claim that

$$
\begin{aligned}
\mathbb{E}\bigg[ &\mathbf{V}_t^{\text{train}}\operatorname{Diag}\left( \frac{\lambda}{\sigma_1^{(n_1)} + \lambda}, \dots, \frac{\lambda}{\sigma_d^{(n_1)} + \lambda} \right)(\mathbf{V}_t^{\text{train}})^\top (\mathbf{X}_t^{\text{val}})^\top \\
&\cdot \mathbf{X}_t^{\text{val}}\mathbf{V}_t^{\text{train}}\operatorname{Diag}\left( \frac{\lambda^2}{(\sigma_1^{(n_1)} + \lambda)^2}, \dots, \frac{\lambda^2}{(\sigma_d^{(n_1)} + \lambda)^2} \right)(\mathbf{V}_t^{\text{train}})^\top (\mathbf{X}_t^{\text{val}})^\top \\
&\cdot \mathbf{X}_t^{\text{val}}\mathbf{V}_t^{\text{train}}\operatorname{Diag}\left( \frac{\lambda}{\sigma_1^{(n_1)} + \lambda}, \dots, \frac{\lambda}{\sigma_d^{(n_1)} + \lambda} \right)(\mathbf{V}_t^{\text{train}})^\top \bigg]
\end{aligned}
\tag{33}
$$

is a diagonal matrix $c\mathbf{I}_d$ with all the diagonal elements identical. We can show the claim bying taking expectation with respect to $\mathbf{X}_t^{\text{val}}$ first. Since $\mathbf{V}_t^{\text{train}}$ is an orthogonal matrix, $\mathbf{X}_t^{\text{val}}\mathbf{V}_t^{\text{train}}$ has the same distribution as $\mathbf{X}_t^{\text{val}}$ and independent of $\mathbf{X}_t$. We verify that any off-diagonal element of the matrix expectation

$$
\begin{aligned}
\mathbf{A} := \ \mathbb{E}_{\mathbf{X}_t^{\text{val}}}\bigg[ &(\mathbf{V}_t^{\text{train}})^\top (\mathbf{X}_t^{\text{val}})^\top \mathbf{X}_t^{\text{val}}\mathbf{V}_t^{\text{train}}\operatorname{Diag}\left( \frac{\lambda^2}{(\sigma_1^{(n_1)} + \lambda)^2}, \dots, \frac{\lambda^2}{(\sigma_d^{(n_1)} + \lambda)^2} \right) \\
&\cdot (\mathbf{V}_t^{\text{train}})^\top (\mathbf{X}_t^{\text{val}})^\top \mathbf{X}_t^{\text{val}}\mathbf{V}_t^{\text{train}} \bigg]
\end{aligned}
$$

is zero. We denote $\mathbf{X}_t^{\text{val}}\mathbf{V}_t^{\text{train}} = [\mathbf{x}_1, \dots, \mathbf{x}_n]^\top \in \mathbb{R}^{n_2 \times d}$ with $\mathbf{x}_i \overset{\text{iid}}{\sim} \mathsf{N}(\mathbf{0}, \mathbf{I}_d)$. For $k \neq \ell$, the $(k, \ell)$-th entry $A_{k,\ell}$ of $\mathbf{A}$ is

$$
\begin{aligned}
A_{k,\ell} &= \mathbb{E}\left[ \sum_j \left( \frac{\lambda^2}{(\sigma_j^{(n_1)} + \lambda)^2} \left( \sum_i x_{k,i}x_{j,i} \right) \left( \sum_i x_{j,i}x_{\ell,i} \right) \right) \right] \\
&= \mathbb{E}\left[ \sum_j \frac{\lambda^2}{(\sigma_j^{(n_1)} + \lambda)^2} \left( \sum_{m,n} x_{k,m}x_{j,m}x_{j,n}x_{\ell,n} \right) \right] \\
&\overset{(i)}{=} 0,
\end{aligned}
$$

where $x_{i,j}$ denotes the $j$-th element of $\mathbf{x}_i$. Equality $(i)$ holds, since either $x_{k,m}$ or $x_{\ell,n}$ only appears once in each summand. Therefore, we can write $\mathbf{A} = \operatorname{Diag}(A_{1,1}, \dots, A_{d,d})$ with $A_{k,k}$ being

$$
\begin{aligned}
A_{k,k} &= \mathbb{E}\left[ \sum_j \frac{\lambda^2}{(\sigma_j^{(n_1)} + \lambda)^2} \left( \sum_{m,n} x_{k,m}x_{j,m}x_{j,n}x_{\ell,n} \right) \right] \\
&= \mathbb{E}\left[ \frac{\lambda^2}{(\sigma_k^{(n_1)} + \lambda)^2} \left( \sum_{m,n} x_{k,m}x_{k,m}x_{k,n}x_{k,n} \right) \right].
\end{aligned}
$$

Observe that $A_{k,k}$ only depends on $\sigma_k^{(n_1)}$. Plugging back into (33), we have

$$
\mathbb{E}\left[\mathbf{V}_t^{\text{train}}\text{Diag}\left(\frac{\lambda}{\sigma_1^{(n_1)}+\lambda},\ldots,\frac{\lambda}{\sigma_d^{(n_1)}+\lambda}\right)(\mathbf{V}_t^{\text{train}})^\top(\mathbf{X}_t^{\text{val}})^\top\right.
$$
$$
\cdot\mathbf{X}_t^{\text{val}}\mathbf{V}_t^{\text{train}}\text{Diag}\left(\frac{\lambda^2}{(\sigma_1^{(n_1)}+\lambda)^2},\ldots,\frac{\lambda^2}{(\sigma_d^{(n_1)}+\lambda)^2}\right)(\mathbf{V}_t^{\text{train}})^\top(\mathbf{X}_t^{\text{val}})^\top
$$
$$
\left.\cdot\mathbf{X}_t^{\text{val}}\mathbf{V}_t^{\text{train}}\text{Diag}\left(\frac{\lambda}{\sigma_1^{(n_1)}+\lambda},\ldots,\frac{\lambda}{\sigma_d^{(n_1)}+\lambda}\right)(\mathbf{V}_t^{\text{train}})^\top\right]
$$
$$
=\mathbb{E}\left[\mathbf{V}_t^{\text{train}}\text{Diag}\left(\frac{\lambda}{\sigma_1^{(n_1)}+\lambda},\ldots,\frac{\lambda}{\sigma_d^{(n_1)}+\lambda}\right)\text{Diag}(A_{1,1},\ldots,A_{d,d})\right.
$$
$$
\left.\cdot\text{Diag}\left(\frac{\lambda}{\sigma_1^{(n_1)}+\lambda},\ldots,\frac{\lambda}{\sigma_d^{(n_1)}+\lambda}\right)(\mathbf{V}_t^{\text{train}})^\top\right]
$$
$$
=\mathbb{E}\left[\mathbf{V}_t^{\text{train}}\text{Diag}\left(\frac{\lambda^2 A_{1,1}}{(\sigma_1^{(n_1)}+\lambda)^2},\ldots,\frac{\lambda^2 A_{d,d}}{(\sigma_d^{(n_1)}+\lambda)^2}\right)(\mathbf{V}_t^{\text{train}})^\top\right]
$$
$$
\overset{(i)}{=}c\mathbf{I}_d,
$$

where equality $(i)$ utilizes the permutation trick in (24). To this end, it is sufficient to find $c$ as

$$
c=\frac{1}{d}\text{tr}\left(\mathbb{E}\left[\mathbf{V}_t^{\text{train}}\text{Diag}\left(\frac{\lambda}{\sigma_1^{(n_1)}+\lambda},\ldots,\frac{\lambda}{\sigma_d^{(n_1)}+\lambda}\right)(\mathbf{V}_t^{\text{train}})^\top(\mathbf{X}_t^{\text{val}})^\top\right.\right.
$$
$$
\cdot\mathbf{X}_t^{\text{val}}\mathbf{V}_t^{\text{train}}\text{Diag}\left(\frac{\lambda^2}{(\sigma_1^{(n_1)}+\lambda)^2},\ldots,\frac{\lambda^2}{(\sigma_d^{(n_1)}+\lambda)^2}\right)(\mathbf{V}_t^{\text{train}})^\top(\mathbf{X}_t^{\text{val}})^\top
$$
$$
\left.\left.\cdot\mathbf{X}_t^{\text{val}}\mathbf{V}_t^{\text{train}}\text{Diag}\left(\frac{\lambda}{\sigma_1^{(n_1)}+\lambda},\ldots,\frac{\lambda}{\sigma_d^{(n_1)}+\lambda}\right)(\mathbf{V}_t^{\text{train}})^\top\right]\right)
$$
$$
=\frac{1}{d}\text{tr}\left(\mathbb{E}\left[\mathbf{X}_t^{\text{val}}\mathbf{V}_t^{\text{train}}\text{Diag}\left(\frac{\lambda^2}{(\sigma_1^{(n_1)}+\lambda)^2},\ldots,\frac{\lambda^2}{(\sigma_d^{(n_1)}+\lambda)^2}\right)(\mathbf{V}_t^{\text{train}})^\top(\mathbf{X}_t^{\text{val}})^\top\right.\right.
$$
$$
\left.\left.\cdot\mathbf{X}_t^{\text{val}}\mathbf{V}_t^{\text{train}}\text{Diag}\left(\frac{\lambda^2}{(\sigma_1^{(n_1)}+\lambda)^2},\ldots,\frac{\lambda^2}{(\sigma_d^{(n_1)}+\lambda)^2}\right)(\mathbf{V}_t^{\text{train}})^\top(\mathbf{X}_t^{\text{val}})^\top\right]\right). \qquad (34)
$$

Observe again that $\mathbf{X}_t^{\text{val}}\mathbf{V}_t^{\text{train}}\in\mathbb{R}^{n_2\times d}$ is a Gaussian random matrix. We rewrite (34) as

$$
c=\frac{1}{d}\mathbb{E}\left[\left(\sum_{i,j=1}^{n_2}\mathbf{v}_i^\top\text{Diag}\left(\frac{\lambda^2}{(\sigma_1^{(n_1)}+\lambda)^2},\ldots,\frac{\lambda^2}{(\sigma_d^{(n_1)}+\lambda)^2}\right)\mathbf{v}_j\right)^2\right], \qquad (35)
$$

where $\mathbf{v}_i\overset{\text{iid}}{\sim}\mathsf{N}(\mathbf{0},\mathbf{I}_d)$ is i.i.d. Gaussian random vectors for $i=1,\ldots,n_2$. To compute (35), we need the following result.

**Claim 7.** *Given any symmetric matrix $\mathbf{A}\in\mathbb{R}^{d\times d}$ and i.i.d. standard Gaussian random vectors $\mathbf{v},\mathbf{u}\overset{\text{iid}}{\sim}\mathsf{N}(\mathbf{0},\mathbf{I}_d)$, we have*

$$
\mathbb{E}\left[(\mathbf{v}^\top\mathbf{A}\mathbf{v})^2\right]=2\|\mathbf{A}\|_{\mathsf{Fr}}^2+\text{tr}^2(\mathbf{A})\quad\text{and} \qquad (36)
$$
$$
\mathbb{E}\left[(\mathbf{v}^\top\mathbf{A}\mathbf{u})^2\right]=\|\mathbf{A}\|_{\mathsf{Fr}}^2. \qquad (37)
$$

*Proof of Claim 7.* We show (36) first. We denote $A_{i,j}$ as the $(i,j)$-th element of $\mathbf{A}$ and $v_i$ as the $i$-th element of $\mathbf{v}$. Expanding the quadratic form, we have

$$
\mathbb{E}\left[(\mathbf{v}^\top\mathbf{A}\mathbf{v})^2\right]=\mathbb{E}\left[\sum_{i,j,k,\ell\leq d}v_iv_jv_kv_\ell A_{i,j}A_{k,\ell}\right]
$$

$$= \mathbb{E}\left[\sum_{i\leq d} v_i^4 A_{i,i}^2\right] + \mathbb{E}\left[\sum_{i\neq j} v_i^2 v_j^2 (A_{i,j}^2 + A_{i,i}A_{j,j} + A_{i,j}A_{j,i})\right]$$

$$= 3\sum_{i\leq d} A_{i,i}^2 + \sum_{i\neq j}(A_{i,j}^2 + A_{i,i}A_{j,j} + A_{i,j}A_{j,i})$$

$$= \operatorname{tr}^2(\mathbf{A}) + 2\sum_{i\leq d} A_{i,i}^2 + \sum_{i\neq j}(A_{i,j}^2 + A_{i,j}A_{j,i})$$

$$= \operatorname{tr}^2(\mathbf{A}) + 2\|\mathbf{A}\|_{\mathsf{Fr}}^2.$$

Next, we show (37) by the cyclic property of race.

$$\mathbb{E}\left[(\mathbf{v}^\top \mathbf{A}\mathbf{u})^2\right] = \operatorname{tr}\left(\mathbb{E}\left[\mathbf{u}\mathbf{u}^\top \mathbf{A}\mathbf{v}\mathbf{v}^\top \mathbf{A}\right]\right) = \operatorname{tr}(\mathbf{A}^2) = \|\mathbf{A}\|_{\mathsf{Fr}}^2.$$

$$\square$$

We back to the computation of (35) using Claim 7.

$$c = \frac{1}{d}\mathbb{E}\left[\sum_{i,j=1}^{n_2}\left(\mathbf{v}_i^\top \operatorname{Diag}\left(\frac{\lambda^2}{(\sigma_1^{(n_1)}+\lambda)^2},\dots,\frac{\lambda^2}{(\sigma_d^{(n_1)}+\lambda)^2}\right)\mathbf{v}_j\right)^2\right]$$

$$= \frac{1}{d}\mathbb{E}\left[\sum_{i=1}^{n_2}\left(\mathbf{v}_i^\top \operatorname{Diag}\left(\frac{\lambda^2}{(\sigma_1^{(n_1)}+\lambda)^2},\dots,\frac{\lambda^2}{(\sigma_d^{(n_1)}+\lambda)^2}\right)\mathbf{v}_i\right)^2\right]$$

$$+ \frac{1}{d}\mathbb{E}\left[\sum_{i\neq j}\left(\mathbf{v}_i^\top \operatorname{Diag}\left(\frac{\lambda^2}{(\sigma_1^{(n_1)}+\lambda)^2},\dots,\frac{\lambda^2}{(\sigma_d^{(n_1)}+\lambda)^2}\right)\mathbf{v}_j\right)^2\right]$$

$$= \frac{n_2}{d}\mathbb{E}\left[\operatorname{tr}^2\left(\operatorname{Diag}\left(\frac{\lambda^2}{(\sigma_1^{(n_1)}+\lambda)^2},\dots,\frac{\lambda^2}{(\sigma_d^{(n_1)}+\lambda)^2}\right)\right)\right]$$

$$+ 2\frac{n_2}{d}\mathbb{E}\left[\left\|\operatorname{Diag}\left(\frac{\lambda^2}{(\sigma_1^{(n_1)}+\lambda)^2},\dots,\frac{\lambda^2}{(\sigma_d^{(n_1)}+\lambda)^2}\right)\right\|_{\mathsf{Fr}}^2\right]$$

$$+ \frac{n_2(n_2-1)}{d}\mathbb{E}\left[\left\|\operatorname{Diag}\left(\frac{\lambda^2}{(\sigma_1^{(n_1)}+\lambda)^2},\dots,\frac{\lambda^2}{(\sigma_d^{(n_1)}+\lambda)^2}\right)\right\|_{\mathsf{Fr}}^2\right]$$

$$= \frac{n_2}{d}\left(\mathbb{E}\left[\sum_{i=1}^d \frac{\lambda^2}{(\sigma_i^{(n_1)}+\lambda)^2}\right]^2 + (n_2+1)\mathbb{E}\left[\sum_{i=1}^d \frac{\lambda^4}{(\sigma_i^{(n_1)}+\lambda)^4}\right]\right). \tag{38}$$

Combining (38) and (33), by the independence between $\mathbf{w}_t$ and $\mathbf{X}_t^{\mathsf{train}}, \mathbf{X}_t^{\mathsf{val}}$, we compute (32) as

$$\operatorname{AsymMSE}(\widehat{\mathbf{w}}_{0,T}^{\mathsf{tr\text{-}tr}})$$

$$= \frac{d^2}{n_2^2}\left(\mathbb{E}\left[\sum_{i=1}^d \frac{\lambda^2}{(\lambda+\sigma_i^{(n_1)})^2}\right]\right)^{-2}$$

$$\cdot \frac{n_2}{d}\left(\mathbb{E}\left[\sum_{i=1}^d \frac{\lambda^2}{(\sigma_i^{(n_1)}+\lambda)^2}\right]^2 + (n_2+1)\mathbb{E}\left[\sum_{i=1}^d \frac{\lambda^4}{(\sigma_i^{(n_1)}+\lambda)^4}\right]\right)$$

$$\cdot \mathbb{E}\left[(\mathbf{w}_{0,\star}-\mathbf{w}_t)(\mathbf{w}_{0,\star}-\mathbf{w}_t)^\top\right]$$

$$= \frac{dR^2}{n_2}\frac{\mathbb{E}\left[\sum_{i=1}^d \lambda^2/(\sigma_i^{(n_1)}+\lambda)^2\right]^2 + (n_2+1)\mathbb{E}\left[\sum_{i=1}^d \lambda^4/(\sigma_i^{(n_1)}+\lambda)^4\right]}{\left(\mathbb{E}\left[\sum_{i=1}^d \lambda^2/(\lambda+\sigma_i^{(n_1)})^2\right]\right)^2}.$$

The proof is complete. $\square$

### B.3 Optimal rate of the train-val method at finite $(n, d)$

**Corollary 8** (Optimal rate of the train-val method at finite $(n, d)$). *For any $(n, d)$ and any split ratio $(n_1, n_2) = (n_1, n - n_1)$, the optimal rate (by tuning the regularization $\lambda > 0$) of the train-val method is achieved at*

$$\inf_{\lambda > 0} \text{AsymMSE}\big(\widehat{\mathbf{w}}_{0,T}^{\text{tr-val}}(n_1, n_2; \lambda)\big) = \lim_{\lambda \to \infty} \text{AsymMSE}\big(\widehat{\mathbf{w}}_{0,T}^{\text{tr-val}}(n_1, n_2; \lambda)\big) = \frac{(d + n_2 + 1)R^2}{n_2}.$$

*Further optimizing the rate over $n_2$, the best rate is taken at $(n_1, n_2) = (0, n)$, in which the rate is*

$$\inf_{\lambda > 0,\ n_2 \in [n]} \text{AsymMSE}\big(\widehat{\mathbf{w}}_{0,T}^{\text{tr-val}}(n_1, n_2; \lambda)\big) = \frac{(d + n + 1)R^2}{n}.$$

**Discussion: Using all data as validation**   Corollary 8 suggests that the optimal asymptotic rate of the train-val method is obtained at $\lambda = \infty$ and $(n_1, n_2) = (0, n)$. In other words, the optimal choice for the train-val method is to *use all the data as validation*. In this case, since there is no training data, the inner solver reduces to the identity map: $\mathcal{A}_{\infty,0}(\mathbf{w}_0; \mathbf{X}_t, \mathbf{y}_t) = \mathbf{w}_0$, and the outer loop reduces to learning a single linear model $\mathbf{w}_0$ on all the tasks combined. We remark that while the optimality of such a split ratio is likely an artifact of the data distribution we assumed (noiseless realizable linear model) and may not generalize to other meta-learning problems, we do find experimentally that using more data as validation (than training) can also improve the performance on real meta-learning tasks (see Table 2).

**Proof of Corollary 8**   Fix $n_1 \in [n]$ and $n_2 = n - n_1$. Recall from Lemma 5 that

$$\text{AsymMSE}\big(\widehat{\mathbf{w}}_{0,T}^{\text{tr-val}}(n_1, n_2; \lambda)\big) = \frac{dR^2}{n_2} \cdot \frac{\mathbb{E}\left[\left(\sum_{i=1}^d \lambda^2/(\sigma_i^{(n_1)} + \lambda)^2\right)^2 + (n_2 + 1)\sum_{i=1}^d \lambda^4/(\sigma_i^{(n_1)} + \lambda)^4\right]}{\left(\mathbb{E}\left[\sum_{i=1}^d \lambda^2/(\sigma_i^{(n_1)} + \lambda)^2\right]\right)^2}.$$

Clearly, as $\lambda \to \infty$, we have

$$\lim_{\lambda \to \infty} \text{AsymMSE}\big(\widehat{\mathbf{w}}_{0,T}^{\text{tr-val}}(n_1, n_2; \lambda)\big) = \frac{dR^2}{n_2} \cdot \frac{d^2 + (n_2 + 1)d}{d^2} = \frac{(d + n_2 + 1)R^2}{n_2}.$$

It remains to show that the above quantity is a lower bound for $\text{AsymMSE}\big(\widehat{\mathbf{w}}_{0,T}^{\text{tr-val}}(n_1, n_2; \lambda)\big)$ for any $\lambda > 0$, which is equivalent to

$$\frac{\mathbb{E}\left[\left(\sum_{i=1}^d \lambda^2/(\sigma_i^{(n_1)} + \lambda)^2\right)^2 + (n_2 + 1)\sum_{i=1}^d \lambda^4/(\sigma_i^{(n_1)} + \lambda)^4\right]}{\left(\mathbb{E}\left[\sum_{i=1}^d \lambda^2/(\sigma_i^{(n_1)} + \lambda)^2\right]\right)^2} \geq \frac{d + n_2 + 1}{d}, \quad \text{for all } \lambda > 0. \tag{39}$$

We now prove (39). For $i \in [n_1]$, define random variables

$$X_i := \frac{\lambda^2}{(\sigma_i^{(n_1)} + \lambda)^2} \in [0, 1] \quad \text{and} \quad Y_i := 1 - X_i \in [0, 1].$$

Then the left-hand side of (39) can be rewritten as

$$\frac{\mathbb{E}\left[\left(d - n_1 + \sum_{i=1}^{n_1} X_i\right)^2 + (n_2 + 1)\left(d - n_1 + \sum_{i=1}^{n_1} X_i^2\right)\right]}{\left(\mathbb{E}[d - n_1 + \sum_{i=1}^n X_i]\right)^2}$$

$$= \frac{\mathbb{E}\left[\left(d - \sum_{i=1}^{n_1} Y_i\right)^2 + (n_2 + 1)\left(d - 2\sum_{i=1}^{n_1} Y_i + \sum_{i=1}^{n_1} Y_i^2\right)\right]}{\left(\mathbb{E}[d - \sum_{i=1}^{n_1} Y_i]\right)^2}$$

$$= \frac{d^2 + (n_2 + 1)d - 2(d + n_2 + 1)\mathbb{E}[\sum Y_i] + \mathbb{E}\left[\left(\sum Y_i\right)^2\right] + (n_2 + 1)\mathbb{E}\left[\sum Y_i^2\right]}{d^2 - 2d\mathbb{E}[\sum Y_i] + \left(\mathbb{E}[\sum Y_i]\right)^2}$$

By algebraic manipulation, inequality (39) is equivalent to showing that

$$\frac{\mathbb{E}\big[(\sum Y_i)^2\big] + (n_2+1)\mathbb{E}\big[\sum Y_i^2\big]}{\big(\mathbb{E}[\sum Y_i]\big)^2} \geq \frac{d+n_2+1}{d}. \tag{40}$$

Clearly, $\mathbb{E}[(\sum Y_i)^2] \geq (\mathbb{E}[\sum Y_i])^2$. By Cauchy-Schwarz we also have

$$\mathbb{E}\Big[\sum Y_i^2\Big] \geq \frac{1}{n_1}\mathbb{E}\Big[\Big(\sum Y_i\Big)^2\Big] \geq \frac{1}{n_1}\Big(\mathbb{E}\Big[\sum Y_i\Big]\Big)^2.$$

Therefore we have

$$\frac{\mathbb{E}\big[(\sum Y_i)^2\big] + (n_2+1)\mathbb{E}\big[\sum Y_i^2\big]}{\big(\mathbb{E}[\sum Y_i]\big)^2} \geq 1 + \frac{n_2+1}{n_1} \geq 1 + \frac{n_2+1}{d} = \frac{d+n_2+1}{d},$$

where we have used that $n_1 \leq n \leq d$. This shows (40) and consequently (39). $\qquad\square$

### B.4 RATE OF THE TRAIN-TRAIN METHOD IN THE PROPORTIONAL LIMIT

**Theorem 9** (Exact rates of the train-train method in the proportional limit). *In the high-dimensional limiting regime $d, n \to \infty$, $d/n \to \gamma$ where $\gamma \in (0, \infty)$ is a fixed shape parameter, for any $\lambda > 0$*

$$\lim_{d,n\to\infty,d/n=\gamma} \mathrm{AsymMSE}\big(\widehat{\mathbf{w}}_{0,T}^{\mathsf{tr\text{-}tr}}(n;\lambda)\big) = \rho_{\lambda,\gamma}R^2.$$

*where $\rho_{\lambda,\gamma} = 4\gamma^2\big[(\gamma-1)^2 + (\gamma+1)\lambda\big]/(\lambda+1+\gamma-\sqrt{(\lambda+\gamma+1)^2-4\gamma})^2/\big((\lambda+\gamma+1)^2-4\gamma\big)^{3/2}$.*

**Proof of Theorem 9** Let $\widehat{\mathbf{\Sigma}}_n := \frac{1}{n}\mathbf{X}_t\mathbf{X}_t^\top$ denote the sample covariance matrix of the inputs in a single task ($t$). By Lemma 5, we have

$$\begin{aligned}
\mathrm{AsymMSE}(\widehat{\mathbf{w}}_{0,T}^{\mathsf{tr\text{-}tr}}(n;\lambda)) &= R^2 \cdot \frac{\frac{1}{d}\mathbb{E}\Big[\sum_{i=1}^d \sigma_i(\widehat{\mathbf{\Sigma}}_n)^2/(\sigma_i(\widehat{\mathbf{\Sigma}}_n)+\lambda)^4\Big]}{\Big(\frac{1}{d}\mathbb{E}\Big[\sum_{i=1}^d \sigma_i(\widehat{\mathbf{\Sigma}}_n)/(\sigma_i(\widehat{\mathbf{\Sigma}}_n)+\lambda)^2\Big]\Big)^2} \\
&= R^2 \cdot \underbrace{\frac{1}{d}\mathbb{E}\Big[\mathrm{tr}\Big((\widehat{\mathbf{\Sigma}}_n+\lambda\mathbf{I}_d)^{-4}\widehat{\mathbf{\Sigma}}_n^2\Big)\Big]}_{\mathrm{I}_{n,d}} \Big/ \Big\{\underbrace{\frac{1}{d}\mathbb{E}\Big[\mathrm{tr}\Big((\widehat{\mathbf{\Sigma}}_n+\lambda\mathbf{I}_d)^{-2}\widehat{\mathbf{\Sigma}}_n\Big)\Big]}_{\mathrm{II}_{n,d}}\Big\}^2.
\end{aligned} \tag{41}$$

We now evaluate quantities $\mathrm{I}_{n,d}$ and $\mathrm{II}_{n,d}$ in the high-dimensional limit of $d, n \to \infty$, $d/n \to \gamma \in (0,\infty)$. Consider the (slightly generalized) Stieltjes transform of $\widehat{\mathbf{\Sigma}}_n$ defined for all $\lambda_1, \lambda_2 > 0$:

$$s(\lambda_1,\lambda_2) := \lim_{d,n\to\infty,\ d/n\to\gamma} \frac{1}{d}\mathbb{E}\Big[\mathrm{tr}\Big((\lambda_1\mathbf{I}_d+\lambda_2\widehat{\mathbf{\Sigma}}_n)^{-1}\Big)\Big].$$

As the entries of $\mathbf{X}_t$ are i.i.d. $\mathsf{N}(0,1)$, the above limiting Stieltjes transform is the Stieltjes form of the Marchenko-Pastur law, which has a closed form (see, e.g. (Dobriban et al., 2018, Equation (7)))

$$\begin{aligned}
s(\lambda_1,\lambda_2) &= \lambda_2^{-1}s(\lambda_1/\lambda_2,1) = \frac{1}{\lambda_2} \cdot \frac{\gamma-1-\lambda_1/\lambda_2+\sqrt{(\lambda_1/\lambda_2+1+\gamma)^2-4\gamma}}{2\gamma\lambda_1/\lambda_2} \\
&= \frac{\gamma-1-\lambda_1/\lambda_2+\sqrt{(\lambda_1/\lambda_2+1+\gamma)^2-4\gamma}}{2\gamma\lambda_1}.
\end{aligned} \tag{42}$$

Now observe that differentiating $s(\lambda_1,\lambda_2)$ yields quantity II (known as the derivative trick of Stieltjes transforms). Indeed, we have

$$\begin{aligned}
-\frac{d}{d\lambda_2}s(\lambda_1,\lambda_2) &= -\frac{d}{d\lambda_2}\lim_{d,n\to\infty,\ d/n\to\gamma}\frac{1}{d}\mathbb{E}\Big[\mathrm{tr}\Big((\lambda_1\mathbf{I}_d+\lambda_2\widehat{\mathbf{\Sigma}}_n)^{-1}\Big)\Big] \\
&= \lim_{d,n\to\infty,\ d/n\to\gamma}\frac{1}{d}\mathbb{E}\Big[-\frac{d}{d\lambda_2}\mathrm{tr}\Big((\lambda_1\mathbf{I}_d+\lambda_2\widehat{\mathbf{\Sigma}}_n)^{-1}\Big)\Big] \\
&= \lim_{d,n\to\infty,\ d/n\to\gamma}\frac{1}{d}\mathbb{E}\Big[\mathrm{tr}\Big((\lambda_1\mathbf{I}_d+\lambda_2\widehat{\mathbf{\Sigma}}_n)^{-2}\widehat{\mathbf{\Sigma}}_n\Big)\Big].
\end{aligned} \tag{43}$$

(Above, the exchange of differentiation and limit is due to the uniform convergence of the derivatives, which holds at any $\lambda_1, \lambda_2 > 0$. See Appendix B.4.1 for a detailed justification.) Taking $\lambda_1 = \lambda$ and $\lambda_2 = 1$, we get

$$\lim_{d,n\to\infty,\ d/n\to\gamma} \mathrm{II}_{n,d} = \lim_{d,n\to\infty,\ d/n\to\gamma} \frac{1}{d}\mathbb{E}\Big[\mathrm{tr}\Big((\lambda\mathbf{I}_d + \widehat{\boldsymbol{\Sigma}}_n)^{-2}\widehat{\boldsymbol{\Sigma}}_n\Big)\Big] = -\frac{d}{d\lambda_2}s(\lambda_1,\lambda_2)|_{\lambda_1=\lambda,\lambda_2=1}.$$

Similarly we have

$$\lim_{d,n\to\infty,\ d/n\to\gamma} \mathrm{I}_{n,d} = \lim_{d,n\to\infty,d/n\to\gamma} \frac{1}{d}\mathbb{E}\Big[\mathrm{tr}\Big((\lambda\mathbf{I}_d + \widehat{\boldsymbol{\Sigma}}_n)^{-4}\widehat{\boldsymbol{\Sigma}}_n^2\Big)\Big] = -\frac{1}{6}\frac{d}{d\lambda_1}\frac{d^2}{d\lambda_2^2}s(\lambda_1,\lambda_2)|_{\lambda_1=\lambda,\lambda_2=1}.$$

Evaluating the right-hand sides from differentiating the closed-form expression (42), we get

$$\lim_{d,n\to\infty,\ d/n\to\gamma} \mathrm{I}_{n,d} = \frac{1}{2\gamma}\cdot\frac{\lambda+1+\gamma}{\sqrt{(\lambda+1+\gamma)^2-4\gamma}} - \frac{1}{2\gamma},$$

$$\lim_{d,n\to\infty,\ d/n\to\gamma} \mathrm{II}_{n,d} = \frac{(\gamma-1)^2+(\gamma+1)\lambda}{((\lambda+1+\gamma)^2-4\gamma)^{5/2}}.$$

Substituting back to (41) yields that

$$\lim_{d,n\to\infty,\ d/n\to\gamma} \mathrm{AsymMSE}(\widehat{\mathbf{w}}_{0,T}^{\mathrm{tr\text{-}tr}}(n;\lambda)) = \lim_{d,n\to\infty,\ d/n\to\gamma} R^2\cdot\mathrm{I}_{n,d}/\mathrm{II}_{n,d}^2$$

$$= R^2\cdot\frac{4\gamma^2\big[(\gamma-1)^2+(\gamma+1)\lambda\big]}{((\lambda+1+\gamma)^2-4\gamma)^{5/2}\cdot\Big(\frac{\lambda+1+\gamma}{\sqrt{(\lambda+1+\gamma)^2-4\gamma}}-1\Big)^2}$$

$$= R^2\cdot\frac{4\gamma^2\big[(\gamma-1)^2+(\gamma+1)\lambda\big]}{((\lambda+1+\gamma)^2-4\gamma)^{3/2}\cdot\Big(\lambda+1+\gamma-\sqrt{(\lambda+1+\gamma)^2-4\gamma}\Big)^2}.$$

This proves the desired result. □

### B.4.1 INTERCHANGING DERIVATIVE AND EXPECTATION / LIMIT

Here we rigorously establish the interchange of the derivative and the expectation / limit used in (43). For convenience of notation let $\boldsymbol{\Sigma} = \widehat{\boldsymbol{\Sigma}}_n = \mathbf{X}_t^\top\mathbf{X}_t/n$ denote the empirical covariance matrix of $\mathbf{X}_t$. We wish to show that

$$\frac{d}{d\lambda_2}\lim_{d,n\to\infty,d/n\to\gamma}\frac{1}{d}\mathbb{E}\big[\mathrm{tr}((\lambda_1\mathbf{I}_d + \lambda_2\boldsymbol{\Sigma})^{-1})\big] = \lim_{d,n\to\infty,d/n\to\gamma}\frac{1}{d}\mathbb{E}\Big[\frac{d}{d\lambda_2}\mathrm{tr}((\lambda_1\mathbf{I}_d + \lambda_2\boldsymbol{\Sigma})^{-1})\Big].$$

This involves the interchange of derivative and limit, and then the interchange of derivative and expectation.

**Interchange of derivative and expectation** First, we show that for any fixed $(d,n)$,

$$\frac{d}{d\lambda_2}\mathbb{E}\big[\mathrm{tr}((\lambda_1\mathbf{I}_d + \lambda_2\boldsymbol{\Sigma})^{-1})\big] = \mathbb{E}\Big[\frac{d}{d\lambda_2}\mathrm{tr}((\lambda_1\mathbf{I}_d + \lambda_2\boldsymbol{\Sigma})^{-1})\Big].$$

By definition of the derivative, we have

$$\frac{d}{d\lambda_2}\mathbb{E}\big[\mathrm{tr}((\lambda_1\mathbf{I}_d + \lambda_2\boldsymbol{\Sigma})^{-1})\big] = \lim_{t\to0}\mathbb{E}\Big[\frac{\mathrm{tr}((\lambda_1\mathbf{I}_d + \lambda_2\boldsymbol{\Sigma} + t\boldsymbol{\Sigma})^{-1}) - \mathrm{tr}((\lambda_1\mathbf{I}_d + \lambda_2\boldsymbol{\Sigma})^{-1})}{t}\Big].$$

For any $\mathbf{A} \succ \mathbf{0}$, the function $t \mapsto \mathrm{tr}((\mathbf{A}+t\mathbf{B})^{-1})$ is continuously differentiable at $t=0$ with derivative $-\mathrm{tr}(\mathbf{A}^{-2}\mathbf{B})$, and thus locally Lipschitz around $t=0$ with Lipschitz constant $|\mathrm{tr}(\mathbf{A}^{-2}\mathbf{B})| + 1$. Applying this in the above expectation with $\mathbf{A} = \lambda_1\mathbf{I}_d + \lambda_2\boldsymbol{\Sigma} \succeq \lambda_1\mathbf{I}_d$ and $\mathbf{B} = \boldsymbol{\Sigma}$, we get that for sufficiently small $|t|$, the fraction inside the expectation is upper bounded by $|\mathrm{tr}(\lambda_1^{-2}\boldsymbol{\Sigma})| + 1 < \infty$ uniformly over $t$. Thus by the Dominated Convergence Theorem, the limit can be passed into the expectation, which yields the expectation of the derivative.

**Interchange of derivative and limit** Define $f_{n,d}(\lambda_2) := \frac{1}{d}\mathbb{E}\big[\text{tr}\big((\lambda_1 \mathbf{I}_d + \lambda_2 \mathbf{\Sigma})^{-1}\big)\big]$. It suffices to show that

$$\frac{d}{d\lambda_2} \lim_{d,n\to\infty, d/n\to\gamma} f_{n,d}(\lambda_2) = \lim_{d,n\to\infty, d/n\to\gamma} f'_{n,d}(\lambda_2),$$

where

$$f'_{n,d}(\lambda_2) = \mathbb{E}\left[\frac{d}{d\lambda_2}\frac{1}{d}\text{tr}\big((\lambda_1\mathbf{I}_d + \lambda_2\mathbf{\Sigma})^{-1}\big)\right] = -\frac{1}{d}\mathbb{E}\big[\text{tr}\big((\lambda_1\mathbf{I}_d + \lambda_2\mathbf{\Sigma})^{-2}\mathbf{\Sigma}\big)\big]$$

by the result of the preceding part.

As $f_{n,d}(\lambda_2) \to s(\lambda_1, \lambda_2)$ pointwise over $\lambda_2$ by properties of the Wishart matrix (Bai & Silverstein, 2010) and each individual $f_{n,d}$ is differentiable, it suffices to show that the derivatives $f'_{n,d}(\widetilde{\lambda}_2)$ converges uniformly for $\widetilde{\lambda}_2$ in a neighborhood of $\lambda_2$. Observe that can rewrite $f'_{n,d}$ as

$$f'_{n,d}(\widetilde{\lambda}_2) = -\mathbb{E}_{\widehat{\mu}_{n,d}}\left[\mathbb{E}_{\lambda\sim\widehat{\mu}_{n,d}}\left[g_{\widetilde{\lambda}_2}(\lambda)\right]\right],$$

where $\widehat{\mu}_{n,d}$ is the empirical distribution of the eigenvalues of $\mathbf{\Sigma}$, and

$$g_{\widetilde{\lambda}_2}(\lambda) := \frac{\lambda}{(\lambda_1 + \widetilde{\lambda}_2\lambda)^2} \leq \frac{1}{\lambda_1\widetilde{\lambda}_2} \quad \text{for all } \lambda \geq 0.$$

Therefore, as $\widehat{\mu}_{n,d}$ converges weakly to the Marchenko-Pastur distribution with probability one and $g_{\widetilde{\lambda}_2}$ is uniformly bounded for $\widetilde{\lambda}_2$ in a small neighborhood of $\lambda_2$, we get that $f'_{n,d}(\widetilde{\lambda}_2)$ does converge uniformly to the expectation of $g_{\widetilde{\lambda}_2}(\lambda)$ under the Marchenko-Pastur distribution. This shows the desired interchange of derivative and limit.

## B.5 PROOF OF THEOREM 4

Throughout this proof we assume that $R^2 = 1$ without loss of generality (as all the rates are constant multiples of $R^2$).

**Part I: Optimal rate for $L^{\text{tr-tr}}$** By Theorem 9, we have

$$\inf_{\lambda>0} \lim_{d,n\to\infty, d/n=\gamma} \text{AsymMSE}\big(\widehat{\mathbf{w}}_{0,T}^{\text{tr-tr}}(n;\lambda)\big)$$

$$= \inf_{\lambda>0} \underbrace{\frac{4\gamma^2\big[(\gamma-1)^2 + (\gamma+1)\lambda\big]}{(\lambda+1+\gamma - \sqrt{(\lambda+\gamma+1)^2 - 4\gamma})^2 \cdot ((\lambda+\gamma+1)^2 - 4\gamma)^{3/2}}}_{:=f(\lambda,\gamma)}.$$

In order to bound $\inf_{\lambda>0} f(\lambda,\gamma)$, picking any $\lambda = \lambda(\gamma)$ gives $f(\lambda(\gamma),\gamma)$ as a valid upper bound, and our goal is to choose $\lambda$ that yields a bound as tight as possible. Here we consider the choice

$$\lambda = \lambda(\gamma) = \max\{1 - \gamma/2, \gamma - 1/2\} = (1 - \gamma/2)\mathbf{1}\{\gamma \leq 1\} + (\gamma - 1/2)\mathbf{1}\{\gamma > 1\}$$

which we now show yields the claimed upper bound.

**Case 1: $\gamma \leq 1$** Substituting $\lambda = 1 - \gamma/2$ into $f(\lambda,\gamma)$ and simplifying, we get

$$f(1 - \gamma/2, \gamma) = \frac{2(\gamma^2 - 3\gamma + 4)}{(2 - \gamma/2)^3} =: g_1(\gamma).$$

Clearly, $g_1(0) = 1$ and $g_1(1) = 32/27$. Further differentiating $g_1$ twice gives

$$g_1''(\gamma) = \frac{\gamma^2 + 7\gamma + 4}{(2 - \gamma/2)^5} > 0 \quad \text{for all } \gamma \in [0,1].$$

Thus $g_1$ is convex on $[0,1]$, from which we conclude that

$$g_1(\gamma) \leq (1-\gamma)\cdot g_1(0) + \gamma\cdot g_1(1) = 1 + \frac{5}{27}\gamma.$$

**Case 2:** $\gamma > 1$  Substituting $\lambda = \gamma - 1/2$ into $f(\lambda, \gamma)$ and simplifying, we get

$$f(\gamma - 1/2, \gamma) = \frac{2\gamma^2(4\gamma^2 - 3\gamma + 1)}{(2\gamma - 1/2)^3} =: g_2(\gamma).$$

We have $g_2(1) = g_1(1) = 32/27$. Further differentiating $g_2$ gives

$$g_2'(\gamma) = -\frac{1}{(4\gamma - 1)^2} - \frac{6}{(4\gamma - 1)^3} - \frac{6}{(4\gamma - 1)^4} + 1 < 1 \quad \text{for all } \gamma > 1.$$

Therefore we have for all $\gamma > 1$ that

$$g_2(\gamma) = g_2(1) + \int_1^\gamma g_2'(t)dt \le g_2(1) + \gamma - 1 = \gamma + \frac{5}{27}.$$

Combining Case 1 and 2, we get

$$\inf_{\lambda > 0} f(\lambda, \gamma) \le g_1(\gamma)$$

$$\le \mathbf{1}\{\gamma \le 1\} + g_2(\gamma)\mathbf{1}\{\gamma > 1\} \le \left(1 + \frac{5}{27}\gamma\right)\mathbf{1}\{\gamma \le 1\} + \left(\frac{5}{27} + \gamma\right)\mathbf{1}\{\gamma > 1\}$$

$$= \max\left\{1 + \frac{5}{27}\gamma, \frac{5}{27} + \gamma\right\}.$$

This is the desired upper bound for $L^{\text{tr-tr}}$.

**Equality at $\gamma = 1$**  We finally show that the above upper bound becomes an equality when $\gamma = 1$. At $\gamma = 1$, we have

$$f(\lambda, 1) = \frac{8\lambda}{(\lambda + 2 - \sqrt{\lambda^2 + 4\lambda})^2(\lambda^2 + 4\lambda)^{3/2}} = \frac{8\lambda^{-4}}{(1 + 2/\lambda - \sqrt{1 + 4/\lambda})^2(1 + 4/\lambda)^{3/2}}.$$

Make the change of variable $t = \sqrt{1 + 4/\lambda}$ so that $\lambda^{-1} = (t^2 - 1)/4$, minimizing the above expression is equivalent to minimizing

$$\frac{(t^2 - 1)^4/32}{(t^2/2 - t + 1/2)^2 t^3} = \frac{(t + 1)^4}{8t^3}$$

over $t > 1$. It is straightforward to check (by computing the first and second derivatives) that the above quantity is minimized at $t = 3$ with value $32/27$. In other words, we have shown

$$\inf_{\lambda > 0} f(\lambda, 1) = \frac{32}{27} = \max\left\{1 + \frac{5}{27}\gamma, \frac{5}{27} + \gamma\right\}\Big|_{\gamma = 1},$$

that is, the equality holds at $\gamma = 1$.

**Part II: Optimal rate for $L^{\text{tr-val}}$**  We now prove the result on $L^{\text{tr-val}}$, that is,

$$\inf_{\lambda > 0, s \in (0,1)} \lim_{d,n \to \infty, d/n = \gamma} \text{AsymMSE}\left(\widehat{\mathbf{w}}_{0,T}^{\text{tr-val}}(ns, n(1-s); \lambda)\right)$$

$$\overset{(i)}{=} \lim_{d,n \to \infty, d/n = \gamma} \underbrace{\inf_{\lambda > 0, n_1 + n_2 = n} \text{AsymMSE}\left(\widehat{\mathbf{w}}_{0,T}^{\text{tr-val}}(n_1, n_2; \lambda)\right)}_{\frac{d+n+1}{n}} \overset{(ii)}{=} 1 + \gamma.$$

First, equality (ii) follows from Corollary 8 and the fact that $(d + n + 1)/n \to 1 + \gamma$. Second, the "$\ge$" direction of equality (i) is trivial (since we always have "inf lim $\ge$ lim inf"). Therefore we get the "$\ge$" direction of the overall equality, and it remains to prove the "$\le$" direction.

For the "$\le$" direction, we fix any $\lambda > 0$, and bound $\text{AsymMSE}(\widehat{\mathbf{w}}_{0,T}^{\text{tr-val}}(n_1, n_2; \lambda))$ (and consequently its limit as $d, n \to \infty$.) We have from Lemma 5 that

$$\text{AsymMSE}(\widehat{\mathbf{w}}_{0,T}^{\text{tr-val}}(n_1, n_2; \lambda))$$

$$= \frac{d}{n_2} \cdot \frac{\mathbb{E}\left[\left(\sum_{i=1}^{d} \lambda^2/(\sigma_i^{(n_1)} + \lambda)^2\right)^2 + (n_2 + 1)\sum_{i=1}^{d} \lambda^4/(\sigma_i^{(n_1)} + \lambda)^4\right]}{\left(\mathbb{E}\left[\sum_{i=1}^{d} \lambda^2/(\sigma_i^{(n_1)} + \lambda)^2\right]\right)^2}$$

$$\leq \frac{d}{n_2} \cdot \frac{d^2 + (n_2 + 1)d}{\left(\mathbb{E}\left[\sum_{i=1}^{d} \lambda^2/(\sigma_i^{(n_1)} + \lambda)^2\right]\right)^2}$$

$$= \frac{d + n_2 + 1}{n_2} \cdot \frac{1}{\left(\mathbb{E}\left[\frac{1}{d}\sum_{i=1}^{d} \lambda^2/(\sigma_i^{(n_1)} + \lambda)^2\right]\right)^2}$$

Observe that

$$\mathbb{E}\left[\frac{1}{d}\sum_{i=1}^{d} \frac{\lambda^2}{(\sigma_i^{(n_1)} + \lambda)^2}\right] \overset{(i)}{\geq} \mathbb{E}\left[\frac{\lambda^2}{\left(\sum_{i=1}^{d}\sigma_i^{(n_1)}/d + \lambda\right)^2}\right]$$

$$\overset{(ii)}{\geq} \frac{\lambda^2}{\left(\mathbb{E}\left[\sum_{i=1}^{d}\sigma_i^{(n_1)}/d\right] + \lambda\right)^2} \overset{(iii)}{=} \frac{\lambda^2}{(1 + \lambda)^2},$$

where (i) follows from the convexity of $t \mapsto \lambda^2/(t + \lambda)^2$ on $t \geq 0$; (ii) follows from the same convexity and Jensen's inequality, and (iii) is since $\mathbb{E}\left[\sum_{i=1}^{d}\sigma_i^{(n_1)}\right] = \mathbb{E}\left[\text{tr}(\frac{1}{n_1}\mathbf{X}_t^\top \mathbf{X}_t)\right] = \mathbb{E}\left[\|\mathbf{X}_t\|_{\mathsf{Fr}}^2/n_1\right] = d$. Applying this in the preceeding bound yields

$$\text{AsymMSE}(\widehat{\mathbf{w}}_{0,T}^{\mathsf{tr\text{-}val}}(n_1, n_2; \lambda)) \leq \frac{d + n_2 + 1}{n_2} \cdot \frac{(1 + \lambda)^2}{\lambda^2}.$$

Further plugging in $n_1 = ns$ and $n_2 = n(1 - s)$ for any $s \in (0, 1)$ yields

$$\lim_{d,n\to\infty, d/n\to\gamma} \text{AsymMSE}(\widehat{\mathbf{w}}_{0,T}^{\mathsf{tr\text{-}val}}(ns, n(1 - s); \lambda)) \leq \frac{\gamma + 1 - s}{1 - s} \cdot \frac{(1 + \lambda)^2}{\lambda^2}.$$

Finally, the right-hand side is minimized at $\lambda \to \infty$ and $s = 0$, from which we conclude that

$$\inf_{\lambda>0, s\in(0,1)} \lim_{d,n\to\infty,\ d/n\to\gamma} \text{AsymMSE}(\widehat{\mathbf{w}}_{0,T}^{\mathsf{tr\text{-}val}}(n_1, n_2; \lambda)) \leq 1 + \gamma,$$

which is the desired "$\leq$" direction. $\qquad\square$

## C  CONNECTIONS TO BAYESIAN ESTIMATOR

Here we discuss the relationship between our train-train meta-learining estimator using ridge regression solvers and a Bayesian estimator under a somewhat natural hierarchical generative model for the realizable setting in Section 4. We show that these two estimators are not equal in general, albeit they have some similarities in their expressions.

We consider the following hierarchical probabilitistic model:

$$\mathbf{w}_{0,\star} \sim \mathsf{N}\left(0, \frac{\sigma_w^2}{d}\mathbf{I}_d\right), \quad \mathbf{w}_t | \mathbf{w}_{0,\star} \overset{\text{iid}}{\sim} \mathsf{N}\left(\mathbf{w}_{0,\star}, \frac{R^2}{d}\mathbf{I}_d\right), \quad \mathbf{y}_t = \mathbf{X}_t \mathbf{w}_t + \sigma \mathbf{z}_t \text{ where } \mathbf{z}_t \overset{\text{iid}}{\sim} \mathsf{N}(0, \mathbf{I}_n).$$

This model is similar to our realizable linear model (6), except that $\mathbf{w}_0$ has a prior and that there is observation noise in the data (such that data likelihoods and posteriors are well-defined). We also note that the $R^2/d$ variance for $\mathbf{w}_t$ guarantees that $\mathbb{E}[\|\mathbf{w}_t - \mathbf{w}_{0,\star}\|^2] = R^2$, consistent with our definition (7).

**Bayesian estimator**   We now derive the Bayesian posterior mean estimator of $\mathbf{w}_{0,\star}$, which requires us to compute the posterior distribution of $\mathbf{w}_{0,\star}$ given the data $\{(\mathbf{X}_t, \mathbf{y}_t)\}_{t=1}^T$ [4].

We begin by computing the likelihood of one task by marginalizing over $\mathbf{w}_t$:

$$p(\mathbf{X}_t, \mathbf{y}_t | \mathbf{w}_{0,\star}) \propto \int p(\mathbf{w}_t | \mathbf{w}_{0,\star}) \cdot p(\mathbf{y}_t | \mathbf{X}_t, \mathbf{w}_t) d\mathbf{w}_t$$

$$\propto \int \exp\left(-\frac{\|\mathbf{w}_t - \mathbf{w}_{0,\star}\|^2}{2R^2/d}\right) \cdot \exp\left(-\frac{\|\mathbf{y}_t - \mathbf{X}_t \mathbf{w}_t\|^2}{2\sigma^2}\right) d\mathbf{w}_t$$

$$\overset{(i)}{\propto} \exp\left(-\frac{\|\mathbf{w}_{0,\star}\|^2}{2R^2/d} + \frac{1}{2}\left(\frac{\mathbf{w}_{0,\star}}{R^2/d} + \frac{\mathbf{X}_t^\top \mathbf{y}_t}{\sigma^2}\right)^\top \left(\frac{\mathbf{X}_t^\top \mathbf{X}_t}{\sigma^2} + \frac{\mathbf{I}_d}{R^2/d}\right)^{-1} \left(\frac{\mathbf{w}_{0,\star}}{R^2/d} + \frac{\mathbf{X}_t^\top \mathbf{y}_t}{\sigma^2}\right)\right)$$

$$\propto \exp\left(-\frac{1}{2}\mathbf{w}_{0,\star}^\top \left(\left(\mathbf{X}_t^\top \mathbf{X}_t + \frac{d\sigma^2}{R^2}\mathbf{I}_d\right)^{-1} \frac{\mathbf{X}_t^\top \mathbf{X}_t}{R^2/d}\right)\mathbf{w}_{0,\star} + \mathbf{w}_{0,\star}^\top \left(\mathbf{X}_t^\top \mathbf{X}_t + \frac{d\sigma^2}{R^2}\mathbf{I}_d\right)^{-1} \frac{\mathbf{X}_t^\top \mathbf{y}_t}{R^2/d}\right),$$

where (i) is obtained by integrating a multivariate Gaussian density over $\mathbf{w}_t$, and "$\propto$" drops all the terms that do not depend on $\mathbf{w}_{0,\star}$. Therefore, by the Bayes rule, the overall posterior distribution of $\mathbf{w}_{0,\star}$ is given by

$$p\left(\mathbf{w}_{0,\star} | \{(\mathbf{X}_t, \mathbf{y}_t)\}_{t=1}^T\right) \propto p(\mathbf{w}_{0,\star}) \cdot \prod_{t=1}^T p(\mathbf{X}_t, \mathbf{y}_t | \mathbf{w}_{0,\star})$$

$$\propto \exp\left(-\frac{\|\mathbf{w}_{0,\star}\|^2}{2\sigma_w^2/d}\right) \cdot$$

$$\prod_{t=1}^T \exp\left(-\frac{1}{2}\mathbf{w}_{0,\star}^\top \left(\left(\mathbf{X}_t^\top \mathbf{X}_t + \frac{d\sigma^2}{R^2}\mathbf{I}_d\right)^{-1} \frac{\mathbf{X}_t^\top \mathbf{X}_t}{R^2/d}\right)\mathbf{w}_{0,\star} + \mathbf{w}_{0,\star}^\top \left(\mathbf{X}_t^\top \mathbf{X}_t + \frac{d\sigma^2}{R^2}\mathbf{I}_d\right)^{-1} \frac{\mathbf{X}_t^\top \mathbf{y}_t}{R^2/d}\right).$$

This means that the posterior distribution of $\mathbf{w}_{0,\star}$ is Gaussian, with mean , i.e. the Bayesian estimator, equal to [5]

$$\widehat{\mathbf{w}}_{0,T}^{\mathsf{Bayes}} := \mathbb{E}\left[\mathbf{w}_{0,\star} \mid \{(\mathbf{X}_t, \mathbf{y}_t)\}_{t=1}^T\right] = (\mathbf{A}_T^{\mathsf{Bayes}})^{-1} \mathbf{c}_T^{\mathsf{Bayes}},$$

where

$$\mathbf{A}_T^{\mathsf{Bayes}} := \frac{d}{\sigma_w^2}\mathbf{I}_d + \sum_{t=1}^T \left(\mathbf{X}_t^\top \mathbf{X}_t + \frac{d\sigma^2}{R^2}\mathbf{I}_d\right)^{-1} \frac{\mathbf{X}_t^\top \mathbf{X}_t}{R^2/d},$$

$$\mathbf{c}_T^{\mathsf{Bayes}} := \sum_{t=1}^T \left(\mathbf{X}_t^\top \mathbf{X}_t + \frac{d\sigma^2}{R^2}\mathbf{I}_d\right)^{-1} \frac{\mathbf{X}_t^\top \mathbf{y}_t}{R^2/d}.$$

We note that $\widehat{\mathbf{w}}_{0,T}^{\mathsf{Bayes}}$ has a similar form as our train-train estimator, but is not exactly the same. Indeed, recall the closed form of our train-train estimator is (cf. (10))

$$\widehat{\mathbf{w}}_{0,T}^{\mathsf{tr\text{-}tr}} = (\mathbf{A}_T^{\mathsf{tr\text{-}tr}})^{-1} \mathbf{c}_T^{\mathsf{tr\text{-}tr}},$$

where

$$\mathbf{A}_T^{\mathsf{tr\text{-}tr}} = \sum_{t=1}^T \left(\mathbf{X}_t^\top \mathbf{X}_t + n\lambda\mathbf{I}_d\right)^{-2} \mathbf{X}_t^\top \mathbf{X}_t,$$

$$\mathbf{c}_T^{\mathsf{tr\text{-}tr}} = \sum_{t=1}^T \left(\mathbf{X}_t^\top \mathbf{X}_t + n\lambda\mathbf{I}_d\right)^{-2} \mathbf{X}_t^\top \mathbf{y}_t.$$

As $\widehat{\mathbf{w}}_{0,T}^{\mathsf{Bayes}}$ uses the **inverse** and $\widehat{\mathbf{w}}_{0,T}^{\mathsf{tr\text{-}tr}}$ uses the **squared inverse**, these two sets of estimators are not the same in general, no matter how we tune the $\lambda$ in the train-train estimator. This is true even if we set $\sigma_w = \infty$ so that the prior of $\mathbf{w}_{0,\star}$ becomes degenerate (and the Bayesian estimator reduces to the MLE).

---

[4] Hereafter we treat $\mathbf{X}_t$ as fixed, as the density of $\mathbf{X}_t$ won't affect the Bayesian calculation.

[5] Any density $p(\mathbf{w}) \propto \exp(-\mathbf{w}^\top \mathbf{A}\mathbf{w}/2 + \mathbf{w}^\top \mathbf{c})$ specifies a Gaussian distreibution $\mathsf{N}(\boldsymbol{\mu}, \boldsymbol{\Sigma})$, where $\mathbf{A} = \boldsymbol{\Sigma}^{-1}$ and $\mathbf{c} = \boldsymbol{\Sigma}^{-1}\boldsymbol{\mu}$, so that $\boldsymbol{\mu} = \mathbf{A}^{-1}\mathbf{c}$.

## D    DETAILS ON THE FEW-SHOT IMAGE CLASSIFICATION EXPERIMENT

Here we provide additional details of the few-shot image classification experiment in Section 5.2.

**Optimization and architecture**    For both methods, we run a few gradient steps on the inner argmin problem to obtain (an approximation of) $\mathbf{w}_t$, and plug $\mathbf{w}_t$ into the $\nabla_{\mathbf{w}_0}\ell_t^{\{\text{tr-val,tr-tr}\}}(\mathbf{w}_0)$ (which involves $\mathbf{w}_t$ through implicit function differentiation) for optimizing $\mathbf{w}_0$ in the outer loop.

For both train-train and train-val methods, we use the standard 4-layer convolutional network in (Finn et al., 2017; Zhou et al., 2019) as the backbone (i.e. the architecture for $\mathbf{w}_t$). We further tune their hyper-parameters, such as the regularization constant $\lambda$, the learning rate (initial learning rate and its decay strategy), and the gradient clipping threshold.

**Datasets**    We experiment on miniImageNet (Ravi & Larochelle, 2017) and tieredImageNet (Ren et al., 2018). MiniImageNet consists of 100 classes of images from ImageNet (Krizhevsky et al., 2012) and each class has 600 images of resolution $84 \times 84 \times 3$. We use 64 classes for training, 16 classes for validation, and the remaining 20 classes for testing (Ravi & Larochelle, 2017). Tiered-ImageNet consists of 608 classes from the ILSVRC-12 data set (Russakovsky et al., 2015) and each image is also of resolution $84 \times 84 \times 3$. TieredImageNet groups classes into broader hierarchy categories corresponding to higher-level nodes in the ImageNet. Specifically, its top hierarchy has 20 training categories (351 classes), 6 validation categories (97 classes) and 8 test categories (160 classes). This structure ensures that all training classes are distinct from the testing classes, providing a more realistic few-shot learning scenario.

### D.1    EFFECT OF THE SPLIT RATIO FOR THE TRAIN-VAL METHOD

We further tune the split $(n_1, n_2)$ in the train-val method and report the results in Table 2. As can be seen, as the number of test samples $n_2$ increases, the percent classification accuracy on both the miniImageNet and tieredImageNet datasets becomes higher. This testifies our theoretical affirmation in Corollary 8. However, note that even if we take the best split $(n_1, n_2) = (5, 25)$ (and compare again with Table 1), the train-val method still performs worse than the train-train method.

We remark that our theoretical results on train-train performing better than train-val (in Section 4) rely on the assumptions that the data can be exactly realized by the representation and contains no label noise. Our experimental results here may suggest that the miniImageNet and tieredImageNet few-shot tasks may have a similar structure (there exists a NN representation that almost perfectly realizes the label with no noise) that allows the train-train method to perform better than the train-val method.

Table 2: Investigation of the effects of training/validation splitting ratio in the train-val method (iMAML) to the few-shot classification accuracy (%) on miniImageNet and tieredImageNet.

| datasets | $n_1 = 25, n_2 = 5$ | $n_1 = 15, n_2 = 15$ | $n_1 = 5, n_2 = 25$ |
|---|---|---|---|
| miniImageNet | $62.09 \pm 0.97$ | $63.56 \pm 0.95$ | $63.92 \pm 1.04$ |
| tieredImageNet | $66.45 \pm 1.05$ | $67.30 \pm 0.98$ | $67.50 \pm 0.94$ |

## E    COMPARISON WITH CROSS-VALIDATION ON SYNTHETIC DATA

We test the effect of using cross-validation for the train-val method on the same synthetic data (realizable linear centroid meta-learning) as in Section 5.1.

**Method**    We fix the number of per-task data $n = 20$, and use 4-fold cross validation in the following two settings: $(n_1, n_2) = (5, 15)$, and $(n_1, n_2) = (15, 5)$. In both cases, we partition the data into 4 parts each with 5 data points, and we roulette over 4 possible partitions of which one as train and which one as validation. The estimated optimal $\widehat{\mathbf{w}}_0^{\text{cv}}$ is obtained by minimize the averaged

train-val loss over the 4 partitions:

$$\ell_t^{\mathsf{cv}}(\mathbf{w}_0) := \frac{1}{4} \sum_{j=1}^{4} \frac{1}{2n_{\mathsf{val}}} \left\| \mathbf{y}_t^{\mathsf{val},j} - \mathbf{X}_t^{\mathsf{val},j} \mathcal{A}_\lambda(\mathbf{w}_0; \mathbf{X}_t^{\mathsf{train},j}, \mathbf{y}_t^{\mathsf{train},j}) \right\|^2,$$

$$\widehat{\mathbf{w}}_0^{\mathsf{cv}} = \arg\min_{\mathbf{w}_0} \frac{1}{T} \sum_{t=1}^{T} \ell_t^{\mathsf{cv}}(\mathbf{w}_0),$$

where superscript $j$ denotes the index of the cross-validation. The performance is depicted in Figure 2.

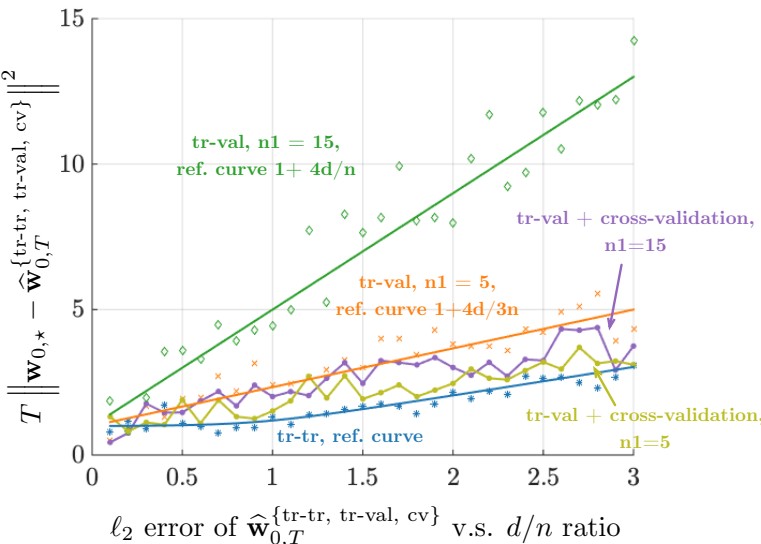

Figure 2: The scaled (by $T$) $\ell_2$-error of $\widehat{\mathbf{w}}_{0,T}^{\{\mathsf{tr\text{-}tr},\mathsf{tr\text{-}val},\mathsf{cv}\}}$ as the ratio $d/n$ varies from 0 to 3 ($n = 20$ and $T = 1000$ are fixed). For the cross-validation method, the regularization coefficient $\lambda = 0.5$ is tuned.

**Result**    As showin in Figure 2, for both $(n_1, n_2) = (15, 5)$ and $(n_1, n_2) = (5, 15)$, using cross-validation consistently beats the performance of the train-val method. This demonstrates the variance-reduction effect of cross-validation. Note that the best performance (among the cross-validation methods) is still achieved at $n_1 = 5$, similar as for the vanilla train-val method. However, numerically, the best cross-validation performance is still not as good as the train-train method.

**Leave-one-out cross-validation**    Figure 3 left further tests with an increased number of per-task samples $n = 40$, and incorporates the train-val method with the leave-one-out cross-validation, i.e., $(n_1, n_2) = (39, 1)$ and $(n_1, n_2) = (1, 39)$. We repeat the experiment 10 times for plotting the error bar (shaded area). We see that the train-train method still outperforms the train-val method with leave-one-out validation.

We further increase the per-task sample size $n$ to 200, and test the leave-one-out method with a sample split of $(n_1, n_2) = (1, 199)$. We adopt a matrix inverse trick to mitigate the computational overhead of finding $\mathcal{A}_\lambda(\mathbf{w}_0; \mathbf{X}_t^{\mathsf{train},j}, \mathbf{y}_t^{\mathsf{train},j})$. To ease the computation, we also vary $d$ from 0 to 400 on a coarse grid (with an increment of 80). From Figure 3 right, we see that the leave-one-out method can slightly beat the train-train method for some $d/n$ values. Compared to $n = 20$ and $n = 40$ experiments, this is the first time of seeing leave-one-out method outperforms the train-train method. We suspect that the per-task sample size $n$ plays a vital role in the power of the leave-one-out method: a large $n$ tends to have a strong variance reduction effect in the leave-one-out method, so that the performance can be improved. Yet using the leave-one-out method with a large $n$ invokes a high computational burden.

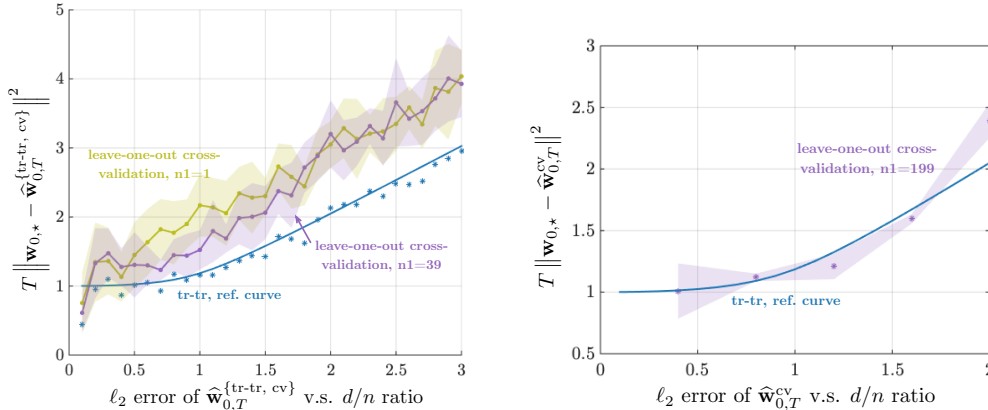

Figure 3: The scaled (by $T$) $\ell_2$-error of $\widehat{\mathbf{w}}_{0,T}^{\{\text{tr-tr,cv}\}}$ as the ratio $d/n$ varies from 0 to 3 ($n \in \{40, 200\}$ and $T = 1000$ are fixed). For the cross-validation method, the regularization coefficient $\lambda = 0.5$. Left: $n = 40$. Leave-out-out CV performs worse than the train-train method. Right: $n = 200$. Leave one-out CV appears better than the train-train method for $d/n \in \{1.2, 1.6\}$.

## F    Nonasymptotic analysis under general scalings of $d, n, T$

We sketch an nonasymptotic analysis of the train-train method in the realizable linear model in Section 4. We assume in addition that $\mathbf{w}_t \stackrel{\text{iid}}{\sim} \mathsf{N}(\mathbf{w}_{0,\star}, \frac{R^2}{d}\mathbf{I}_d)$[6].

We begin by recalling from Appendix B.1 and B.2 that the train-train and train-val estimators have closed-form expressions

$$\widehat{\mathbf{w}}_{0,T}^{\text{tr-tr}} = \left(\sum_{t=1}^{T} \mathbf{A}_t\right)^{-1} \sum_{t=1}^{T} \mathbf{A}_t \mathbf{w}_t, \tag{44}$$

$$\widehat{\mathbf{w}}_{0,T}^{\text{tr-val}} = \left(\sum_{t=1}^{T} \mathbf{B}_t\right)^{-1} \sum_{t=1}^{T} \mathbf{B}_t \mathbf{w}_t, \tag{45}$$

where

$$\mathbf{A}_t := \lambda^2 \left(\frac{\mathbf{X}_t^\top \mathbf{X}_t}{n} + \lambda \mathbf{I}_d\right)^{-2} \frac{\mathbf{X}_t^\top \mathbf{X}_t}{n},$$

$$\mathbf{B}_t := \lambda^2 \left(\frac{\mathbf{X}_t^{\text{train}\top} \mathbf{X}_t^{\text{train}}}{n_1} + \lambda \mathbf{I}_d\right)^{-1} \frac{\mathbf{X}_t^{\text{val}\top} \mathbf{X}_t^{\text{val}}}{n_2} \left(\frac{\mathbf{X}_t^{\text{train}\top} \mathbf{X}_t^{\text{train}}}{n_1} + \lambda \mathbf{I}_d\right)^{-1}.$$

**Nonasymptotic result for $\widehat{\mathbf{w}}_{0,T}^{\text{tr-tr}}$**    For simplicity, we restrict attention on the analysis of the train-train estimator $\widehat{\mathbf{w}}_{0,T}^{\text{tr-tr}}$, whose closed form expression is given in (44). Analysis of the train-val estimator can be done in a similar fashion.

We sketch a proof for the following

**Result**: Let $T = \Omega(d)$ and $(d, n)$ is such that

$$f_{\text{tr-tr}}(d, n) = \frac{\mathbb{E}\left[\frac{1}{d} \sum_{i=1}^{d} (\sigma_i^{(n)})^2 / (\sigma_i^{(n)} + \lambda)^4\right]}{\left(\frac{1}{d}\mathbb{E}\left[\sum_{i=1}^{d} \sigma_i^{(n)} / (\sigma_i^{(n)} + \lambda)^2\right]\right)^2} = \Theta(1)$$

---

[6]This Gaussian assumption on $\mathbf{w}_t$ is mainly for technical convenience, and can be relaxed to that $w_{t,i} - w_{0,\star,i}$ are i.i.d. with variance $R^2/d$ and sub-Gaussian with parameter $O(R^2/d)$ without changing the proof.

(which for example holds if $(d, n)$ are in the proportional limit), then with high probability we have

$$
\mathrm{MSE}(\widehat{\mathbf{w}}_{0,T}^{\mathsf{tr\text{-}tr}}) = \frac{R^2}{d}\left(1 \pm O\left(\frac{1}{\sqrt{d}}\right)\right) \cdot \left[\frac{1}{T}\cdot\left(d \cdot f_{\mathsf{tr\text{-}tr}}(d,n) \pm O\left(d\sqrt{\frac{\log(T/\delta)}{T}}\right)\right)\right]
$$

$$
\approx \frac{R^2}{T}\cdot f_{\mathsf{tr\text{-}tr}}(d,n) = \frac{1}{T}\mathrm{AsymMSE}(\widehat{\mathbf{w}}_{0,T}^{\mathsf{tr\text{-}tr}}),
$$

that is, the MSE for the train-train method concentrates around the asymptotic MSE.

**Proof Sketch**   We first define the matrix

$$
\boldsymbol{\Sigma}_T := \left(\sum_{t=1}^{T}\mathbf{A}_t\right)^{-2}\sum_{t=1}^{T}\mathbf{A}_t^2 = \frac{1}{T}\cdot\left(\frac{\sum_{t=1}^{T}\mathbf{A}_t}{T}\right)^{-2}\frac{\sum_{t=1}^{T}\mathbf{A}_t^2}{T},
$$

which will be key to our analysis. Observe that conditioned on $\mathbf{A}_t$ (and only looking at the randomness of $\mathbf{w}_t$), we have

$$
\widehat{\mathbf{w}}_{0,T}^{\mathsf{tr\text{-}tr}} - \mathbf{w}_{0,\star} = \left(\sum_{t=1}^{T}\mathbf{A}_t\right)^{-1}\sum_{t=1}^{T}\mathbf{A}_t(\mathbf{w}_t - \mathbf{w}_{0,\star}) \sim \mathsf{N}\left(\mathbf{0}, \frac{R^2}{d}\boldsymbol{\Sigma}_T\right).
$$

Therefore, applying the Hanson-Wright inequality (Vershynin, 2018, Theorem 6.2.1) yields that

$$
\mathrm{MSE}(\widehat{\mathbf{w}}_{0,T}^{\mathsf{tr\text{-}tr}}) = \left\|\mathbf{w}_{0,T}^{\mathsf{tr\text{-}tr}} - \mathbf{w}_{0,\star}\right\|^2 \in \frac{R^2}{d}\left(\mathrm{tr}(\boldsymbol{\Sigma}_T) \pm \max\left\{\|\boldsymbol{\Sigma}_T\|_{\mathsf{Fr}}\sqrt{\log\frac{1}{\delta}}, \|\boldsymbol{\Sigma}_T\|_{\mathrm{op}}\log\frac{1}{\delta}\right\}\right)
$$

with probability at least $1 - \delta$. In the following, we argue that $\mathrm{tr}(\boldsymbol{\Sigma}_T)$ is the dominating term in the above bound and concentrates around the asymptotic MSE in Lemma 5, whereas the error terms within the max are lower order errors compared with $\mathrm{tr}(\boldsymbol{\Sigma}_T)$.

**Concentration of $\mathrm{tr}(\boldsymbol{\Sigma}_T)$**   Recall that $\mathbf{A}_t$ are i.i.d. PSD matrices in $\mathbb{R}^{d\times d}$. We have

$$
\mathrm{tr}(\boldsymbol{\Sigma}_T) = \frac{1}{T}\left\langle\left(\frac{\sum_{t=1}^{T}\mathbf{A}_t}{T}\right)^{-2}, \frac{\sum_{t=1}^{T}\mathbf{A}_t^2}{T}\right\rangle
$$

$$
= \frac{1}{T}\left\{\underbrace{\left\langle\mathbb{E}[\mathbf{A}_1]^{-2}, \mathbb{E}[\mathbf{A}_1^2]\right\rangle}_{\mathrm{I}} + \underbrace{\left\langle\left(\frac{\sum_{t=1}^{T}\mathbf{A}_t}{T}\right)^{-2} - \mathbb{E}[\mathbf{A}_1]^{-2}, \mathbb{E}[\mathbf{A}_1^2]\right\rangle}_{\mathrm{II}} + \underbrace{\left\langle\left(\frac{\sum_{t=1}^{T}\mathbf{A}_t}{T}\right)^{-2}, \frac{\sum_{t=1}^{T}\mathbf{A}_t^2}{T} - \mathbb{E}[\mathbf{A}_1^2]\right\rangle}_{\mathrm{III}}\right\}.
$$

We argue that I is the main term that depends on $(d, n)$ and is independent of $T$. Further, $\mathbf{A}_1$ has eigenvalues $\lambda^2\sigma_i/(\lambda + \sigma_i)^2$ where $\{\sigma_i\}_{i=1}^{d}$ are the eigenvalues of $\mathbf{X}_1^\top\mathbf{X}_1/n$, and $\mathbf{A}_1$ has uniformly distributed eigenvectors. Following the same analysis of Lemma 5, we see that

$$
\mathrm{I} = d^2 \cdot \rho_{\mathsf{tr\text{-}tr}} = d \cdot \underbrace{\frac{\mathbb{E}\left[\frac{1}{d}\sum_{i=1}^{d}(\sigma_i^{(n)})^2/(\sigma_i^{(n)} + \lambda)^4\right]}{\left(\frac{1}{d}\mathbb{E}\left[\sum_{i=1}^{d}\sigma_i^{(n)}/(\sigma_i^{(n)} + \lambda)^2\right]\right)^2}}_{:=f_{\mathsf{tr\text{-}tr}}(d,n)}.
$$

This is exactly the same quantity that appeared in the asymptotic MSE for the train-train method in Theorem 4. In the following we assume that $d, n$ is such that $f_{\mathsf{tr\text{-}tr}}(d, n) = \Theta(1)$.

We further argue that terms II and III are low-order terms compared with term I. Indeed, term I is $O(d)$. Applying matrix concentrations (e.g. the matrix Bernstein inequality), we can get that terms II and III are of order $\max\{\sqrt{d\log(T/\delta)/T}, d\log(T/\delta)/T\}$ with probability at least $1 - \delta$. Combining the terms yields that

$$
\mathrm{tr}(\boldsymbol{\Sigma}_T) \in \frac{1}{T}\left\{d \cdot f_{\mathsf{tr\text{-}tr}}(d,n) \pm d\sqrt{\frac{\log(T/\delta)}{T}}\right\}
$$

with high probability.

**Controlling the error in the MSE**  We now control $\|\mathbf{\Sigma}_T\|_{\mathsf{Fr}}$ and (consequently) $\|\mathbf{\Sigma}_T\|_{\mathrm{op}}$. We wish to show that

$$\|\mathbf{\Sigma}_T\|_{\mathsf{Fr}} \leq O\left(\frac{1}{\sqrt{d}}\right) \cdot \mathrm{tr}(\mathbf{\Sigma}_T)$$

with high probability. This can be seen from the combination of two results: (1) $\mathrm{tr}(\mathbf{\Sigma}_T) = \Omega(d/T)$ by the preceding part, and (2)

$$\|\mathbf{\Sigma}_T\|_{\mathsf{Fr}} \leq \frac{1}{T} \cdot \frac{1}{\lambda_{\min}(\sum_{t \leq T} \mathbf{A}_t)^2} \left\| \sum_{t=1}^{T} \mathbf{A}_t^2/T \right\|_{\mathsf{Fr}} \leq O(\sqrt{d}/T).$$

The above requires (1) $\|\mathbf{A}_t\|_{\mathsf{Fr}} \leq O(\sqrt{d})$, which is true since we have $0 \lesssim \mathbf{A}_t \lesssim \lambda \mathbf{I}_d$ and thus $\|\mathbf{A}_t^2\|_{\mathsf{Fr}} \leq \lambda^2 \sqrt{d}$, and (2) $\lambda_{\min}(\sum_{t \leq T} \mathbf{A}_t/T) = \Omega(1)$ with high probability, which is true whenever $T = \Omega(d)$ because of matrix concentration and the fact that $\lambda_{\min}(\mathbb{E}[\mathbf{A}_1]) = \Omega(1)$.

**Putting together**  We have with high probability that

$$\mathrm{MSE}(\widehat{\mathbf{w}}_{0,T}^{\mathsf{tr\text{-}tr}}) = \frac{R^2}{d}\left(1 \pm O\left(\frac{1}{\sqrt{d}}\right)\right) \cdot \left[\frac{1}{T} \cdot \left(d \cdot f_{\mathsf{tr\text{-}tr}}(d,n) \pm O\left(d\sqrt{\frac{\log(T/\delta)}{T}}\right)\right)\right].$$

as long as $T = \Omega(d)$ and $d, n$ is such that $f_{\mathsf{tr\text{-}tr}}(d,n) = \Theta(1)$. $\quad\square$

### F.1 SYNTHETIC EXPERIMENT

We verify our theoretical findings under general scalings of $d, n, T$. We adopt the data generating model in Section 5.1 again. To compare the performance of the train-train and train-val methods, we choose the number of tasks $T = 300$ and per-task sample size $n = 100$. We vary $d$ from 0 to 300. Note that now $T$ is comparable to $n$ and $d$ and much smaller than $T = 1000$ used in previous experiments. Figure 4 shows the close fit of reference curves to the simulated $\ell_2$ errors.

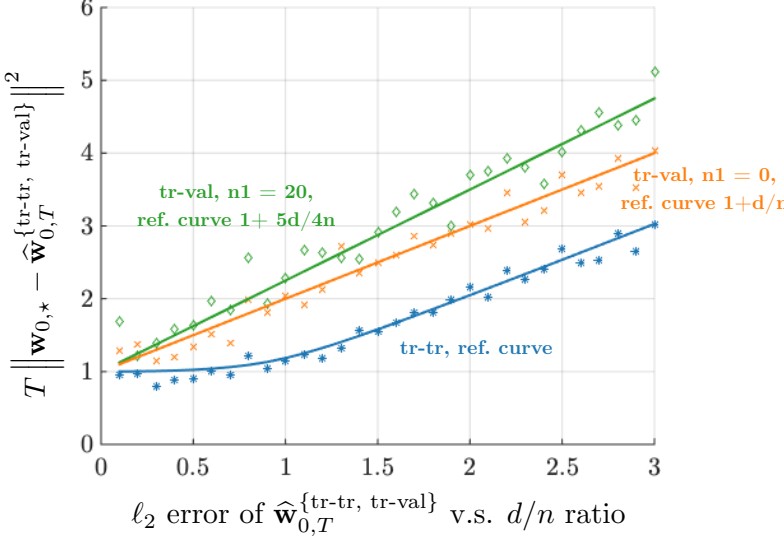

Figure 4: The scaled (by $T$) $\ell_2$-error of $\widehat{\mathbf{w}}_{0,T}^{\{\mathsf{tr\text{-}tr},\mathsf{tr\text{-}val}\}}$ as the ratio $d/n$ varies from 0 to 3 in the general scaling setting ($n = 100$ and $T = 300$ are fixed). The regularization coefficient $\lambda$ is fine tuned for the train-train method and $\lambda = 2000$ for the train-val method.

