# OpenReview forum: "How Important is the Train-Validation Split in Meta-Learning?"
_ICLR.cc/2021/Conference — Reject_

### Official Review · AnonReviewer1 · 2020-10-24
**In this paper, the meta-learning is formulated as a multiple ridge regression problems and the standard statistical asymptotic theory is used to derive some theoretical results. The impact of the results will be rather limited.**

**Rating:** 5
**Confidence:** 3

**Review:**

In this paper, the authors study the theoretical properties of meta-learning. In particular, the train-validation split to tackle the linear centroid meta-learning problem is investigated using statistical asymptotic theory. First, the authors proved that the train-validation method has statistical consistency, while the train-train method has a statistical bias to the centroid. Under the noise-free setting, however, both methods have statistical consistency. Furthermore, the train-train method is superior to the train-validation method in the sense of the asymptotic MSE. Based on the asymptotic analysis the optimal ratio of the data splitting for the train-validation method was also derived. The theoretical findings are confirmed by some numerical experiments.

In this paper, the meta-learning is formulated as a multiple ridge regression problem and the standard statistical asymptotic theory is used to derive some theoretical results. The impact of the results will be rather limited.

In the paper, the difference between the train-val method and the train-train method was investigated. The theoretical properties are well-known in the standard context of statistics and machine learning. That is, the overfitting to the training data yields a serious bias to the generalization error. Information criteria such as AIC, BIC, etc provide the bias correction to fill the gap. I guess that the authors are aware of the close relationship between the present formulation of the meta-learning and the classical statistical theory. The authors could mention its relationship and could go to further intriguing problems such as the analysis of meta-learning using lasso-type regularization.

In my option, the theoretical contribution of this paper is quite limited.

Some more comments below:
- In each task, the same number of data is assumed to be observed. Was that just for the sake of simplicity? How is the theoretical results modified when the sample size varies from task to task?
- In Proposition 1, the meaning of the Cov(\nabla\ell_t) is not very understandable.
- The authors analyzed the statistical properties of the meta-learning using realizable linear models, or in other words, the noise-free models. The noise-free model is extremal situation, and a more general situation could be studied.
- In Corollary 6, can the author provide an intuitive reason why splitting with zero training and all validation is optimal.
- In section 4.2, the proportional limit was studied. Is it possible to derive a theoretical bound in the case that T, n, and d take comparable numbers?
- The authors did not provide any interpretation of the conclusion of Corollary 8. Showing a persuasive reason is of great help to the readers.

---

> ### Author Response · Authors · 2020-11-19
> **Response**
>
> Thank you for the valuable feedback! We respond to the comments as follows.
>
> **Delta over classical statistical theory, more sophisticated regularization**
>
> Yeah, we are aware of methods such as AIC, BIC, or lasso for reducing the overfitting. However, even though our L2 (ridge) regularization is conceptually simpler than these and somewhat easier to analyze in standard linear regression, it is much harder to analyze in our meta-learning setting.
>
> For example, our analysis of the optimally tuned train-train method involves non-trivial analyses of the spectrum such as random matrix theory and Steiltjes transform, which are not part of the classical statistical theory. Our analysis of the train-val method also involves novel technical arguments about the optimal regularization that are not present in classical statistical theory.
>
> Given that such a simple and natural L2 regularized inner algorithm already yields new and subtle results like our Theorem 4 (prev. Corollary 8) for meta-learning, we believe our results have enough technical novelty and significance.
>
> **Corollary 6 (now Corollary 8) -- intuitions why no train & all validation is optimal?**
>
> Our intuition is that validation data is more “valuable” in this case than training data, as the final loss is evaluated on the validation data. If the validation data is small, then the loss function as a function of $w_0$ will be very noisy even if the training data is large.
>
> **Other scaling regimes of $T, n, d$, e.g. they are all proportional**
>
> We think this is a good research question indeed. However, we believe finding other scaling regimes where we can compare train-train and train-val can be difficult, as our comparison is only on the level of leading constant (in the AsymMSE), and specifying the constant is only possible by taking some asymptotics. If we only compare the O(.) style rates, then train-train likely has the same rate as train-val.
>
> We think $d=\Theta(Tn)$ may be one other possible regime to compare the constants (as suggested by AnonReviewer2). However, as we noted in that response, we believe we currently lack the technical tool for handling that case.
>
> **A more persuasive interpretation of Corollary 8 (now Thm 4)**
>
> We believe the superior asymptotic MSE of the train-train method is due to the fact that the train-train method is able to use the data more efficiently (all the data goes into both training and validation) than the train-val method. We also note that while it is unclear whether this is theoretically true on other models as well, our real data experiments in Section 5.2 does support this finding on meta-learning with deep networks as well.
>
> We have added the above discussion in our revision in the “Implications” paragraph after Thm 4.

---

> > ### Comment · AnonReviewer1 · 2020-11-23
> > **the score 4 to 5**
> >
> > Thank you so much for your thoughtful response. Most of my concern was resolved. I think that the authors' explanation of the interpretation of some theoretical results is convincing. Also, I understood the problem of other scaling regimes is out of this paper.
> >
> > However, I still have a concern about the technical contribution of this paper. The term "classical statistics" I used was not very appropriate. The random matrix theory has also been intensively used as the standard tool of multivariate statistics to analyze the variance-covariance estimators' statistical properties, as shown in the famous textbook by Anderson. Hence, I still believe that this paper's technical contribution is rather limited, though the application to meta-learning is interesting. However, I could not find relevant references to show my belief. That's the reason that I raised my score.

---

> > ### Author Response · Authors · 2020-11-25
> > **Response + paper updates**
> >
> > Thank you again for your follow-up!
> >
> > Re the distinction between “classical statistics” and random matrix theory: We totally agree :), and we are aware of RMT being extensively used also in recent work e.g. for analyzing high-dimensional ridge regression, or double descent. We believe our result could be an interesting addition to this list, as our meta-learning results are new and technique-wise our results are not directly implied by these prior work.
> >
> > We have also revised our paper again and added a non-asymptotic analysis which enables us to relax the condition of Theorem 4 to $T=\Omega(d)$ (instead of $T\to\infty$), along with new experiments in the $T=O(d)$ setting and leave-one-out cross-validation. Please feel free to take a look this revision (our revision log and response to AnonReviewer2 contains more details).

---

### Official Review · AnonReviewer2 · 2020-10-25
**More discussions are required for the model settings.**

**Rating:** 5
**Confidence:** 3

**Review:**

This paper compares two approaches of data splitting in meta-learning: train-validation split and train-train split. This paper shows that the best of the two approaches depends on the specific problem.

Strength: It is easy to follow the paper and it is very convincing why the train-train approach could be better than the train-validation approach.

Weakness: I am not an expert in meta-learning and I could be wrong with my comments. But my concerns are:

1) Both inner loop and outer loop are solving the linear regression problems. In fact, because the estimate of the inner loop $\mathcal{A}(w_0; X, y)$ is linear in $w_0$, we can combine the inner loop and outer loop together, and our estimate for $w_0$ is a linear regression estimate where the new features are constructed by the original features and response, and the loss is not only averaged by the sample size $n$ of each task but also the number of tasks $T$. It is unclear to me how ''this problem captures
the essence of meta-learning with non-linear models (such as neural networks) in practice'' as claimed on page 2. Furthermore, as $T$ goes to infinity, it becomes the classic linear regression regime where dimension fixed and sample size goes to infinity. Because of this, it is intuitive and looks straightforward to check that the train-validation split gives a consistent estimate and the train-train split gives a consistent estimate in the realizable setting but not in general. I would say to improve the significance of the paper, the authors could like at the regime where d, n, T both grows to infinity at certain rates e.g. $d=\Theta(Tn)$.

2) One major reason that the train-validation split is inferior to the train-train split is that a significant amount of data is missed in the training. My question is why we do not apply K-fold cross-validation instead of a simple train-validation split. In the case of ridge regression, it is possible to do a leave-one-out cross-validation efficiently, i.e., we can obtain the leave-one-out prediction risk without fitting the regression $n$ times but using the estimate trained by the entire data. For general meta-learning, K-fold cross-validation should be a better approach than a train-validation split. In the linear regression setting, leave-one-out cross-validation should be a more proper approach. Then it will be kind of fair to compare K-fold cross-validation with the train-train split and it will be interesting to see train-train split can be still better.

3) Because of the linear regression settings with $T\rightarrow \infty$, I think it is closely related to a Bayesian hierarchical model with the Gaussian prior on the coefficients (due to the ridge penalty in the inner loop). Is the Bayesian estimator be consistent as well when $T\rightarrow infty$? Some discussion about this connection would be good.

Because of these concerns, I currently recommend a reject. But it is possible that I will change my mind if the authors can address my concerns well.

---

> ### Author Response · Authors · 2020-11-19
> **Response**
>
> Thank you for your valuable feedback. We respond to the questions as follows.
>
> **Significance of T -> infinity regime**
>
> We believe our result is significant and not a trivial consequence of classical statistics theory. We agree that our inner and outer loop can be combined and give closed-form expressions for our estimators (train-train and train-val). However, our problem is still very different from classical asymptotic analysis of linear regression.
>
> The key difference is that our train-train and train-val estimators have a two-level structure (see, e.g. Eq (10)), and thus are more complicated than “pooled linear regression” type estimators. In order to compare these two methods, we need to perform a “$T\to\infty$ asymptotics with fixed $n$”, rather than a pooled $Tn\to\infty$ asymptotics. Further, as $T\to\infty$ but $d,n$ are still comparable, our analysis required a precise understanding of the spectrum of $X_t^\top X_t$ using techniques such as the Stieltjes transform (e.g. Theorem 9), as well as how the information between different tasks can be shared. These analyses are not present in classical statistics theory.
>
> We also remark that our main technical contribution is in comparison of the asymptotic MSE (Theorem 4) for the two methods. Our consistency results (Prop 1-2, Thm 3) are agreeably more straightforward and presented for the purpose of setting up our Thm 4.
>
> **Why our problem captures essence of nonlinear meta-learning**
>
> Initially, we claimed this as our inner loop algorithm (biased ridge regression) is similar to gradient descent based finetuning of neural networks, which is used in practice (such as MAML).
>
> Further, although our model is linear rather than neural networks, we already see the subtle, non-trivial comparison of train-train and train-val in Thm 4. Therefore we believe our problem already conveys an interesting theoretical bit for meta-learning in general.
>
> **Other scaling regimes, e.g. $d=\Theta(Tn)$**
>
> We think this is an interesting research question. $d=\Theta(Tn)$ is another regime where we can compare the constants for the train-train and train-val method, as both methods will have a $O(1)$ excess risk. However, we believe this regime is technically not readily approachable, as we must instead pool the $Tn$ data from different tasks and analyze much more complicated random matrices. (The random matrices will be more involved than linear regression with $Tn$ data because of the meta-learning structure.)
>
> We also remark that our $T\gg d, n$ regime can indeed be a realistic setting for modeling the practice. For example, if we do 5-way classification on ImageNet (with 1000 classes), there would be $T={1000 \choose 5}$ distinct tasks, which is much larger than the dimension / number of samples per task.
>
> **Cross-validation**
>
> We think this is a great question too -- doing a “roulette” K-fold multiple split should certainly reduce the variance of the train-val method (and make better use of the data).
>
> We added a set of CV experiments in our simulation setting (Appendix E) and we find that K-fold CV can indeed improve over the naive train-val method, but still performs worse than train-train. We conjecture this may also be true in theory. However, currently it seems unclear to us how to rigorously prove this in theory, as cross-validation introduces undesired dependency in the loss functions (and makes the losses not iid).
>
> **Connection to Bayesian estimator**
>
> We agree that our train-train meta-learning algorithm is close to the Bayesian estimator with a certain prior. We did some calculations and found out that the Bayesian estimator under a pretty natural prior is not exactly the same as our train-train algorithm. Therefore, even though classical results show that the Bayesian estimator is consistent (under the right model), such results do not imply our result. We have added the detailed derivation and some discussions of the Bayesian estimator in Appendix C.

---

> > ### Comment · AnonReviewer2 · 2020-11-21
> > **Increased the score from 4 to 5**
> >
> > I am glad that the K-fold cross-validation result is added to the empirical results. That is the main reason I have increased the score to 5. I will increase to 6 or 7 if the following issues have been addressed.
> >
> > 1) I still think T goes to infinity while d,n is fixed is not a preferable setting for this linear regression model set up. It is true that d, n is still fixed and we need to apply random theory results. But I think the proof is quite standard especially under the isotropic Gaussian feature setting. For example, the differentiate trick Reviewer 4 has mentioned has already been used in Dobriban's paper and Hastie's paper. I would like to see some generalizations of the current result, either isotropic Gaussian assumption has been relaxed (see Hastie's paper or Wu's paper for results for the more general covariance matrix) or other non-trivial scaling settings have been analyzed.
> >
> > 2) I am happy that the K-fold cross-validation result is added. But I do feel that the sample size of n=20 is too small for any asymptotic results to be accurate and K=4 is too small. One can easily empirically check that the K-fold cross-validation prediction risk estimate is biased in the proportion limit, i.e., $n/d\rightarrow c$ (or theoretically check it using double descent literature). A more accurate risk estimate would be leave-one-out cross-validation (or K=n). I know it is time-consuming to run the estimation n times. But in the setting of ridge regression, using matrix inversion lemma, it is possible to express the leave-one-out risk estimate exactly using only the full sample estimate. It would be more convincing if the leave-one-out results are also worse. And, I do believe, one can obtain some theoretical results based on the leave-one-out cross-validation.
> >
> >
> > [1] Dobriban, E., & Wager, S. (2018). High-dimensional asymptotics of prediction: Ridge regression and classification. The Annals of Statistics, 46(1), 247-279.
> > [2] Hastie, T., Montanari, A., Rosset, S., & Tibshirani, R. J. (2019). Surprises in high-dimensional ridgeless least squares interpolation. arXiv preprint arXiv:1903.08560.
> > [3] Wu, D., & Xu, J. (2020). On the Optimal Weighted $\ell_2 $ Regularization in Overparameterized Linear Regression. arXiv preprint arXiv:2006.05800.

---

> > > ### Author Response · Authors · 2020-11-23
> > > **Will update on the suggested directions**
> > >
> > > Thank you for the score increase and the quick follow-up!
> > >
> > > We are still actively thinking about the suggested directions, and will post another update ASAP (before the reviewing period ends). Currently, we think that extending to non-Gaussian / non-isotropic features may be much harder than Hastie's or Wu's results for ridge regression. The excess risk in their ridge regression problem is $\|\hat{\beta} - \beta\|_{\Sigma}^2$ where $\Sigma$ is the input covariance, and the $\Sigma$ cancels nicely with other stuff so that RMT is easily applicable in their analysis. In contrast, the excess risk of our meta-learning problem turns out to be not $\Sigma$ but a more sophisticated matrix that does not have a closed form and could not result in a nice cancellation.

---

> > > ### Author Response · Authors · 2020-11-25
> > > **Update on suggested questions**
> > >
> > > Thank you again for following-up with the thoughtful questions!
> > >
> > > **Other scaling regimes of d, n, T; non-asymptotic analysis**
> > >
> > > We thought about this problem again and found out a way to perform a non-asymptotic analysis on the two meta-learning algorithms. We can now show our Theorem 4 to the more relaxed, non-asymptotic setting where $d,n$ are either $O(1)$ or in the proportional limit, and $T = \Omega(d)$. This shows that our main theoretical result (Thm 4) does not rely critically on the asymptotic setting of $T\to\infty$ can hold more generally.
> > >
> > > We have added the statement and a detailed proof sketch for this result (for the train-train estimator) in Appendix F.
> > >
> > > Finally, we added a simulation experiment under $n=100$, $T=300$, and $d/n\in[0,3]$ (Figure 4). This is a setting where $n, d, T$ are jointly proportional and T is not substantially larger than $(n,d)$. We found that the (non-asymptotic) MSE agrees excellently with our theoretical prediction in Thm 4 & Appendix F, which further provides justification of our non-asymptotic analysis in Appendix F.
> > >
> > > **Leave-one-out cross-validation (LOO CV)**
> > >
> > > We tried out LOO CV experimentally under two settings, $n=40$ and $n=200$. In both settings, we let the train-validation split be $(n_1, n_2)=(n-1, 1)$, and do a $n$-fold LOO CV. As suggested, we used the matrix inversion lemma, which helped accelerate the inverse computations (on rank-one perturbations of the full ridge data matrix).
> > >
> > > We find interesting results (Figure 3, Appendix E):
> > > * At $n=40$, the LOO CV estimator still performs worse than the train-train estimator.
> > > * At $n=200$, the LOO CV estimator performs better than the train-train estimator for some values of $d/n$.
> > >
> > > We think these results do suggest that $n$-fold LOO CV may have the chance to perform better than the train-train method in theory (when the regularization is optimally tuned), likely due to its stronger variance reduction effect. How to analyze the LOO CV can be a good question for future work.
> > >
> > > We also remark that although interesting, this new experiment should not undermine our main result that train-train > train-val with no cross-validation, as the later is still the common practice in meta-learning.

---

### Official Review · AnonReviewer4 · 2020-10-26
**Interesting, non-trivial theoretical result**

**Rating:** 6
**Confidence:** 2

**Review:**

In meta-learning, a common practice is to do a train/validation split of the data within each that, so that optimization of meta-parameters is performed on validation, not training, losses. In this paper, the author argue that this split is important for correcting model misspecification, but if the model is correctly specified, not doing a split might actually lead to better learning rates. They provide theoretical justification under a simple linear model, and some experiments on synthetic and real data.

Disclaimer: I have good expertise in random matrix theory but I am not as knowledgeable about the meta-learning literature. Thus, my review will mostly focus on the results themselves, with the hope that other reviewers can better evaluate how this work compares to other work in the area.

Globally, I think the results of this paper are interesting. The proof of Theorem 7 (the trickiest result) is clever. I think the high-level structure of the paper could be improved and results better presented, but the "core" of the paper holds its ground. Per the disclaimer I cannot judge the novelty of the work. Overall I would lean towards acceptance.

[pros]
1. The central result of the paper is, in my mind, Corollary 8. Its precursor, Theorem 7, has a clever proof that was different from what I expected. Namely, I thought it would be based on linear spectral statistics results as found in Bai and Silverstein (2004), but instead they noticed their statistic of interest can be computed from the nice trick of directly differentiating (a generalization of) the Stieltjes transform. It's very elegant.

2. I think Propositions 2-3 are the other interesting results, namely by showing that consistency of the methods are intimately linked to model specification.

3. To obtain a workable theoretical analysis, the authors analyzed a simple linear model, (6). However, this model also leads to counterintuitive results, such as Corollary 6. What happens when the model is very flexible, like a neural net? Issues of model misspecification become less relevant, so one might think we are more in the setting of Section 4. However, it's possible that the gap in learning rates between train-val and train-train is an artifact of how simple the linear model is. I think the "real data" experiments in Section 5 are interesting in that respect and suggest that it remains true with deep models, so this is an important contribution as well.

[cons]
1. I think the structure of the paper could be much improved. Since Corollary 8 is the most important result, it should be given the status of a Theorem. Theorem 4, 5, and 7 should disappear. I think Corollary 6 and its ensuing discussion should disappear as well. Theorem 4 is ok, but maybe should be given a lesser status. I think if Proposition 2-3 could be merged into a single Theorem it would be nice too. I would try to eliminate or shorten as much as possible Section 2.1, I found this a bit distracting. The rest of Section 2 is fine for me. Otherwise, I would really emphasize the two scenarios: under model misspecification (Section 3) and under correct model specification with a working example, the linear model (Section 4), perhaps by changing the titles of the current sections. Finally, as discussed in my "pro" point 3, the "real data" experiments in Section 5 are really important, and it's a pity a lot of details are relegated to the appendix. Since I suggest cutting a lot of theoretical results, I would maybe bring back a lot of details currently dumped in Section A.2, since I think this deep, real data example is really crucial for making the case that the paper is relevant to normal practice.

2. I would like to see more details as to why interchange of the derivative and both the limit and the integral is justified in the proof of Theorem 7 (uniform convergence).

Other comments:
- Calling \hat{w}_{0, T}^{tr-val} in Proposition 2 an "algorithm" doesn't make much sense, I think "estimator" is more accurate.
- There seems to be a major typo in the statement of Theorem 7: you talk of the train-val method but the math has tr-tr written.

References

[1] Bai, Z. D., and Jack W. Silverstein. "CLT for Linear Spectral Statistics of Large-Dimensional Sample Covariance Matrices." Annals of Probability (2004): 553-605.

---

> ### Author Response · Authors · 2020-11-19
> **Response**
>
> Thank you for the positive feedback! We respond to the comments as follows.
>
> **Structure of paper**
>
> We liked your suggestions about the structure! We have restructured our paper accordingly:
> * Section 4 now only contains two main theorems, Thm 3 (for consistency) and Thm 4 (for comparing the asymptotic MSEs). All other theorems and corollaries are deferred to the Appendix and sketched in Section 4.2 as proof highlights.
> * The majority of the deep meta-learning experiment now appears in the main text in Section 5.2. Specifically, Table 1 now appears in the main paper.
> * Section 2.1 was shortened.
>
> Please feel free to take a look at our new structure and let us know of any further suggestions.
>
> **Interchange of derivative and limit / expectation**
>
> Good catch. We have added a detailed justification of these interchanges in Appendix B.4.1.
>
> We have also fixed the typos according to your comments.

---

### Official Review · AnonReviewer3 · 2020-10-28
**A theory to show the importance of train-val split in meta-learning**

**Rating:** 6
**Confidence:** 3

**Review:**

The authors verify the importance of train-validation split in meta-learning theoretically, which is commonly used in the meta-learning paradigms. By analyzing the linear centroid meta-learning problem, the authors show that the splitting method converges to the optimal prior as expected, whereas the non-splitting method does not in general, without structural assumptions on the data. The authors validate the theories empirically through both simulations and real meta-learning tasks.

The paper provides new insights on the usage of the train-validation splits in meta-learning theoretically. Some popular methods such as iMAML are also used as comparisons in the experiments.

Here are some suggestions:
1. Will the batch-size, the meta-training way influence the results in Table 1. More ablations like Table 2 should be investigated.

2. The authors could discuss whether the theory can be applied to other kinds of meta-learning methods and how the theoretical results can guide the design of new meta-learning methods.

---

> ### Author Response · Authors · 2020-11-19
> **Response**
>
> Thank you for the positive feedback. We respond to the comments as follows.
>
> >Will the batch-size, the meta-training way influence the results in Table 1. More ablations like Table 2 should be investigated
>
> For the batch-size, our experiments in Table 1 are done with 4 tasks per minibatch for all the settings. We have tuned this parameter within {2, 4, 6, 8} for the train-train method in our early stage experiments (unfortunately we did not record the numbers) and found this parameter does not have a significant effect on the final accuracy (the effect is less than 1% in absolute sense).
>
> We’re glad you liked our experiment in Table 2! We also remark that our Table 2, apart from being an ablation, also suggests that our Corollary 8 (previously Corollary 6) may also hold more broadly on real data as well: the best split ratio for the train-val method is to put most data on the validation.
>
> >Can our theory guide the design of new meta-learning methods
>
> Our main theoretical result suggests that the train-train method + optimally tuned regularization may be a strong alternative to the train-validation split. Therefore, we recommend meta-learning researchers to try out the train-train method (no train-validation split) on top of any meta-learning algorithm. Experimentally, we verified the superiority of train-train on one specific meta-learning algorithm; testing this on other algorithms should be an interesting direction for future work.

---

### Author Response · Authors · 2020-11-19
**Revision posted**

We have made a revision to the paper to incorporate the reviewers’ suggestions. The main changes are
* (Most of) Our **deep meta-learning experiment** is now moved into the main text as Section 5.2. We encourage reviewers to give this experiment section a look.
* We **restructured Section 4** so that there are only two main theorems --- Theorem 3 for consistency, and Thm 4 for the comparison of asymptotic MSE (our main result). Other intermediate results are now sketched in Section 4.2 as proof highlights, with the full versions deferred to the Appendix. We specifically remark that our previous main result, Corollary 8, now becomes Theorem 4.
* A new simulation experiment for the train-val method with **cross-validation** (roulette multiple splits), showing that cross-validation can improve over naive unaveraged train-val, but still performs worse than train-train (Appendix E).
* Some new discussions on the connection to the Bayesian estimator (Appendix C).

Our major changes are marked in red for clarity. We emphasize that most of the additions and changes are for discussions or paper restructuring. Our main theoretical / empirical results were not modified.

---

> ### Author Response · Authors · 2020-11-25
> **Revision updated**
>
> We have updated our paper again and added the following materials according to the reviewers' suggestions:
> * We added a non-asymptotic analysis of the train-train estimator, which could relax our Theorem 4 (comparison of train-train and train-val) to the more relaxed setting of $T=\Omega(d)$ instead of $T\to\infty$. We’ve added the statement and a detailed proof sketch of this result in Appendix F.
> This new analysis demonstrates that the $T\to\infty$ asymptotic setting is not crucial to our theory and our results may hold more broadly in other scaling regimes.
> * New simulation suggesting that $T=O(d)$ may indeed be enough for our main theoretical result to hold (Figure 4), justifying our relaxation in Appendix F.
> * New simulation on the leave-one-out cross-validation (Figure 3), showing that it can perform better than train-train under certain regimes.
> Remark: This is not to confuse with our main result that analyzes the train-val method with no cross-validation, which is the standard practice in meta-learning.

---

### Decision · Program_Chairs · 2021-01-07
**Final Decision**

**Decision:**

Reject

**Comment:**

The authors investigate theoretical properties of meta-learning. In particular, the train-validation split to tackle the linear centroid meta-learning problem is investigated in asymptotically regimes (a non-asymptotic analysis of the train-train estimator has been added later on). It is shown that the train-validation method has statistical consistency, while the train-train method has a statistical bias to the centroid. Yet, in the noise-free setting both methods have statistical consistency. Furthermore, the train-train method is superior to the train-validation method in the sense of the asymptotic MSE. Based on the asymptotic analysis the optimal ratio of the data splitting for the train-validation method is derived. The theoretical findings are corroborated by some numerical experiments.
The main theoretical result suggests that the train-train method + optimally tuned regularization is a strong alternative to the train-validation split, hence the authors' recommendation that for meta-learning train-train method should be preferred.

The reviewers and the area chair appreciated the theoretical findings and the subsequent effort to improve the paper the authors put in place during the discussion period. Unfortunately, the tight competition among papers in this year's edition of the conference makes this specific submission not compelling enough for publication.
I would like to encourage the authors to sumbit to another ML venue in the near future, while considering improvements in their experimental validation (as also suggested by some reviewers).